# Understanding Task Vectors in In-Context Learning: Emergence, Functionality, and Limitations

## Abstract

Task vectors offer a compelling mechanism for accelerating inference in in-context learning (ICL) by distilling task-specific information into a single, reusable representation. Despite their empirical success, the underlying principles governing their emergence and functionality remain unclear. This work proposes the *Linear Combination Conjecture*, positing that task vectors act as single in-context demonstrations formed through linear combinations of the original ones. We provide both theoretical and empirical support for this conjecture. First, we show that task vectors naturally emerge in linear transformers trained on triplet-formatted prompts through loss landscape analysis. Next, we predict the failure of task vectors on representing high-rank mappings and confirm this on practical LLMs. Our findings are further validated through saliency analyses and parameter visualization, suggesting an enhancement of task vectors by injecting multiple ones into few-shot prompts. Together, our results advance the understanding of task vectors and shed light on the mechanisms underlying ICL in transformer-based models.

## 1 Introduction

In-context learning (ICL) is a core capability of large language models (LLMs), allowing them to perform new tasks without parameter updates by conditioning on a few input-output examples in the prompt [2]. Unlike traditional training, ICL relies on attention-based mechanisms to infer task structure directly from context. This surprising generalization ability has led to growing interest in uncovering the principles of learning purely from contextual examples [21, 3, 4, 15, 5].

A recent work investigates the task vector method [7] (concurrent works include function vectors [16] and in-context vectors [13]), a technique that distills underlying task information from ICL demonstrations into a single vector. Typically, ICL prompts are structured as sequences of triplets, each encoding a semantic mapping, in addition to a query at the end (e.g., *"hot → cold, up → down, day → night, dark →"*). Task vectors are then extracted from the hidden states of the last (→) token. Once obtained, these vectors can be injected into the same position in new prompts (e.g., *"big →"*), enabling the model to generalize to unseen inputs in a zero-shot fashion.

Task vectors have been shown to naturally emerge even in small transformer models trained from scratch on synthetic data [24], suggesting that their formation is a general property of attention-based architectures. Recent studies further demonstrate that task vectors can be enhanced by aggregating hidden states across multiple layers and multiple arrow tokens [12]. Beyond language models, task vectors are also found effective in large-scale visual [8] and multi-modal [9] models.

Despite their empirical effectiveness, the underlying mechanism of task vectors, especially how they emerge, function, and encode task information, remains poorly understood. This paper takes a step toward unveiling the principles behind it by introducing the following conjecture:

Submitted to 39th Conference on Neural Information Processing Systems (NeurIPS 2025). Do not distribute.

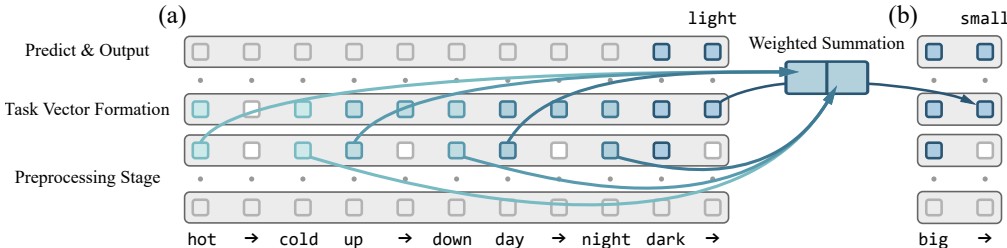

Figure 1: Overview of task vector and our main conjecture. (a) Task vector emerges during ICL as a linear combination of preceding in-context demonstrations. (b) It can then be injected into zero-shot prompts and functions as a single, representative demonstration, facilitating efficient prediction.

> **Linear Combination Conjecture**
>
> *The injected task vector functions as a single in-context demonstration,*
> *formed through a linear combination of the original demonstrations (hidden states).*

Figure 1 provides an intuitive illustration for our conjecture. In the following sections, we validate this conjecture through various empirical and theoretical perspectives. These analyses comprehensively explain how task vectors naturally emerge within attention-based model architectures, effectively encode task-related information, and facilitate inference in zero-shot prompts. Our work advances the understanding of the underlying mechanisms behind ICL, clarifying both the efficacy and limitations of task vectors in transformer-based LLMs. The highlights of this paper are as follows:

- **Theoretical Justification in Linear Transformers:** We theoretically characterize the critical points of linear-attention transformers and demonstrate how they solve random linear regression tasks through embedding concatenation and gradient descent. With a triplet-formatted input prompt structure, task vectors naturally emerge at arrow tokens as linear combinations of the in-context demonstrations. These vectors serve as redundancy against information loss induced by dropout, thereby improving robustness. Empirically, the learned linear model parameters closely align with the predicted structure and successfully replicate the task vector mechanism.

- **Empirical Verification in Practical LLMs:** We visualize the information flow in LLMs with saliency analysis and observe patterns consistent with linear models, suggesting they share similar underlying mechanisms. According to our conjecture, inference with task vectors is analogous to 1-shot ICL, which is inherently limited to rank-one meta-predictors under the gradient descent perspective. To validate this, we introduce a series of bijection tasks that are provably unsolvable by rank-one predictors, and empirically confirm this failure in real-world transformers. Building on these insights, we enhance the standard task vector method by injecting multiple vectors into few-shot prompts, resulting in consistent performance gains across a range of ICL tasks.

## 2 Setting: Random Linear Regression with Linear-Attention Transformers

**Notations:** We write $[n] = \{1, \cdots, n\}$. The Hadamard product is denoted by $\circ$, and the Kronecker product by $\otimes$. The identity matrix of dimension $n$ is denoted by $I_n$, while $0_n$ and $0_{m \times n}$ represent zero vectors or matrices of the corresponding dimensions. Subscripts are omitted when the dimensions are clear from context. We define $\mathcal{M}(M) = \left\{ \Lambda \in \mathbb{R}^{\dim(M)} \mid \Lambda = M \circ A, \ A \in \mathbb{R}^{\dim(M)} \right\}$ as the set of masked matrices induced by the binary mask $M$. For a general matrix $A$, the element at the $i$-th row and $j$-th column is denoted by $A_{i,j}$, and the sub-block from rows $i$ to $k$ and columns $j$ to $l$ is denoted by $A_{i:k,j:l}$. $\mathrm{diag}(A_1, \cdots, A_n)$ represents the block-diagonal matrix constructed by $\{A_i\}_{i=1}^n$.

**Random Linear Regression:** Following the settings in literature [6, 17, 1, 20], we consider training linear transformers on random instances of linear regression. Let $\{x_i\}_{i=1}^{n+1}$, where $x_i \in \mathbb{R}^d$, denote covariates drawn i.i.d. from distribution $P_x$, and let $\{w_i\}_{i=1}^d$, where $w_i \in \mathbb{R}^d$, denote coefficients drawn i.i.d. from distribution $P_w$. Define the coefficient matrix as $W = [w_1 \ \cdots \ w_d]^\top \in \mathbb{R}^{d \times d}$. The responses are then generated as $y_i = W x_i$ for $i \in [n+1]$. We denote by $X, Y \in \mathbb{R}^{d \times n}$ the matrices whose columns are $x_i$ and $y_i$, respectively, for $i \in [n]$. The query covariate and response are denoted by $x_{\text{test}} = x_{n+1}$ and $y_{\text{test}} = y_{n+1}$ respectively.

**Linear Self-Attention Transformer:** Following prior works [17, 1, 20], we consider transformers composed of linear self-attention layers. Let $Z_0 \in \mathbb{R}^{2d \times d_p}$ denote the input matrix constructed from $X$, $Y$ and $x_{\text{test}}$ but excluding $y_{\text{test}}$, where $d_p$ denotes the number of tokens. The transformer is defined by stacking $L$ attention blocks with skip connections, where the $l$-th layer is expressed as:

$$Z_l = Z_{l-1} + \frac{1}{n} \text{Attn}_{V_l, Q_l}(Z_{l-1}), \qquad \text{Attn}_{V,Q}(Z) = VZM(Z^\top Q Z). \tag{1}$$

Here, the trainable parameters are $\{V_l, Q_l\}_{l=1}^L$, where $V_l \in \mathbb{R}^{2d \times 2d}$ represents a reparameterization of the projection and value matrices, and $Q_l \in \mathbb{R}^{2d \times 2d}$ denotes the query and key matrices. Following the work [1], we adopt a masking matrix $M = \text{diag}(I_{d_p-1}, 0)$ to prevent attention from earlier tokens to the final one. The output of the transformer is defined as $\text{TF}(Z_0; \{V_l, Q_l\}_{l=1}^L) = (Z_L)_{(d+1:2d), d_p}$ (i.e., the latter half of the last column). This definition aligns with the structure of the input $Z_0$, which will be further discussed in subsequent sections. During training, the parameters are optimized to minimize the expected ICL risk over random linear regression instances:

$$\mathcal{L}(\{V_l, Q_l\}_{l=1}^L) = \mathbb{E}_{Z_0, W} \left\| \text{TF}(Z_0; \{V_l, Q_l\}_{l=1}^L) + W x_{\text{test}} \right\|_2^2. \tag{2}$$

## 3 Emergence of Task Vectors in Linear-Attention Transformers

Firstly, we present theoretical evidence indicating that task vectors naturally arise even in simple linear transformers. Specifically, we analyze the loss landscape of the in-context risk, focusing on the properties of its critical points. As a startup, recall the standard linear regression setup [1, 20], where the $(x_i, y_i)$ pairs for each demonstration are concatenated to form the input prompt:

$$Z_0 = \begin{bmatrix} X & x_{\text{test}} \\ Y & 0 \end{bmatrix} = \begin{bmatrix} x_1 & x_2 & \cdots & x_n & x_{\text{test}} \\ y_1 & y_2 & \cdots & y_n & 0 \end{bmatrix} \in \mathbb{R}^{2d \times (n+1)}. \tag{3}$$

According to existing analyses [1, 25, 14], each attention layer in this setting performs one step of gradient descent on the coefficient matrix $W$. Specifically, the theoretically optimal single-layer (possibly nonlinear) attention [10] implements the following predictive function [1] when the covariates are drawn from $P_x = \mathcal{N}(0, I_d)$, by selecting $V_1 \propto \text{diag}(0_{d \times d}, I_d)$ and $Q_1 \propto \text{diag}(I_d, 0_{d \times d})$:

$$\text{TF}(Z_0; (V_1, Q_1)) = -\frac{1}{n} Y \sigma(X)^\top \sigma(x_{\text{test}}), \quad \text{where } \sigma : \mathbb{R}^d \mapsto \mathbb{R}^r \text{ is a kernel function.} \tag{4}$$

Here, we abbreviate $[\sigma(x_1) \quad \cdots \quad \sigma(x_n)]$ as $\sigma(X)$. Equation (4) employs $W' \propto Y \sigma(X)^\top$ as an estimate of coefficient matrix $W$, yielding prediction $\hat{y}_{\text{test}} = W' \sigma(x_{\text{test}})$. In this paper, we consider alternative settings more reflective of practical scenarios, where $x_i$ and $y_i$ are separated as distinct tokens. As noted [26], such separation necessitates the usage of position encodings for bi-directional attention. Following prior analysis [11], we assume that position encodings are appended to the input tokens, and reformulate the layer-wise update rule of self-attention as:

$$\text{Attn}_{V,Q}(Z) = VZM \begin{bmatrix} Z^\top & P^\top \end{bmatrix} Q \begin{bmatrix} Z \\ P \end{bmatrix}, \quad \text{where } P \in \mathbb{R}^{d_p \times d_p}. \tag{5}$$

For analytical tractability, we take $P = I_{d_p}$ as one-hot position encodings. Inspired by the parameter structure in [1] and eq. (4), we further impose the following constraints on the trainable parameters:

$$V_l = \text{diag}(A_l, B_l), \quad Q_l = \text{diag}(C_l, 0_{d \times d}, D_l), \quad \text{where } A_l, B_l, C_l \in \mathbb{R}^{d \times d}, \ D_l \in \mathbb{R}^{d_p \times d_p}. \tag{6}$$

These parameterizations ensure that the projection and attention operations act independently on the covariate, response, and positional components of the input. This structural decoupling is essential for understanding how the transformer identifies the dependency between each $(x_i, y_i)$ pair and revealing the actual optimization algorithm being executed by the model. The proofs for the main theoretical results in this paper are available in Appendix B.

### 3.1 Warm-up: Learning with Pairwise Demonstrations

We begin by analyzing the optimization of linear transformers on pairwise demonstrations. Following previous approach [6, 19, 22], we decompose each demonstration in eq. (3) into a pair of tokens $Z_0^i = \begin{bmatrix} x_i & 0 \\ 0 & y_i \end{bmatrix} \in \mathbb{R}^{2d \times 2}$ to better reflect the practical ICL prompt structure:

$$Z_0 = \begin{bmatrix} Z_0^1 & \cdots & Z_0^n & Z_0^{\text{test}} \end{bmatrix} = \begin{bmatrix} x_1 & 0 & \cdots & x_n & 0 & x_{\text{test}} & 0 \\ 0 & y_1 & \cdots & 0 & y_n & 0 & 0 \end{bmatrix} \in \mathbb{R}^{(2d) \times (2n+2)}. \tag{7}$$

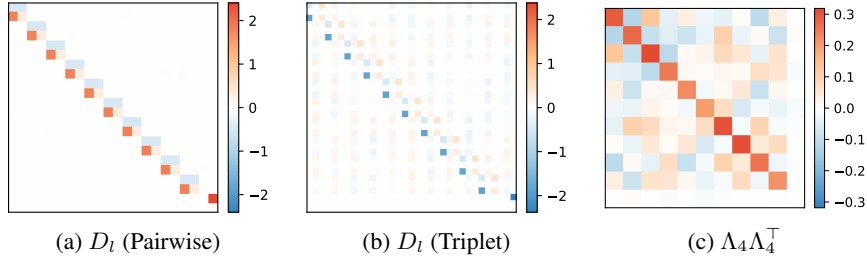

| (a) $D_l$ (Pairwise) | (b) $D_l$ (Triplet) | (c) $\Lambda_4 \Lambda_4^\top$ |

Figure 2: Visualization of learned $D_l$ weights. (a) Pairwise demonstrations yield a block-diagonal structure aligned with Theorem 1. (b) Triplet demonstrations yield a richer structure aligned with Theorem 2. (c) The learned matrix $\Lambda_4$ has nearly orthonormal rows as suggested by Proposition 3.

The following theorem suggests that certain critical points of the in-context risk effectively solve the regression problem by first concatenating each pair of $(x_i, y_i)$ into the same tokens, and then executing a variant of the gradient descent algorithm to compute the prediction. To simplify notation, we denote $A = \{A_l\}_{l=1}^L$ (similarly for $B$, $C$, and $D$) and present:

**Theorem 1 (Critical Points; Pairwise Demonstrations).** *Assume $P_x = \mathcal{N}(0, \Sigma)$, $P_w = \mathcal{N}(0, \Sigma^{-1})$ with some $\Sigma \in \mathbb{R}^{d \times d}$ satisfying $\Sigma \succ 0$. Define $\mathcal{S}_I, \mathcal{S}_\Sigma \subset \mathbb{R}^{d \times d}$ and $\mathcal{S}_P \subset \mathbb{R}^{d_p \times d_p}$ as*

$$\mathcal{S}_I = \{\lambda I_d \mid \lambda \in \mathbb{R}\}, \quad \mathcal{S}_\Sigma = \{\lambda \Sigma^{-1} \mid \lambda \in \mathbb{R}\}, \quad \mathcal{S}_P = \{\mathrm{diag}(I_n \otimes \Lambda_1, \Lambda_2) \mid \Lambda_1, \Lambda_2 \in \mathbb{R}^{2 \times 2}\}.$$

*Consider optimizing an $L$-layer linear transformer with pairwise demonstrations and parameter configuration given in eq. (6), we then have*

$$\inf_{A,B \in \mathcal{S}_I^L, \, C \in \mathcal{S}_\Sigma^L, \, D \in \mathcal{S}_P^L} \sum_{H \in A \cup B \cup C \cup D} \left\| \nabla_H \mathcal{L}\left( \{V_l, Q_l\}_{l=1}^L \right) \right\|_F^2 = 0.$$

To understand the behavior of these critical points within a self-attention layer, we fix $\Sigma = I_d$ and take $A_l, B_l = I_d$, $C_l = -\lambda I_d$, and $D_l = \mathrm{diag}(I_n \otimes \Lambda_1, \Lambda_2)$. Let the first and last $d$ rows of $Z_l$ be denoted by $X_l$ and $Y_l$, respectively. Under these settings, the update rule of each layer becomes:

$$Z_l = Z_{l-1} - \lambda Z_{l-1} M X_{l-1}^\top X_{l-1} + \begin{bmatrix} Z_{l-1}^1 \Lambda_1 & \cdots & Z_{l-1}^n \Lambda_1 & Z_{l-1}^{\mathrm{test}} \mathrm{diag}(1,0) \Lambda_2 \end{bmatrix}. \quad (8)$$

The above update can be decomposed into the following two distinct components:

- **Gradient Descent:** The first component, $Z_l \leftarrow Z_{l-1} - \lambda Z_{l-1} M X_{l-1}^\top X_{l-1}$, implements the GD++ algorithm [17]. This variant enhances convergence speed over standard gradient descent by improving the condition number of the Gram matrix $X_{l-1}^\top X_{l-1}$. Notably, this operation modifies only $X_l$ but not $Y_l$ for the first layer, as implied by the structure of $Q_l$ (eq. (6)).

- **Embedding Concatenation:** The second component, $Z_l^i \leftarrow Z_{l-1}^i + Z_{l-1}^i \Lambda_1$ for $i \in [n]$, mixes each pair of $(x_i, y_i)$ tokens. Given that $x_i$ and $y_i$ tokens are initially linearly separable as in our formulation, this operation concatenates each $(x_i, y_i)$ pair, thereby *transforming pairwise demonstrations into the original single-token format*. For the query token $Z_l^{\mathrm{test}}$, this operation copies $x_{\mathrm{test}}$ into the final token, reconstructing the structure in eq. (3), where each non-final token directly concatenates $(x_i, y_i)$ of a demonstration, and the final token contains only $x_{\mathrm{test}}$.

In summary, our analysis reveals that for pairwise demonstrations, the first attention layer leverages position encodings to distinguish between covariate and response tokens, subsequently concatenating them to form a single-token prompt structure. The remaining layers then apply the GD++ algorithm, mirroring the learning dynamics on single-token demonstrations. As a result, **an $L$-layer linear transformer allocates one layer for embedding concatenation and utilizes the remaining $L - 1$ layers to perform gradient descent**. In Figure 2a, we visualize the learned $D_l$ weights under the setting of Theorem 1, and observe that they closely match the critical point structure of $\mathcal{S}_P$.

### 3.2 Emergence of Task Vectors with Triplet Demonstrations

Next, to better reflect the prompt structure of practical ICL, we insert additional zero tokens between each pair of $(x_i, y_i)$ to simulate the arrow ($\rightarrow$) tokens. This reformulates each demonstration as a triplet $(x_i, \rightarrow, y_i)$, enabling us to analyze the critical points with these triplet demonstrations:

$$Z_0 = \begin{bmatrix} x_1 & 0 & 0 & \cdots & x_n & 0 & 0 & x_{\mathrm{test}} & 0 & 0 \\ 0 & 0 & y_1 & \cdots & 0 & 0 & y_n & 0 & 0 & 0 \end{bmatrix} \in \mathbb{R}^{(2d) \times (3n+3)}. \quad (9)$$

**Theorem 2** (**Critical Points; Triplet Demonstrations**). *Assume $P_x = \mathcal{N}(0, \Sigma)$, $P_w = \mathcal{N}(0, \Sigma^{-1})$ with some $\Sigma \in \mathbb{R}^{d \times d}$ satisfying $\Sigma \succ 0$. Define $\mathcal{S}_I, \mathcal{S}_\Sigma \subset \mathbb{R}^{d \times d}$ and $\mathcal{S}_P \subset \mathbb{R}^{d_p \times d_p}$ as*

$$\mathcal{S}_I = \{\lambda I_d \mid \lambda \in \mathbb{R}\}, \quad \mathcal{S}_\Sigma = \{\lambda \Sigma^{-1} \mid \lambda \in \mathbb{R}\},$$

$$\mathcal{S}_P = \Big\{ \mathrm{diag}(I_n \otimes \Lambda_1, \Lambda_2) + I_{n+1} \otimes \Lambda_3 + \Lambda_4 \otimes \Lambda_5 \Big|$$

$$\Lambda_1, \Lambda_2 \in \mathcal{M}\left(\begin{smallmatrix} 1 & 0 & 1 \\ 0 & 0 & 0 \\ 1 & 0 & 1 \end{smallmatrix}\right), \Lambda_3 \in \mathcal{M}\left(\begin{smallmatrix} 0 & 0 & 0 \\ 0 & 1 & 0 \\ 0 & 0 & 0 \end{smallmatrix}\right), \Lambda_4 \in \mathbb{R}^{(n+1) \times (n+1)}, \Lambda_5 \in \mathcal{M}\left(\begin{smallmatrix} 0 & 1 & 0 \\ 0 & 0 & 0 \\ 0 & 1 & 0 \end{smallmatrix}\right) \Big\}.$$

*Consider optimizing an L-layer linear transformer with triplet demonstrations and parameter configuration given in eq.* (6)*, we then have*

$$\inf_{A,B \in \mathcal{S}_I^L, \ C \in \mathcal{S}_\Sigma^L, \ D \in \mathcal{S}_P^L} \sum_{H \in A \cup B \cup C \cup D} \left\| \nabla_H \mathcal{L}\big(\{V_l, Q_l\}_{l=1}^L\big) \right\|_F^2 = 0.$$

To analyze the behavior of each attention layer, we note that the critical points for the matrices $A_l$, $B_l$, and $C_l$ remain consistent with Theorem 1, thereby implementing the GD++ algorithm. For the matrix $D_l$, we decompose its structure into three distinct components:

- **Embedding Concatenation:** The first component, $\mathrm{diag}(I_n \otimes \Lambda_1, \Lambda_2)$, mixes each pair of $(x_i, y_i)$ tokens, effectively concatenating them — analogous to the operation analyzed in the previous section. This converts all non-arrow tokens into single-token demonstrations.

- **Self Magnification:** The second component, $I_{n+1} \otimes \Lambda_3$, scales the embeddings corresponding to each arrow ($\rightarrow$) token by a fixed constant and adds them back to themselves.

- **Task Vector Formation:** The third component, $\Lambda_4 \otimes \Lambda_5$, performs a weighted summation across all demonstrations in the prompt. This operation is central to the emergence of task vectors. Let $[\beta_1 \ \cdots \ \beta_{n+1}] \in \mathbb{R}^{n \times (n+1)}$ denote the first $n$ rows of $\Lambda_4$ (we will soon show that the last row of $\Lambda_4$ converges to zero), *the first self-attention layer then outputs $n + 1$ linear combinations of the demonstrations as the hidden states for the arrow tokens*, expressed as $z_{\mathrm{tv}}^i = \left[\begin{smallmatrix} \alpha_1 X \beta_i \\ \alpha_2 Y \beta_i \end{smallmatrix}\right]$ for $i \in [n + 1]$, where $\alpha_1, \alpha_2 \in \mathbb{R}$ are the two non-zero entries of $\Lambda_5$. These vectors can then be injected into zero-shot prompts and function as single-token demonstrations.

This mechanism provides strong theoretical evidence for our linear combination conjecture, demonstrating that **task vectors naturally emerge from the optimization dynamics of linear-attention transformers operating on triplet-formatted prompts**. Notably, the structure of $\mathcal{S}_P$ closely aligns with our visualization of $D_l$ in Figure 2b, confirming our theoretical analysis. We now further investigate the structure of the weight matrix $\Lambda_4$, and present the following result:

**Proposition 3** (**Optimal Task Vector Weights**). *Assume $P_x, P_w = \mathcal{N}(0, I_d)$. Consider optimizing a 2-layer linear-attention transformer with triplet demonstrations and parameter configuration given in eq.* (6)*, and assume $C_1 = 0$. Let*

$$D_1 = \mathrm{diag}(I_n \otimes \Lambda_1, \Lambda_2) + I_{n+1} \otimes \Lambda_3 + \Lambda_4 \otimes \Lambda_5 \in \mathcal{S}_P$$

*be any minimizer of the in-context risk $\mathcal{L}\big(\{V_l, Q_l\}_{l=1}^L\big)$, we then have $\Lambda_4 \in \mathcal{S}_U$, where*

$$\mathcal{S}_U = \big\{ \Lambda \mid \Lambda\Lambda^\top = \lambda \, \mathrm{diag}(I_n, 0), \lambda \in \mathbb{R} \big\}.$$

This result suggests that the optimal $\Lambda_4$ weight matrix satisfies two key properties: (1) the last row is zero, and (2) the first $n$ rows are mutually orthonormal. These conditions imply that the learned weight vectors $\beta_1, \cdots, \beta_{n+1}$ are likely to be distinct. Therefore, the $n + 1$ task vectors produce diverse linear combinations of the demonstrations, thereby enriching the representation within the input prompt. This implication is verified in Figure 2c. While task vectors are typically extracted from the final arrow ($\rightarrow$) token in standard usage, here we consider all arrow tokens as task vectors as bi-directional attention allows each to aggregate information from the full prompt.

## 4  Validating the Linear Combination Conjecture on Bijection Tasks

We then present an empirical observation that supports our conjecture. Consider the setting where task vectors are injected into zero-shot prompts. Based on our prior analysis, the injected task vector $z_{\mathrm{tv}}$ is formed as a linear combination of the original demonstrations. As a result, we show that the injected prompt reconstructs the single-token structure in eq. (3) with only 1 demonstration:

$$Z_0 = [z_{\mathrm{test}} \quad z_{\mathrm{tv}} \quad 0] = \begin{bmatrix} x_{\mathrm{test}} & x_{\mathrm{tv}} & 0 \\ 0 & y_{\mathrm{tv}} & 0 \end{bmatrix} = \begin{bmatrix} x_{\mathrm{test}} & X\beta & 0 \\ 0 & Y\beta & 0 \end{bmatrix} \in \mathbb{R}^{2d \times 3}, \tag{10}$$

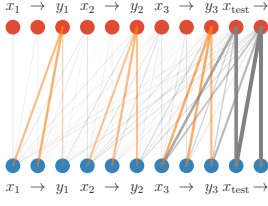 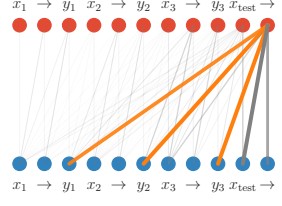 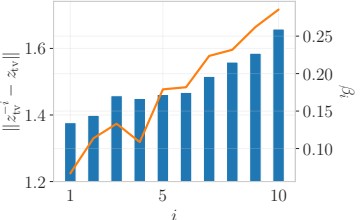

(a) Saliency Map ($l = 10$)      (b) Saliency Map ($l = 12$)      (c) Task Vector Weights

Figure 3: Visualization of saliency matrices as bipartite graphs between layer $l$ (●) and $l + 1$ (●), where edge widths indicate saliency magnitude; and variations in the extracted task vector after perturbing the $i$-th demonstration (▌), alongside the predicted weights (—) obtained by optimizing Proposition 6. (a) Each $y_i$ token is primarily attending to its corresponding $(x_i, y_i)$ pair, reflecting embedding concatenation. (b) The final ($\rightarrow$) token attends broadly to all $y_i$ tokens, indicating task vector formation — this occurs just before the optimal injection layer ($l = 13$). (c) The predicted task vector weights closely match the trend of empirical results, validating our theoretical model.

where the weight vector $\beta \in \mathbb{R}^n$ comes from the last column of $\Lambda_4$ (Theorem 2). After the first layer, the $\Lambda_2$ matrix of $\mathcal{S}_P$ moves $x_{\text{test}}$ to the last token, reducing the prompt to a single-shot, single-token demonstration. According to the optimal single-layer transformer (eq. (4)), the estimated coefficient matrix is now $W' = Y\beta(X\beta)^\top$, which is rank-one. Therefore, if our main conjecture holds, task vectors will be inherently limited in their expressiveness: *they can only realize rank-one coefficient matrices*. This implication also naturally extends to multi-layer transformers.

While our analysis is conducted on linear-attention transformers, we demonstrate that similar learning patterns also emerge within practical LLMs. Specifically, we visualize the layer-wise information flow between tokens using saliency maps [18], where the saliency score for each attention matrix is computed as $S(A_l) = \sum_h |A_{l,h} \cdot \partial \mathcal{L}/\partial A_{l,h}|$, $A_{l,h}$ denotes the attention matrix of the $h$-th head at layer $l$, and $\mathcal{L}$ is the ICL loss (i.e., the cross-entropy loss for predicting $y_{\text{test}}$). As demonstrated in Figures 3a and 3b, the saliency maps reveal certain patterns matching the ones of embedding concatenation and weighted summation. Importantly, the latter occurs immediately before the optimal task vector injection layer. This suggests that real-world models implement a similar algorithm to solve ICL tasks and, consequently, inherit the same expressiveness limitation.

To verify this, we construct a specialized class of ICL tasks, named bijection tasks. Specifically, given a bijective mapping from domain $\mathcal{X}$ to codomain $\mathcal{Y}$, one can combine it with its inverse mapping to form a new task that maps $\mathcal{X} \cup \mathcal{Y}$ onto itself. For instance, combining the "to uppercase" task with its inverse "to lowercase" yields a bijection task that maps each letter to its opposite case, and a valid ICL prompt takes the form: *"a $\rightarrow$ A, B $\rightarrow$ b, c $\rightarrow$ C, D $\rightarrow$"*. Note that this differs from task superposition [23], as each input corresponds to a unique, well-defined output. We then establish a key limitation of rank-one coefficient matrices in addressing such tasks:

**Proposition 4.** *Let $x, y \in \mathbb{R}^d$ be non-zero vectors. Then the following are equivalent: (1) There exists a rank-one matrix $W \in \mathbb{R}^{d \times d}$ such that $y = Wx$ and $x = Wy$; (2) $x = y$ or $x = -y$.*

This result highlights that *rank-one coefficient matrices cannot solve general bijection tasks*, and are restricted to only the identity mapping ($x = y$) or the negation mapping ($x = -y$). We further verify this implication in real-world LLMs: as summarized in Table 1, both ICL and the task vector method perform well on the original tasks and their inverses. Nevertheless, for the bijection tasks, while ICL preserves performance in many cases, the task vector method consistently fails, confusing examples from the two domains and yielding near-random predictions (50%). For instance, in the "to uppercase" task, task vectors can predict the correct letter but fail to distinguish between uppercase and lowercase. The only notable exceptions are the copy task (corresponding to the $x = y$ case in Proposition 4) and the antonym task (corresponding to $x = -y$).

Together, these findings empirically validate our conjecture: **the task vector approach, which is restricted to rank-one coefficient matrices, cannot solve general bijection tasks**. While a variety of ICL tasks have been explored to assess the capabilities of task vectors [7, 16, 12], the fundamental limitation of task vectors in addressing these bijection tasks has not been previously identified.

Table 1: Comparison of the accuracies of ICL and task vector on bijection tasks (Llama-7B, $n = 10$). We use gray text to indicate accuracies lower than $60\%$.

| Task | Domain $\mathcal{X}$ | Domain $\mathcal{Y}$ | Example | $\mathcal{X} \to \mathcal{Y}$ ICL | $\mathcal{X} \to \mathcal{Y}$ TV | $\mathcal{Y} \to \mathcal{X}$ ICL | $\mathcal{Y} \to \mathcal{X}$ TV | $\mathcal{X} \leftrightarrow \mathcal{Y}$ ICL | $\mathcal{X} \leftrightarrow \mathcal{Y}$ TV |
|---|---|---|---|---|---|---|---|---|---|
| To Upper | $\{a, \cdots, z\}$ | $\{A, \cdots, Z\}$ | a → A | 1.00 | 0.91 | 1.00 | 0.99 | 1.00 | 0.55 |
| Translation | English | French | hello → bonjour | 0.83 | 0.84 | 0.82 | 0.70 | 0.54 | 0.35 |
| | English | Italian | hello → ciao | 0.84 | 0.78 | 0.82 | 0.74 | 0.70 | 0.47 |
| | English | Spanish | hello → hola | 0.92 | 0.88 | 0.89 | 0.75 | 0.64 | 0.43 |
| Linguistic | Present | Gerund | go → going | 0.99 | 0.95 | 1.00 | 0.97 | 0.80 | 0.41 |
| | Present | Past | go → went | 0.98 | 0.91 | 0.99 | 0.96 | 0.52 | 0.33 |
| | Present | Past Perfect | go → gone | 0.82 | 0.82 | 0.94 | 0.65 | 0.55 | 0.33 |
| | Singular | Plural | dog → dogs | 0.88 | 0.78 | 0.94 | 0.89 | 0.76 | 0.51 |
| Copy | $\{a, \cdots, z, A, \cdots, Z\}$ | | A → A | - | | - | | 1.00 | 0.98 |
| Antonym | Adjectives | | happy → sad | 0.89 | 0.83 | | - | 0.83 | 0.73 |

## 5 Further Discussions

**Inseparable Covariates and Responses.** In our main analysis, we assume that $x_i$ and $y_i$ embeddings are linearly separable, allowing the addition $x_i + y_i$ to act a concatenation operation. However, recognizing that this assumption does not generally hold for real-world transformers, we extend our analysis to the following setting, where $x_i$ and $y_i$ are no longer linearly separable. While this still imposes a $2d$-dimensional requirement on the hidden space, such a constraint is easily satisfied in practical transformers, given the high dimensionality of their internal representations.

$$Z_0 = \begin{bmatrix} 0 & 0 & \cdots & 0 & 0 & 0 & 0 \\ x_1 & y_1 & \cdots & x_n & y_n & x_{\text{test}} & 0 \end{bmatrix} \in \mathbb{R}^{(2d) \times (2n+2)}. \tag{11}$$

We slightly modify the sparsity constraints for the first layer, and require $(D_0)_{2i,:} = 0$ for $i \in [n+1]$:

$$V_0 = \begin{bmatrix} 0 & A_0 \\ 0_{d \times d} & 0 \end{bmatrix}, \quad Q_0 = \begin{bmatrix} 0_{2d \times 2d} & 0 \\ 0 & D_0 \end{bmatrix}, \quad \text{where } A_0 \in \mathbb{R}^{d \times d}, \ D_0 \in \mathbb{R}^{d_p \times d_p}. \tag{12}$$

With these conditions, we are ready to establish the critical points for inseparable demonstrations. Note that $V_0$ and $Q_0$ do not involve $B_0$ and $C_0$, so the sequences $B$ and $C$ have size $L - 1$.

**Theorem 5.** *Under the same settings as Theorem 1, define $\mathcal{S}_I, \mathcal{S}_\Sigma \subset \mathbb{R}^{d \times d}$ and $\mathcal{S}_P \subset \mathbb{R}^{d_p \times d_p}$ as*

$$\mathcal{S}_I = \{\lambda I_d \mid \lambda \in \mathbb{R}\}, \quad \mathcal{S}_\Sigma = \{\lambda \Sigma^{-1} \mid \lambda \in \mathbb{R}\}, \quad \mathcal{S}_P = \{\text{diag}(I_n \otimes \Lambda_1, \Lambda_2) \mid \Lambda_1, \Lambda_2 \in \mathbb{R}^{2 \times 2}\}.$$

*Consider optimizing an $L$-layer linear transformer with inseparable pairwise demonstrations and parameter configuration given in eq. (12) for the first layer and eq. (6) for the remaining layers, then*

$$\inf_{A \in \mathcal{S}_I^L, \ B \in \mathcal{S}_I^{L-1}, \ C \in \mathcal{S}_\Sigma^{L-1}, \ D \in \mathcal{S}_P^L} \sum_{H \in A \cup B \cup C \cup D} \left\| \nabla_H \mathcal{L}\left(\{V_l, Q_l\}_{l=1}^L\right) \right\|_F^2 = 0.$$

This result suggests that for inseparable demonstrations, the first layer performs a functionally similar concatenation operation by "moving" the embedding of each $x_i$ to the corresponding $y_i$ position. This enables the model to reconstruct the single-token structure without linear separability.

**Optimal Weights for Causal Task Vectors.** While task vectors naturally emerge in linear transformers, their embeddings do not directly help minimize the ICL risk, as evidenced by the identical performance between pairwise and triplet formatted prompts (Figures 4a and 4b). Instead, we show that task vectors contribute to minimizing the training (i.e., LLM pretraining) risk when token-wise dropout is applied, acting as redundancies for in-context demonstrations that may be randomly dropped during training. This redundancy ensures that essential task information is preserved and continues to facilitate accurate prediction despite partial context loss.

**Proposition 6.** *Under the same settings as Proposition 3, consider adding token-wise dropouts $O_l$:*

$$Z_l = Z_{l-1} O_l + \frac{1}{n} \text{Attn}_{V_l, Q_l}(Z_{l-1}) O_l, \quad \text{where } O_l = \text{diag}(o_l^1, \cdots, o_l^{d_p}), \ o_l^i \overset{i.i.d.}{\sim} \text{Bern}(p).$$

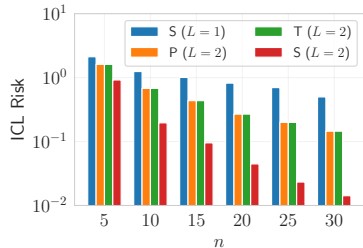 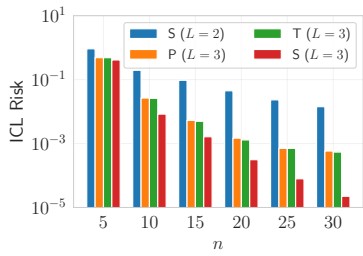 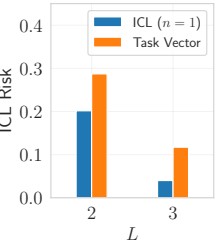

(a) Single- vs. Multi-Token ($L = 2$)     (b) Single- vs. Multi-Token ($L = 3$)     (c) ICL vs. TV

Figure 4: (a, b) Comparison of the best ICL risk achieved using single (S), pairwise (P), and triplet (T) formatted prompts. (c) Performance comparison between 1-shot ICL and task vector.

*Then any minimizer $\Lambda_4$ of the in-context risk $\mathcal{L}\big(\{V_l, Q_l\}_{l=1}^L\big)$ satisfies $(\Lambda_4)_{n+1,:} = 0$ and:*

$$(\Lambda_4)_{1:n,:} \propto \arg\min_\Lambda \ c_1 \|\Lambda\|_4^4 + c_2 \sum_{i=1}^n \|\Lambda_{i,:}\|_2^4 + c_3 \sum_{j=1}^{n+1} \|\Lambda_{:,j}\|_2^4 + c_4 \big\|\Lambda\Lambda^\top\big\|_F^2, \ \ s.t. \ \|\Lambda\|_F^2 = 1.$$

*where $c_1, \cdots, c_4$ are non-negative constants depending on $V_l$, $Q_l$, and $p$.*

This result suggests that dropout introduces additional higher-order regularization on the task vector weights, encouraging them to distribute more uniformly across demonstrations. Furthermore, when considering causal attention (i.e., enforcing $\Lambda_4$ to be upper-triangular), it induces a decaying weight pattern from later to earlier demonstrations, which is also consistently observed in practical transformer models as evidenced in Figure 3c. While dropout is not always applied during LLM pre-training or fine-tuning, the injection of position encodings and use of normalization act as alternative sources of perturbation, thereby promoting the emergence of such redundancy.

**Extra EOS Tokens.** In our theoretical analysis, we consistently impose an additional zero token at the end of the input prompt. While this token can be interpreted as an EOS token in practical models, such a design choice is uncommon in standard ICL tasks. We justify this modeling decision with:

**Proposition 7** (Informal). *Given any $L$-layer, single-head, $d$-dimensional linear-attention transformer with EOS tokens, there exists an equivalent $L$-layer, two-head, $2d$-dimensional linear-attention transformer operating without EOS tokens.*

This equivalence suggests that the same learning dynamics can be realized through multi-head architectures without relying on explicit EOS tokens. Specifically, one head in this setting is dedicated to task vector formation, while the other handles ICL prediction. This separation allows the model to retain the functional role of the EOS token implicitly within its hidden states. Consequently, our prior theoretical analysis can be naturally extended to practical models that omit explicit EOS tokens.

## 6 Experimental Studies

### 6.1 Synthetic Results with Random Linear Regression

In this section, we validate our critical points analysis with synthetic linear regression tasks. Specifically, we examine the achievable ICL risk of linear transformers trained with single-token (eq. (3)), pairwise (eq. (7)), and triplet (eq. (9)) demonstrations. We set the input dimension to $d = 4$ and $P_x = P_w = \mathcal{N}(0, I_d)$. For each setting, we train multiple models with different random seeds and report the minimum ICL risk achieved as a proxy for the global optimum. The comparative results across different numbers of layers $L$ and demonstration formats are shown in Figures 4a and 4b.

These results support our theoretical analysis: when trained with pairwise or triplet demonstrations, the transformer recovers the GD++ algorithm similar to the single-token case. Notably, the performance of $L$-layer transformers with pairwise (P) and triplet (T) demonstrations closely aligns, indicating a shared underlying learning pattern. Moreover, their performance consistently lies between that of single-token (S) case $L$-layer and $(L-1)$-layer models. The observed improvement over the $(L-1)$-layer single-token baselines comes from the additional GD++ performed solely on $x_i$ tokens in the first layer, effectively acting as a "half-step" of gradient descent.

Table 2: Accuracy comparison between standard ICL (Baseline), the task vector method (TaskV), and our strategy (TaskV-M). The experiment is conducted on Llama-13B with $n = 10$.

| Method | | Knowledge | Algorithmic | Translation | Linguistic | Bijection | Average |
|---|---|---|---|---|---|---|---|
| 0-shot | Baseline | $6.90 \pm 2.08$ | $15.60 \pm 1.72$ | $7.00 \pm 1.65$ | $12.44 \pm 1.74$ | $8.27 \pm 1.33$ | $10.28 \pm 0.98$ |
| | TaskV | $\mathbf{68.80} \pm 2.66$ | $\mathbf{86.20} \pm 1.61$ | $\mathbf{73.53} \pm 0.91$ | $\mathbf{85.24} \pm 1.80$ | $\mathbf{50.67} \pm 2.32$ | $\mathbf{72.26} \pm 1.01$ |
| 1-shot | Baseline | $69.50 \pm 3.86$ | $73.67 \pm 1.56$ | $57.80 \pm 2.01$ | $56.22 \pm 1.57$ | $44.76 \pm 2.44$ | $58.11 \pm 0.63$ |
| | TaskV | $79.50 \pm 2.35$ | $88.47 \pm 0.75$ | $\mathbf{80.67} \pm 2.56$ | $\mathbf{89.11} \pm 0.84$ | $60.44 \pm 2.07$ | $78.79 \pm 0.77$ |
| | TaskV-M | $\mathbf{81.30} \pm 2.80$ | $\mathbf{89.53} \pm 0.65$ | $80.13 \pm 2.14$ | $88.71 \pm 0.62$ | $\mathbf{61.78} \pm 0.96$ | $\mathbf{79.34} \pm 0.37$ |
| 2-shot | Baseline | $78.80 \pm 3.30$ | $85.07 \pm 1.37$ | $75.67 \pm 2.64$ | $76.80 \pm 1.18$ | $56.49 \pm 2.87$ | $72.92 \pm 0.59$ |
| | TaskV | $84.60 \pm 2.11$ | $88.40 \pm 0.68$ | $\mathbf{84.33} \pm 0.92$ | $\mathbf{90.13} \pm 0.92$ | $62.44 \pm 2.16$ | $80.82 \pm 0.42$ |
| | TaskV-M | $\mathbf{85.70} \pm 1.63$ | $\mathbf{89.27} \pm 1.10$ | $84.13 \pm 1.15$ | $89.64 \pm 0.86$ | $\mathbf{64.49} \pm 2.02$ | $\mathbf{81.48} \pm 0.37$ |
| 3-shot | Baseline | $86.20 \pm 2.69$ | $88.07 \pm 1.06$ | $80.00 \pm 1.67$ | $84.04 \pm 1.19$ | $62.18 \pm 1.52$ | $78.51 \pm 0.42$ |
| | TaskV | $90.20 \pm 2.23$ | $88.67 \pm 0.89$ | $\mathbf{86.27} \pm 2.31$ | $92.31 \pm 0.48$ | $66.53 \pm 0.94$ | $83.53 \pm 0.41$ |
| | TaskV-M | $\mathbf{90.30} \pm 1.50$ | $\mathbf{89.87} \pm 0.83$ | $86.07 \pm 2.17$ | $\mathbf{92.36} \pm 0.72$ | $\mathbf{68.13} \pm 0.76$ | $\mathbf{84.15} \pm 0.52$ |
| 4-shot | Baseline | $84.80 \pm 2.06$ | $88.07 \pm 0.61$ | $83.27 \pm 1.82$ | $88.89 \pm 1.91$ | $67.16 \pm 1.47$ | $81.52 \pm 0.66$ |
| | TaskV | $88.70 \pm 1.69$ | $89.53 \pm 1.34$ | $86.27 \pm 1.08$ | $\mathbf{92.76} \pm 0.54$ | $70.44 \pm 1.35$ | $84.66 \pm 0.39$ |
| | TaskV-M | $\mathbf{89.60} \pm 1.43$ | $\mathbf{91.00} \pm 1.01$ | $\mathbf{87.20} \pm 0.62$ | $92.36 \pm 1.44$ | $\mathbf{72.53} \pm 0.94$ | $\mathbf{85.64} \pm 0.29$ |

Additionally, we successfully reproduce the task vector method in linear transformers. Specifically, we extract the hidden state of the final ($\rightarrow$) token from triplet demonstrations after the first layer, and inject this vector into zero-shot prompts consisting of only $x_{\text{test}}$. To simulate the effect of layer normalization used in practical transformers, we normalize the task vectors before inference and the output vectors before ICL risk evaluation. As shown in Figure 4c, the performance of task vectors is parallel to that of standard ICL with a single in-context example. This validates our conjecture that the injected task vector effectively acts as a single demonstration.

## 6.2 Enhancing the Task Vector Method

We further explore an enhancement to the original task vector method. According to our previous analysis, a single injected task vector may not provide sufficient information for inference on complex tasks (e.g., bijection tasks). Moreover, in linear-attention models, each ($\rightarrow$) token functions as an individual in-context demonstration during the gradient descent phase and thus contributes equally to the ICL risk. Motivated by this, we extend the standard task vector method, which modifies only the final arrow token, and propose a multi-vector variant that injects into every single arrow token in few-shot prompts. This enriched injection scheme enables the model to leverage multiple new demonstrations, thereby providing a more informative and distributed context for prediction.

We compare our multi-vector injection strategy (TaskV-M) against standard $N$-shot ICL (Baseline) and the original task vector method (TaskV). For each $N$-shot prompt, we generate $N + 1$ distinct ICL prompts to produce $N + 1$ task vectors, which are then used to replace the embeddings of all arrow tokens in the input. For each task, performance is evaluated over 50 randomly sampled prompts, with mean accuracy and standard deviation reported across 5 independent trials. The final results, summarized in Table 2, span a diverse set of ICL task types, including Knowledge, Algorithmic, Translation, Linguistic, and Bijection, showing that TaskV-M consistently outperforms TaskV, especially on the more challenging bijection tasks. These findings support our analysis that every arrow token contributes meaningfully to the model's ICL capability.

## 7 Conclusion

This paper proposes the linear combination conjecture as a plausible explanation for the emergence and functionality of task vectors in ICL. We support this conjecture with both empirical observations and theoretical analysis, demonstrating how task vectors naturally arise under triplet-formatted demonstrations in simple linear transformer models, and why this method inherently fails on general bijection tasks. While the conjecture may not yet offer a complete characterization of ICL dynamics, it provides a new perspective on the underlying mechanisms and offers a promising direction for interpreting intermediate hidden states in modern transformer-based language models.

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

## A  Auxiliary Lemmas

**Lemma 8** (Proposed in [1]). *Given positive objective function $f(A)$ taking parameters $A = \{A_i\}_{i=1}^n$, where $A_i \in \mathbb{R}^{d_i \times d_i}$. Let $\mathcal{S} = \Pi_{i=1}^n \mathcal{S}_i \subset \Pi_{i=1}^n \mathbb{R}^{d_i \times d_i}$ be a predefined parameter subspace. Define $\widetilde{A}(t, R_i) = \{A_1, \cdots, A_i + tR_i, \cdots, A_n\}$ given $i \in [1, n]$, $R_i \in \mathbb{R}^{d_i \times d_i}$ and $t \in \mathbb{R}$. If for any $A \in \mathcal{S}$ and $R_i \in \mathbb{R}^{d_i \times d_i}$, there exists $\widetilde{R}_i \in \mathcal{S}_i$ such that*

$$\frac{\mathrm{d}}{\mathrm{d}t} f\left(\widetilde{A}(t, \widetilde{R}_i)\right)\bigg|_{t=0} \leq \frac{\mathrm{d}}{\mathrm{d}t} f\left(\widetilde{A}(t, R_i)\right)\bigg|_{t=0},$$

*then we have*

$$\inf_{A \in \mathcal{S}} \sum_{i=1}^n \|\nabla_{A_i} f(A)\|_F^2 = 0.$$

*Proof.* This lemma is proved as part of the main theorems in [1]. We rearrange the proof here to accommodate arbitrary function of matrices. Firstly, notice that for any $R = \{R_i\}_{i=1}^n \in \Pi_{i=1}^n \mathbb{R}^{d_i \times d_i}$,

$$\sum_{i=1}^n \frac{\mathrm{d}}{\mathrm{d}t} f\left(\widetilde{A}(t, \widetilde{R}_i)\right)\bigg|_{t=0} = \frac{\mathrm{d}}{\mathrm{d}t} f(A + tR)\bigg|_{t=0}.$$

Therefore, the provided precondition is equivalent to stating that for any $A \in \mathcal{S}$ and $R \in \Pi_{i=1}^n \mathbb{R}^{d_i \times d_i}$, there exists $\widetilde{R} \in \mathcal{S}$ such that:

$$\frac{\mathrm{d}}{\mathrm{d}t} f\left(A + t\widetilde{R}\right)\bigg|_{t=0} \leq \frac{\mathrm{d}}{\mathrm{d}t} f(A + tR)\bigg|_{t=0}.$$

Let $R = -\nabla_A f(A)$, we then have

$$\frac{\mathrm{d}}{\mathrm{d}t} f(A + tR)\bigg|_{t=0} = \left\langle \frac{\mathrm{d}f(A - t\nabla_A f(A))}{\mathrm{d}(A - t\nabla_A f(A))}, \frac{\mathrm{d}(A - t\nabla_A f(A))}{t} \right\rangle\bigg|_{t=0}$$
$$= \langle \nabla_A f(A), -\nabla_A f(A) \rangle = -\|\nabla_A f(A)\|_F^2.$$

If the infimum of $\|\nabla_A f(A)\|_F^2$ is not zero but some positive value $p$, then the $\mathcal{S}$-constrained gradient flow induced by $\widetilde{R}$ will lead to unbounded descent:

$$\frac{\mathrm{d}}{\mathrm{d}t} f\left(A + t\widetilde{R}\right)\bigg|_{t=0} \leq -p.$$

This contradicts the fact that $f(A) \geq 0$ and concludes the proof.  $\square$

The following lemma is an extension of Lemma 5 in [1] by accommodating multivariate $y$ samples as well as enabling a wider range of demonstration and transformer parameter configurations.

**Lemma 9.** *Let $x_1, \cdots, x_{n+1}$ be i.i.d. samples from an input distribution, and let $W$ be sampled independently of $\{x_i\}_{i=1}^{n+1}$. Let $Z_0 \in \mathbb{R}^{(2d) \times N}$, where $N \in \mathbb{Z}$, be constructed of form*

$$Z_0 = \begin{bmatrix} * & \cdots & * & * \\ * & \cdots & * & 0_d \end{bmatrix} \in \mathbb{R}^{(2d) \times N},$$

*where the $*$ parts can be arbitrarily constructed from $\{x_i\}_{i=1}^{n+1}$ and $W$. Let $\widetilde{Z}_0$ be defined as replacing the zero part of $Z_0$ by $y_{n+1}$:*

$$\widetilde{Z}_0 = \begin{bmatrix} * & \cdots & * & * \\ * & \cdots & * & y_{n+1} \end{bmatrix} \in \mathbb{R}^{(2d) \times N}.$$

*Let $\widetilde{Z}_l$ be the output of the $l$-th layer of the linear transformer, and let $\widetilde{X}_l, \widetilde{Y}_l \in \mathbb{R}^{d \times N}$ be the first and last $d$ rows of $\widetilde{Z}_l$, respectively. Suppose that the $\{Q_l\}_{l=1}^L$ matrices are of form*

$$Q_l = \begin{bmatrix} \underbrace{*}_{d\ columns} & 0_{(2d+d_p) \times d} & \underbrace{*}_{d_p\ columns} \end{bmatrix},$$

*Then the in-context risk of this $L$-layer linear transformer is equivalent to*

$$\mathcal{L}\left(\{V_l, Q_l\}_{l=1}^L\right) = \mathbb{E}_{\widetilde{Z}_0, W}\left[\mathrm{tr}\left((I_N - M)\widetilde{Y}_L^\top \widetilde{Y}_L (I_N - M)\right)\right]. \tag{13}$$

*Proof.* Let the $V_l$ and $Q_l$ matrices be represented as:

$$V_l = \begin{bmatrix} V_l^1 \\ V_l^2 \end{bmatrix}, \quad Q_l = \begin{bmatrix} Q_l^1 & 0 & Q_l^2 \end{bmatrix},$$

where $V_l^1, V_l^2 \in \mathbb{R}^{d \times 2d}, Q_l^1 \in \mathbb{R}^{(2d+d_p) \times d}, Q_l^2 \in \mathbb{R}^{(2d+d_p) \times d_p}$. Then the update rule in eq. (5) can be rephrased as

$$X_l = X_{l-1} + \frac{1}{n} V_l^1 Z_{l-1} M \big[ Z_{l-1}^\top, P \big] \big( Q_l^1 X_{l-1} + Q_l^2 P \big),$$

$$Y_l = Y_{l-1} + \frac{1}{n} V_l^2 Z_{l-1} M \big[ Z_{l-1}^\top, P \big] \big( Q_l^1 X_{l-1} + Q_l^2 P \big).$$

Let $\Delta_Z = \widetilde{Z}_0 - Z_0$, i.e. an all-zero matrix except that the last half of the last column is $y_{n+1}$. Let $\Delta_X$ and $\Delta_Y$ be its first and last $d$ rows respectively, then $\Delta_X = 0$ and $\Delta_Y = \begin{bmatrix} 0 & \cdots & 0 & y_{n+1} \end{bmatrix}$. Note that $\widetilde{Z}_l = Z_l + \Delta_Z$ holds for $l = 0$ trivially. Now suppose it holds for some $l = k - 1$, then

$$\begin{aligned}
\widetilde{X}_k &= \widetilde{X}_{k-1} + \frac{1}{n} V_k^1 \widetilde{Z}_{k-1} M \big[ \widetilde{Z}_{k-1}^\top, P \big] \big( Q_k^1 \widetilde{X}_{k-1} + Q_k^2 P \big) \\
&= X_{k-1} + \frac{1}{n} V_k^1 Z_{k-1} M \big[ Z_{k-1}^\top, P \big] \big( Q_k^1 X_{k-1} + Q_k^2 P \big) \\
&\quad + \frac{1}{n} V_k^1 \Delta_Z M \big[ Z_{k-1}^\top, P \big] \big( Q_k^1 X_{k-1} + Q_k^2 P \big) \\
&\quad + \frac{1}{n} V_k^1 Z_{k-1} M \big[ \Delta_Z^\top, 0_{d_p \times d_p} \big] \big( Q_k^1 X_{k-1} + Q_k^2 P \big) \\
&\quad + \frac{1}{n} V_k^1 \Delta_Z M \big[ \Delta_Z^\top, 0_{d_p \times d_p} \big] \big( Q_k^1 X_{k-1} + Q_k^2 P \big) \\
&= X_{k-1} + \frac{1}{n} V_k^1 Z_{k-1} M \big[ Z_{k-1}^\top, P \big] \big( Q_k^1 X_{k-1} + Q_k^2 P \big) = X_k,
\end{aligned}$$

where the last step holds by noticing that $\Delta_Z M = 0$. Similarly, one can prove that

$$\widetilde{Y}_k = Y_{k-1} + \Delta_Y + \frac{1}{n} V_k^2 Z_{k-1} M \big[ Z_{k-1}^\top, P \big] \big( Q_k^1 X_{k-1} + Q_k^2 P \big) = Y_k + \Delta_Y.$$

Therefore, it holds that for any $l \in [1, L]$, $\widetilde{Z}_l = Z_l + \Delta_Z$. Recall the in-context risk in eq. (2):

$$\begin{aligned}
\mathcal{L}\big(\{V_l, Q_l\}_{l=1}^L\big) &= \mathbb{E}_{Z_0, W} \big\| (Z_L)_{(d+1:2d), N} + y_{n+1} \big\|_2^2 \\
&= \mathbb{E}_{Z_0, W} \big\| (Y_L + \Delta_Y)(I_N - M) \big\|_2^2 \\
&= \mathbb{E}_{\widetilde{Z}_0, W} \Big[ \text{tr}\Big( (I_N - M) \widetilde{Y}_L^\top \widetilde{Y}_L (I_N - M) \Big) \Big].
\end{aligned}$$

The proof is complete. $\qquad\square$

# B  Proof of Theoretical Results

## B.1  Proof of Proposition 4

*Proof.* We will first prove sufficiency. Let $W = ab^\top$ be a rank-one matrix, where $a, b \in \mathbb{R}^d$. The given conditions imply that $x = Wy = WWx = ab^\top ab^\top x$, we then have $b^\top x = b^\top ab^\top ab^\top x = (b^\top a)^2 b^\top x$. Since $b^\top x \neq 0$, we can conclude that $b^\top a = \pm 1$. Then, $x = ab^\top ab^\top x = \pm ab^\top x = \pm y$.

To prove the necessity, it suffices to show that selecting $W = xx^\top / \|x\|_2^2$ when $x = y$ satisfies the given conditions (alternatively, select $W = -xx^\top / \|x\|_2^2$ when $x = -y$). $\qquad\square$

## B.2  Proof of Theorem 1

*Proof.* To enhance the readability of the notations in this proof, we will drop the constant $\frac{1}{n}$ factor in linear attention. Furthermore, we will simplify $\widetilde{Z}_0$, $\widetilde{X}_0$ and $\widetilde{Y}_0$ in Lemma 9 as $Z_0$, $X_0$ and $Y_0$

respectively. This results in different definitions compared to the original ones, but we will not refer to the original definitions in the remainder of this proof.

$$Z_0 = \begin{bmatrix} X_0 \\ Y_0 \end{bmatrix} = \begin{bmatrix} x_1 & 0 & \cdots & x_n & 0 & x_{\text{test}} & 0 \\ 0 & y_1 & \cdots & 0 & y_n & 0 & y_{\text{test}} \end{bmatrix} \in \mathbb{R}^{(2d) \times (2n+2)}.$$

Let $Z_l$ be the output of the $l$-th layer of the transformer, and let $X_l, Y_l \in \mathbb{R}^{d \times (2n+2)}$ denote the first and last $d$ rows of $Z_l$, respectively. Under the constraint in eq. (6), we can verify that

$$\begin{aligned} X_l &= X_{l-1} + A_l X_{l-1} M(X_{l-1}^\top C_l X_{l-1} + D_l), \\ Y_l &= Y_{l-1} + B_l Y_{l-1} M(X_{l-1}^\top C_l X_{l-1} + D_l). \end{aligned} \tag{14}$$

In the following analysis, we will use $f(A \leftarrow B)$ to denote the result of the function $f$ of $A$ when replacing the value of $A$ with $B$. Additionally, we denote $f(A \leftarrow B * A)$ as $f(A \overset{*}{\leftarrow} B)$ for any operator $*$. Therefore, $f(A \overset{+}{\leftarrow} B) = f(A \leftarrow A + B)$. We also denote $f(A \overset{\times}{\leftarrow} B) = f(A \leftarrow BA)$ and $f(A \overset{\diamond}{\leftarrow} B) = f(A \leftarrow AB)$ for convenience.

Our goal is proving that, for any $E \in A \cup B \cup C \cup D$ and an arbitrary matrix $R \in \mathbb{R}^{d \times d}$ ($\mathbb{R}^{d_p \times d_p}$ for $D$), there exists $\widetilde{R} \in \mathcal{S}_I$ ($\mathcal{S}_\Sigma$ for $C$, $\mathcal{S}_P$ for $D$) such that

$$\left. \frac{\mathrm{d}}{\mathrm{d}t} \mathcal{L}(E \overset{+}{\leftarrow} t\widetilde{R}) \right|_{t=0} \leq \left. \frac{\mathrm{d}}{\mathrm{d}t} \mathcal{L}(E \overset{+}{\leftarrow} tR) \right|_{t=0}. \tag{15}$$

Let $\overline{X}_0 = [0, x_1, \cdots, 0, x_{\text{test}}]$ be a function of $X_0$, we then have $Y_0 = W \overline{X}_0$. Let $U_\perp \in \mathbb{R}^{d \times d}$ be a uniformly sampled random orthonormal matrix, and let $U_\Sigma = \Sigma^{1/2} U_\perp \Sigma^{-1/2}$. One can verify that $U_\Sigma^{-1} = \Sigma^{1/2} U_\perp^\top \Sigma^{-1/2}$. By applying Lemma 9 and the fact that $X_0 \overset{d}{=} U_\Sigma X_0$, we have that for any given matrix $R$,

$$\begin{aligned} &\left. \frac{\mathrm{d}}{\mathrm{d}t} \mathcal{L}(E \overset{+}{\leftarrow} tR) \right|_{t=0} \\ &= \left. \frac{\mathrm{d}}{\mathrm{d}t} \mathbb{E}_{X_0, W} \left[ \mathrm{tr}\left( (I - M) Y_L^\top (E \overset{+}{\leftarrow} tR) Y_L (E \overset{+}{\leftarrow} tR)(I - M) \right) \right] \right|_{t=0} \\ &= 2 \mathbb{E}_{X_0, W} \left[ \mathrm{tr}\left( (I - M) Y_L^\top \left. \frac{\mathrm{d}}{\mathrm{d}t} Y_L (E \overset{+}{\leftarrow} tR) \right|_{t=0} (I - M) \right) \right] \\ &= 2 \mathbb{E}_{X_0, W, U_\perp} \left[ \mathrm{tr}\left( (I - M) Y_L^\top (X_0 \overset{\times}{\leftarrow} U_\Sigma) \left. \frac{\mathrm{d}}{\mathrm{d}t} Y_L (X_0 \overset{\times}{\leftarrow} U_\Sigma, E \overset{+}{\leftarrow} tR) \right|_{t=0} (I - M) \right) \right]. \end{aligned}$$

Next, we will show that eq. (15) holds for each one of $A_i, B_i, C_i, D_i$ for any $i \in [1, L]$.

**1. Equation (15) holds for $A_i$.**

We first show that for any $l \in [1, L]$, the following equations hold:

$$X_l(X_0 \overset{\times}{\leftarrow} U_\Sigma) = U_\Sigma X_l, \tag{16}$$

$$\left. \frac{\mathrm{d}}{\mathrm{d}t} X_l(X_0 \overset{\times}{\leftarrow} U_\Sigma, A_i \overset{+}{\leftarrow} tR) \right|_{t=0} = U_\Sigma \left. \frac{\mathrm{d}}{\mathrm{d}t} X_l(A_i \overset{+}{\leftarrow} t U_\Sigma^{-1} R U_\Sigma) \right|_{t=0}. \tag{17}$$

It is straightforward to verify that eq. (16) holds for $l = 0$. Now suppose that eq. (16) holds for some $l = k - 1$, we then have

$$\begin{aligned} &X_k(X_0 \overset{\times}{\leftarrow} U_\Sigma) \\ &= X_{k-1}(X_0 \overset{\times}{\leftarrow} U_\Sigma) + A_l X_{k-1}(X_0 \overset{\times}{\leftarrow} U_\Sigma) M\left( X_{k-1}^\top (X_0 \overset{\times}{\leftarrow} U_\Sigma) C_l X_{k-1}(X_0 \overset{\times}{\leftarrow} U_\Sigma) + D_l \right) \\ &= U_\Sigma X_{k-1} + A_l U_\Sigma X_{k-1} M\left( X_{k-1}^\top U_\Sigma^\top C_l U_\Sigma X_{k-1} + D_l \right) \\ &= U_\Sigma \left( X_{k-1} + A_l X_{k-1} M\left( X_{k-1}^\top C_l X_{k-1} + D_l \right) \right) = U_\Sigma X_k, \end{aligned}$$

where the third equality follows by noticing that when $A_l = a_l I_d$ and $C_l = c_l \Sigma^{-1}$, we have $A_l U_\Sigma = U_\Sigma A_l$ and $U_\Sigma^\top C_l U_\Sigma = C_l$. This concludes the proof of eq. (16).

458  We now turn to the proof of eq. (17). Notice that when $l < i$, we naturally have

$$\frac{\mathrm{d}}{\mathrm{d}t}X_l(X_0 \overset{\times}{\leftarrow} U_\Sigma, A_i \overset{\pm}{\leftarrow} tR)\Big|_{t=0} = U_\Sigma \frac{\mathrm{d}}{\mathrm{d}t}X_l(A_i \overset{\pm}{\leftarrow} tU_\Sigma^{-1}RU_\Sigma)\Big|_{t=0} = 0.$$

459  When $l = i$, it is easy to verify that

$$\begin{aligned}
\frac{\mathrm{d}}{\mathrm{d}t}X_l(X_0 \overset{\times}{\leftarrow} U_\Sigma, A_i \overset{\pm}{\leftarrow} tR)\Big|_{t=0} &= RU_\Sigma X_{l-1}M(X_{l-1}^\top U_\Sigma^\top C_l U_\Sigma X_{l-1} + D_l) \\
&= U_\Sigma \cdot U_\Sigma^{-1}RU_\Sigma M(X_{l-1}^\top C_l X_{l-1} + D_l) \\
&= U_\Sigma \frac{\mathrm{d}}{\mathrm{d}t}X_l(A_i \overset{\pm}{\leftarrow} tU_\Sigma^{-1}RU_\Sigma)\Big|_{t=0}.
\end{aligned}$$

460  Now suppose that eq. (17) holds for some $l = k - 1 \geq i$, one can verify that:

$$\begin{aligned}
&\frac{\mathrm{d}}{\mathrm{d}t}X_k(X_0 \overset{\times}{\leftarrow} U_\Sigma, A_i \overset{\pm}{\leftarrow} tR)\Big|_{t=0} \\
&= \frac{\mathrm{d}}{\mathrm{d}t}X_{k-1}(X_0 \overset{\times}{\leftarrow} U_\Sigma, A_i \overset{\pm}{\leftarrow} tR)\Big|_{t=0} + \frac{\mathrm{d}}{\mathrm{d}t}A_k X_{k-1}(X_0 \overset{\times}{\leftarrow} U_\Sigma, A_i \overset{\pm}{\leftarrow} tR)M \\
&\quad \cdot \left(X_{k-1}^\top(X_0 \overset{\times}{\leftarrow} U_\Sigma, A_i \overset{\pm}{\leftarrow} tR)C_k X_{k-1}(X_0 \overset{\times}{\leftarrow} U_\Sigma, A_i \overset{\pm}{\leftarrow} tR) + D_k\right)\Big|_{t=0} \\
&= \frac{\mathrm{d}}{\mathrm{d}t}X_{k-1}(X_0 \overset{\times}{\leftarrow} U_\Sigma, A_i \overset{\pm}{\leftarrow} tR)\Big|_{t=0} \\
&\quad + A_k \frac{\mathrm{d}}{\mathrm{d}t}X_{k-1}(X_0 \overset{\times}{\leftarrow} U_\Sigma, A_i \overset{\pm}{\leftarrow} tR)\Big|_{t=0} M\left(X_{k-1}^\top(X_0 \overset{\times}{\leftarrow} U_\Sigma)C_k X_{k-1}(X_0 \overset{\times}{\leftarrow} U_\Sigma) + D_k\right) \\
&\quad + A_k X_{k-1}(X_0 \overset{\times}{\leftarrow} U_\Sigma)M \frac{\mathrm{d}}{\mathrm{d}t}X_{k-1}^\top(X_0 \overset{\times}{\leftarrow} U_\Sigma, A_i \overset{\pm}{\leftarrow} tR)\Big|_{t=0} C_k X_{k-1}(X_0 \overset{\times}{\leftarrow} U_\Sigma) \\
&\quad + A_k X_{k-1}(X_0 \overset{\times}{\leftarrow} U_\Sigma)MX_{k-1}^\top(X_0 \overset{\times}{\leftarrow} U_\Sigma)C_k \frac{\mathrm{d}}{\mathrm{d}t}X_{k-1}(X_0 \overset{\times}{\leftarrow} U_\Sigma, A_i \overset{\pm}{\leftarrow} tR)\Big|_{t=0} \\
&= U_\Sigma \frac{\mathrm{d}}{\mathrm{d}t}X_{k-1}(A_i \overset{\pm}{\leftarrow} tU_\Sigma^{-1}RU_\Sigma)\Big|_{t=0} \\
&\quad + U_\Sigma A_k \frac{\mathrm{d}}{\mathrm{d}t}X_{k-1}(A_i \overset{\pm}{\leftarrow} tU_\Sigma^{-1}RU_\Sigma)\Big|_{t=0} M\left(X_{k-1}^\top C_k X_{k-1} + D_k\right) \\
&\quad + U_\Sigma A_k X_{k-1}M \frac{\mathrm{d}}{\mathrm{d}t}X_{k-1}^\top(A_i \overset{\pm}{\leftarrow} tU_\Sigma^{-1}RU_\Sigma)\Big|_{t=0} C_k X_{k-1} \\
&\quad + U_\Sigma A_k X_{k-1}MX_{k-1}^\top C_k \frac{\mathrm{d}}{\mathrm{d}t}X_{k-1}(A_i \overset{\pm}{\leftarrow} tU_\Sigma^{-1}RU_\Sigma)\Big|_{t=0} \\
&= U_\Sigma \frac{\mathrm{d}}{\mathrm{d}t}X_{k-1}(A_i \overset{\pm}{\leftarrow} tU_\Sigma^{-1}RU_\Sigma)\Big|_{t=0} + U_\Sigma \frac{\mathrm{d}}{\mathrm{d}t}A_k X_{k-1}(A_i \overset{\pm}{\leftarrow} tU_\Sigma^{-1}RU_\Sigma)M \\
&\quad \cdot \left(X_{k-1}^\top(A_i \overset{\pm}{\leftarrow} tU_\Sigma^{-1}RU_\Sigma)C_k X_{k-1}(A_i \overset{\pm}{\leftarrow} tU_\Sigma^{-1}RU_\Sigma) + D_k\right)\Big|_{t=0} \\
&= U_\Sigma \frac{\mathrm{d}}{\mathrm{d}t}X_k(A_i \overset{\pm}{\leftarrow} tU_\Sigma^{-1}RU_\Sigma)\Big|_{t=0}.
\end{aligned}$$

461  This completes the proof of eq. (17).

462  Under the condition that $B_l = b_l I_d$ for some $b_l \in \mathbb{R}$, we can simplify eq. (14) as

$$\begin{aligned}
Y_l &= Y_{l-1} + b_l Y_{l-1}M(X_{l-1}^\top C_l X_{l-1} + D_l) \\
&= Y_{l-1}\left(I + b_l M(X_{l-1}^\top C_l X_{l-1} + D_l)\right) \\
&= Y_0 \prod_{j=1}^{l}\left(I + b_j M(X_{j-1}^\top C_j X_{j-1} + D_j)\right).
\end{aligned}$$

463 Define $G_l = \overline{X}_0 \prod_{j=1}^{l} \left( I + b_j M(X_{j-1}^\top C_j X_{j-1} + D_j) \right)$, then it satisfies that $Y_l = W G_l$. We are
464 ready to prove that similar results to eqs. (16) and (17) also hold for $G_l, l \in [1, L]$:

$$G_l(X_0 \overset{\times}{\leftarrow} U_\Sigma) = U_\Sigma G_l, \tag{18}$$

$$\left.\frac{\mathrm{d}}{\mathrm{d}t} G_l(X_0 \overset{\times}{\leftarrow} U_\Sigma, A_i \overset{\pm}{\leftarrow} tR)\right|_{t=0} = \left.U_\Sigma \frac{\mathrm{d}}{\mathrm{d}t} G_l(A_i \overset{\pm}{\leftarrow} tU_\Sigma^{-1} R U_\Sigma)\right|_{t=0}. \tag{19}$$

465 Notice that eq. (18) holds trivially for $l = 0$ as $G_0 = \overline{X}_0$. Now suppose that eq. (18) holds for some
466 $l = k - 1$, we then have

$$G_k(X_0 \overset{\times}{\leftarrow} U_\Sigma) = G_{k-1}(X_0 \overset{\times}{\leftarrow} U_\Sigma)\left(I + b_k M(X_{k-1}^\top(X_0 \overset{\times}{\leftarrow} U_\Sigma)C_k X_{k-1}(X_0 \overset{\times}{\leftarrow} U_\Sigma) + D_k)\right)$$
$$= U_\Sigma G_{k-1}\left(I + b_k M(X_{k-1}^\top C_k X_{k-1} + D_k)\right) = U_\Sigma G_k.$$

467 This concludes eq. (18). As for eq. (19), notice that both sides equal 0 when $l \leq i$. Now suppose that
468 eq. (19) holds for some $l = k - 1 \geq i$, we then have:

$$\left.\frac{\mathrm{d}}{\mathrm{d}t} G_k(X_0 \overset{\times}{\leftarrow} U_\Sigma, A_i \overset{\pm}{\leftarrow} tR)\right|_{t=0}$$

$$= \left.\frac{\mathrm{d}}{\mathrm{d}t} G_{k-1}(X_0 \overset{\times}{\leftarrow} U_\Sigma, A_i \overset{\pm}{\leftarrow} tR)\right|_{t=0} + \frac{\mathrm{d}}{\mathrm{d}t} b_k G_{k-1}(X_0 \overset{\times}{\leftarrow} U_\Sigma, A_i \overset{\pm}{\leftarrow} tR)M$$

$$\cdot \left.\left(X_{k-1}^\top(X_0 \overset{\times}{\leftarrow} U_\Sigma, A_i \overset{\pm}{\leftarrow} tR)C_k X_{k-1}(X_0 \overset{\times}{\leftarrow} U_\Sigma, A_i \overset{\pm}{\leftarrow} tR) + D_k\right)\right|_{t=0}$$

$$= \left.\frac{\mathrm{d}}{\mathrm{d}t} G_{k-1}(X_0 \overset{\times}{\leftarrow} U_\Sigma, A_i \overset{\pm}{\leftarrow} tR)\right|_{t=0}$$

$$+ b_k \left.\frac{\mathrm{d}}{\mathrm{d}t} G_{k-1}(X_0 \overset{\times}{\leftarrow} U_\Sigma, A_i \overset{\pm}{\leftarrow} tR)\right|_{t=0} M\left(X_{k-1}^\top(X_0 \overset{\times}{\leftarrow} U_\Sigma)C_k X_{k-1}(X_0 \overset{\times}{\leftarrow} U_\Sigma) + D_k\right)$$

$$+ b_k G_{k-1}(X_0 \overset{\times}{\leftarrow} U_\Sigma)M \left.\frac{\mathrm{d}}{\mathrm{d}t} X_{k-1}^\top(X_0 \overset{\times}{\leftarrow} U_\Sigma, A_i \overset{\pm}{\leftarrow} tR)\right|_{t=0} C_k X_{k-1}(X_0 \overset{\times}{\leftarrow} U_\Sigma)$$

$$+ b_k G_{k-1}(X_0 \overset{\times}{\leftarrow} U_\Sigma)M X_{k-1}^\top(X_0 \overset{\times}{\leftarrow} U_\Sigma)C_k \left.\frac{\mathrm{d}}{\mathrm{d}t} X_{k-1}(X_0 \overset{\times}{\leftarrow} U_\Sigma, A_i \overset{\pm}{\leftarrow} tR)\right|_{t=0}$$

$$= \left.U_\Sigma \frac{\mathrm{d}}{\mathrm{d}t} G_{k-1}(A_i \overset{\pm}{\leftarrow} tU_\Sigma^{-1} R U_\Sigma)\right|_{t=0}$$

$$+ b_k U_\Sigma \left.\frac{\mathrm{d}}{\mathrm{d}t} G_{k-1}(A_i \overset{\pm}{\leftarrow} tU_\Sigma^{-1} R U_\Sigma)\right|_{t=0} M\left(X_{k-1}^\top C_k X_{k-1} + D_k\right)$$

$$+ b_k U_\Sigma G_{k-1} M \left.\frac{\mathrm{d}}{\mathrm{d}t} X_{k-1}^\top(A_i \overset{\pm}{\leftarrow} tU_\Sigma^{-1} R U_\Sigma)\right|_{t=0} C_k X_{k-1}$$

$$+ b_k U_\Sigma G_{k-1} M X_{k-1}^\top C_k \left.\frac{\mathrm{d}}{\mathrm{d}t} X_{k-1}(A_i \overset{\pm}{\leftarrow} tU_\Sigma^{-1} R U_\Sigma)\right|_{t=0}$$

$$= \left.U_\Sigma \frac{\mathrm{d}}{\mathrm{d}t} G_k(A_i \overset{\pm}{\leftarrow} tU_\Sigma^{-1} R U_\Sigma)\right|_{t=0}.$$

469 This concludes the proof of eq. (19). Consider the in-context risk:

$$\left.\frac{\mathrm{d}}{\mathrm{d}t} \mathcal{L}(A_i \overset{\pm}{\leftarrow} tR)\right|_{t=0}$$

$$= 2\mathbb{E}_{X_0, W, U_\perp}\left[\text{tr}\left(\left.(I - M)Y_L^\top(X_0 \overset{\times}{\leftarrow} U_\Sigma)\frac{\mathrm{d}}{\mathrm{d}t} Y_L(X_0 \overset{\times}{\leftarrow} U_\Sigma, A_i \overset{\pm}{\leftarrow} tR)\right|_{t=0} (I - M)\right)\right]$$

$$= 2\mathbb{E}_{X_0, W, U_\perp}\left[\text{tr}\left(\left.(I - M)G_L^\top U_\Sigma^\top W^\top W U_\Sigma \frac{\mathrm{d}}{\mathrm{d}t} G_L(A_i \overset{\pm}{\leftarrow} tU_\Sigma^{-1} R U_\Sigma)\right|_{t=0} (I - M)\right)\right]$$

$$= 2d\,\mathbb{E}_{X_0}\left[\text{tr}\left((I - M)G_L^\top \Sigma^{-1} \left.\frac{\mathrm{d}}{\mathrm{d}t} \mathbb{E}_{U_\perp}\left[G_L(A_i \overset{\pm}{\leftarrow} tU_\Sigma^{-1} R U_\Sigma)\right]\right|_{t=0} (I - M)\right)\right]$$

$$= 2d\,\mathbb{E}_{X_0}\left[\mathrm{tr}\left((I-M)G_L^\top\Sigma^{-1}\left.\frac{\mathrm{d}}{\mathrm{d}t}G_L(A_i \overset{+}{\underset{-}{\leftarrow}} \mathbb{E}_{U_\perp}[tU_\Sigma^{-1}RU_\Sigma])\right|_{t=0}(I-M)\right)\right]$$

$$= 2d\,\mathbb{E}_{X_0}\left[\mathrm{tr}\left((I-M)G_L^\top\Sigma^{-1}\left.\frac{\mathrm{d}}{\mathrm{d}t}G_L(A_i \overset{+}{\underset{-}{\leftarrow}} trI_d)\right|_{t=0}(I-M)\right)\right]$$

$$= \left.\frac{\mathrm{d}}{\mathrm{d}t}\mathbb{E}_{X_0,W}\left[\mathrm{tr}\left((I-M)Y_L^\top(A_i \overset{+}{\underset{-}{\leftarrow}} trI_d)Y_L(A_i \overset{+}{\underset{-}{\leftarrow}} trI_d)(I-M)\right)\right]\right|_{t=0}$$

$$= \left.\frac{\mathrm{d}}{\mathrm{d}t}\mathcal{L}(A_i \overset{+}{\underset{-}{\leftarrow}} trI_d)\right|_{t=0},$$

470 where $r = \mathbb{E}_{U_\perp}[U_\Sigma^{-1}RU_\Sigma] = \frac{1}{d}\mathrm{tr}(\Sigma^{-1/2}R\Sigma^{1/2})$, and we used the fact that $U_\Sigma^\top\Sigma^{-1}U_\Sigma = \Sigma^{-1}$,

471 and $\left.\frac{\mathrm{d}}{\mathrm{d}t}G_L(A_i \overset{+}{\underset{-}{\leftarrow}} tR)\right|_{t=0}$ is affine in $R$. This concludes that eq. (15) holds for $A_i$, $i \in [1, L]$.

472 **2. Equation (15) holds for $B_i$.**

473 From the recursive expressions in eq. (14), we can conclude that the values of $X_l$ do not depend on
474 $B_i$. Therefore, we naturally have

$$X_l(B_i \overset{+}{\underset{-}{\leftarrow}} tR) = X_l. \tag{20}$$

475 Next, we would like to show that for any $l \in [1, L]$,

$$\mathbb{E}_W\left[W^\top\left.\frac{\mathrm{d}}{\mathrm{d}t}Y_l(B_i \overset{+}{\underset{-}{\leftarrow}} tR)\right|_{t=0}\right] = \Sigma^{-1}\left.\frac{\mathrm{d}}{\mathrm{d}t}G_l(b_i \overset{+}{\underset{-}{\leftarrow}} t\,\mathrm{tr}(R))\right|_{t=0}. \tag{21}$$

476 When $l < i$, we can easily verify eq. (21) since both sides equal 0. When $l = i$, we can get

$$\mathbb{E}_W\left[W^\top\left.\frac{\mathrm{d}}{\mathrm{d}t}Y_l(B_i \overset{+}{\underset{-}{\leftarrow}} tR)\right|_{t=0}\right] = \mathbb{E}_W\left[W^\top RY_{l-1}M\left(X_{l-1}^\top C_l X_{l-1} + D_l\right)\right]$$

$$= \mathbb{E}_W\left[W^\top RW\right]G_{l-1}M\left(X_{l-1}^\top C_l X_{l-1} + D_l\right)$$

$$= \mathrm{tr}(R)\Sigma^{-1}G_{l-1}M\left(X_{l-1}^\top C_l X_{l-1} + D_l\right)$$

$$= \Sigma^{-1}\left.\frac{\mathrm{d}}{\mathrm{d}t}G_l(b_i \overset{+}{\underset{-}{\leftarrow}} t\,\mathrm{tr}(R))\right|_{t=0}.$$

477 Suppose that eq. (21) holds for some $l = k - 1 \geq i$. One can then verify

$$\mathbb{E}_W\left[W^\top\left.\frac{\mathrm{d}}{\mathrm{d}t}Y_k(B_i \overset{+}{\underset{-}{\leftarrow}} tR)\right|_{t=0}\right]$$

$$= \mathbb{E}_W\left[W^\top\left.\frac{\mathrm{d}}{\mathrm{d}t}Y_{k-1}(B_i \overset{+}{\underset{-}{\leftarrow}} tR)\left(I + b_k M(X_{k-1}^\top C_k X_{k-1} + D_k)\right)\right|_{t=0}\right]$$

$$= \mathbb{E}_W\left[W^\top\left.\frac{\mathrm{d}}{\mathrm{d}t}Y_{k-1}(B_i \overset{+}{\underset{-}{\leftarrow}} tR)\right|_{t=0}\right]\left(I + b_k M(X_{k-1}^\top C_k X_{k-1} + D_k)\right)$$

$$= \Sigma^{-1}\left.\frac{\mathrm{d}}{\mathrm{d}t}G_{k-1}(b_i \overset{+}{\underset{-}{\leftarrow}} t\,\mathrm{tr}(R))\right|_{t=0}\left(I + b_k M(X_{k-1}^\top C_k X_{k-1} + D_k)\right)$$

$$= \Sigma^{-1}\left.\frac{\mathrm{d}}{\mathrm{d}t}G_k(b_i \overset{+}{\underset{-}{\leftarrow}} t\,\mathrm{tr}(R))\right|_{t=0}.$$

478 The proof of eq. (21) is complete. Now, look at the in-context risk, we have

$$\left.\frac{\mathrm{d}}{\mathrm{d}t}\mathcal{L}(B_i \overset{+}{\underset{-}{\leftarrow}} tR)\right|_{t=0} = 2\,\mathbb{E}_{X_0,W}\left[\mathrm{tr}\left((I-M)Y_L^\top\left.\frac{\mathrm{d}}{\mathrm{d}t}Y_L(B_i \overset{+}{\underset{-}{\leftarrow}} tR)\right|_{t=0}(I-M)\right)\right]$$

$$= 2\,\mathbb{E}_{X_0}\left[\mathrm{tr}\left((I-M)G_L^\top\,\mathbb{E}_W\left[W^\top\left.\frac{\mathrm{d}}{\mathrm{d}t}Y_L(B_i \overset{+}{\underset{-}{\leftarrow}} tR)\right|_{t=0}\right](I-M)\right)\right]$$

$$= 2\,\mathbb{E}_{X_0}\left[\mathrm{tr}\left((I-M)G_L^\top\Sigma^{-1}\left.\frac{\mathrm{d}}{\mathrm{d}t}G_L(b_i \overset{+}{\underset{-}{\leftarrow}} t\,\mathrm{tr}(R))\right|_{t=0}(I-M)\right)\right]$$

$$= 2\,\mathbb{E}_{X_0,W}\left[\mathrm{tr}\left((I-M)Y_L^\top\left.\frac{\mathrm{d}}{\mathrm{d}t}Y_L(B_i \overset{+}{\underset{-}{\leftarrow}} t\,\mathrm{tr}(R)I_d)\right|_{t=0}(I-M)\right)\right]$$

$$= \frac{\mathrm{d}}{\mathrm{d}t}\mathcal{L}(B_i \overset{+}{\leftarrow} t\operatorname{tr}(R)I_d)\Big|_{t=0}.$$

This concludes that eq. (15) holds for $B_i$, $i \in [1, L]$.

**3. Equation (15) holds for $C_i$.**

Similar to the $A_i$ case, we will first prove that for any $l \in [1, L]$,

$$\frac{\mathrm{d}}{\mathrm{d}t}X_l(X_0 \overset{\times}{\leftarrow} U_\Sigma, C_i \overset{+}{\leftarrow} tR)\Big|_{t=0} = U_\Sigma \frac{\mathrm{d}}{\mathrm{d}t}X_l(C_i \overset{+}{\leftarrow} tU_\Sigma^\top RU_\Sigma)\Big|_{t=0}. \tag{22}$$

The equation above holds trivially for $l < i$. For the case $l = i$, we have

$$\frac{\mathrm{d}}{\mathrm{d}t}X_l(X_0 \overset{\times}{\leftarrow} U_\Sigma, C_i \overset{+}{\leftarrow} tR)\Big|_{t=0}$$

$$= A_j X_{l-1}(X_0 \overset{\times}{\leftarrow} U_\Sigma)MX_{l-1}^\top(X_0 \overset{\times}{\leftarrow} U_\Sigma)RX_{l-1}(X_0 \overset{\times}{\leftarrow} U_\Sigma)$$

$$= U_\Sigma A_j X_{l-1}MX_{l-1}^\top U_\Sigma^\top RU_\Sigma X_{l-1} = U_\Sigma \frac{\mathrm{d}}{\mathrm{d}t}X_l(C_i \overset{+}{\leftarrow} tU_\Sigma^\top RU_\Sigma)\Big|_{t=0}.$$

One can conclude the proof of eq. (22) through a similar reduction as eq. (17) for $l > i$ layers. Next, we establish the corresponding result for $G_l$:

$$\frac{\mathrm{d}}{\mathrm{d}t}G_l(X_0 \overset{\times}{\leftarrow} U_\Sigma, C_i \overset{+}{\leftarrow} tR)\Big|_{t=0} = U_\Sigma \frac{\mathrm{d}}{\mathrm{d}t}G_l(C_i \overset{+}{\leftarrow} tU_\Sigma^\top RU_\Sigma)\Big|_{t=0}. \tag{23}$$

This equation holds trivially for $l < i$. When taking $l = i$, we can verify that

$$\frac{\mathrm{d}}{\mathrm{d}t}G_l(X_0 \overset{\times}{\leftarrow} U_\Sigma, C_i \overset{+}{\leftarrow} tR)\Big|_{t=0} = b_l G_{l-1}(X_0 \overset{\times}{\leftarrow} U_\Sigma)MX_{l-1}^\top(X_0 \overset{\times}{\leftarrow} U_\Sigma)RX_{l-1}(X_0 \overset{\times}{\leftarrow} U_\Sigma)$$

$$= b_l U_\Sigma G_{l-1}(X_0 \overset{\times}{\leftarrow} U_\Sigma)MX_{l-1}^\top U_\Sigma^\top RU_\Sigma X_{l-1}$$

$$= U_\Sigma \frac{\mathrm{d}}{\mathrm{d}t}G_l(C_i \overset{+}{\leftarrow} tU_\Sigma^\top RU_\Sigma)\Big|_{t=0}.$$

For $l > i$ layers, one can follow similar reductions as eq. (19) to finish the proof. We then consider the in-context risk:

$$\frac{\mathrm{d}}{\mathrm{d}t}\mathcal{L}(C_i \overset{+}{\leftarrow} tR)\Big|_{t=0}$$

$$= 2\,\mathbb{E}_{X_0,W,U_\perp}\left[\operatorname{tr}\left((I-M)Y_L^\top(X_0 \overset{\times}{\leftarrow} U_\Sigma)\frac{\mathrm{d}}{\mathrm{d}t}Y_L(X_0 \overset{\times}{\leftarrow} U_\Sigma, C_i \overset{+}{\leftarrow} tR)\Big|_{t=0}(I-M)\right)\right]$$

$$= 2\,\mathbb{E}_{X_0,W,U_\perp}\left[\operatorname{tr}\left((I-M)G_L^\top U_\Sigma^\top W^\top WU_\Sigma \frac{\mathrm{d}}{\mathrm{d}t}G_L(C_i \overset{+}{\leftarrow} tR)\Big|_{t=0}(I-M)\right)\right]$$

$$= 2d\,\mathbb{E}_{X_0}\left[\operatorname{tr}\left((I-M)G_L^\top\Sigma^{-1}\frac{\mathrm{d}}{\mathrm{d}t}\mathbb{E}_{U_\perp}\left[G_L(C_i \overset{+}{\leftarrow} tU_\Sigma^\top RU_\Sigma)\right]\Big|_{t=0}(I-M)\right)\right]$$

$$= 2d\,\mathbb{E}_{X_0}\left[\operatorname{tr}\left((I-M)G_L^\top\Sigma^{-1}\frac{\mathrm{d}}{\mathrm{d}t}G_L(C_i \overset{+}{\leftarrow} tr\Sigma^{-1})\Big|_{t=0}(I-M)\right)\right]$$

$$= \frac{\mathrm{d}}{\mathrm{d}t}\mathbb{E}_{X_0,W}\left[\operatorname{tr}\left((I-M)Y_L^\top(C_i \overset{+}{\leftarrow} tr\Sigma^{-1})Y_L(C_i \overset{+}{\leftarrow} tr\Sigma^{-1})(I-M)\right)\right]\Big|_{t=0}$$

$$= \frac{\mathrm{d}}{\mathrm{d}t}\mathcal{L}(C_i \overset{+}{\leftarrow} tr\Sigma^{-1})\Big|_{t=0},$$

where $r = \mathbb{E}_{U_\perp}[U_\Sigma^\top RU_\Sigma] = \frac{1}{d}\operatorname{tr}(\Sigma^{1/2}R\Sigma^{1/2})$. This concludes that eq. (15) holds for $C_i$.

**4. Equation (15) holds for $D_i$.**

Let $U_p \in \mathbb{R}^{n \times n}$ be a uniformly sampled permutation matrix, i.e., a binary matrix that has exactly one 1 entry in each row and column with all other entries 0. Let $U_\circ = \operatorname{diag}(U_p \otimes I_2, I_2) \in \mathbb{R}^{(2n+2) \times (2n+2)}$.

One can verify that by multiplying $X_0 U_\circ$, it is equal to shuffling the first $n$ 2-column sub-blocks of $X_0$ and keeping the last 2 columns unchanged.

Then, consider a matrix $U_\xi = \mathrm{diag}(\xi_1, \ldots, \xi_{n+1}) \in \mathbb{R}^{(n+1) \times (n+1)}$ where $\xi_i \overset{\text{i.i.d.}}{\sim} \mathrm{Unif}\{\pm 1\}$, i.e., a diagonal matrix with random $\pm 1$ entries. Let $U_\pm = U_\xi \otimes I_2 \in \mathbb{R}^{(2n+2) \times (2n+2)}$. Thus, $U_\pm = U_\pm^\top$ and $X_0 U_\pm$ is randomly flipping the sign of each 2-column sub-block in $X_0$.

We are going to prove that for any $l \in [1, L]$, recalling that $f(A \overset{\diamond}{\leftarrow} B) = f(A \leftarrow AB)$,

$$X_l(X_0 \overset{\diamond}{\leftarrow} U_\pm U_\circ) = X_l U_\pm U_\circ, \tag{24}$$

$$G_l(X_0 \overset{\diamond}{\leftarrow} U_\pm U_\circ) = G_l U_\pm U_\circ. \tag{25}$$

Equation (24) holds trivially for $l = 0$. When eq. (24) holds for some $l = k - 1$, we can verify that

$$
\begin{aligned}
&X_k(X_0 \overset{\diamond}{\leftarrow} U_\pm U_\circ) \\
&= X_{k-1} U_\pm U_\circ + A_k X_{k-1} U_\pm U_\circ M \big( U_\circ^\top U_\pm^\top X_{k-1}^\top C_k X_{k-1} U_\pm U_\circ + D_k \big) \\
&= X_{k-1} U_\pm U_\circ + A_k X_{k-1} U_\pm U_\circ M U_\circ^\top U_\pm^\top \big( X_{k-1}^\top C_k X_{k-1} + U_\pm U_\circ D_k U_\circ^\top U_\pm^\top \big) U_\pm U_\circ \\
&= X_{k-1} U_\pm U_\circ + A_k X_{k-1} M \big( X_{k-1}^\top C_k X_{k-1} + D_k \big) U_\pm U_\circ \\
&= \big( X_{k-1} + A_k X_{k-1} M \big( X_{k-1}^\top C_k X_{k-1} + D_k \big) \big) U_\pm U_\circ = X_k U_\pm U_\circ.
\end{aligned}
$$

It uses the fact that there exists some $D_i^1, D_i^2 \in \mathbb{R}^{2 \times 2}$ such that $D_i = \mathrm{diag}(I_n \otimes D_i^1, D_i^2)$, so shuffling the first $n$ $2 \times 2$ diagonal sub-blocks of $D_i$ does not change the matrix, and we have $U_\circ D_i U_\circ^\top = D_i$. Similarly, we have $U_\pm D_k U_\pm^\top = D_k$. This concludes eq. (24), and eq. (25) could be acquired similarly.

Next, we will establish the following equalities for $X_l$ and $G_l$:

$$\left. \frac{\mathrm{d}}{\mathrm{d}t} X_l(X_0 \overset{\diamond}{\leftarrow} U_\pm U_\circ, D_i \overset{+}{\leftarrow} tR) \right|_{t=0} = \left. \frac{\mathrm{d}}{\mathrm{d}t} X_l(D_i \overset{+}{\leftarrow} t U_\pm U_\circ R U_\circ^\top U_\pm^\top) \right|_{t=0} U_\pm U_\circ, \tag{26}$$

$$\left. \frac{\mathrm{d}}{\mathrm{d}t} G_l(X_0 \overset{\diamond}{\leftarrow} U_\pm U_\circ, D_i \overset{+}{\leftarrow} tR) \right|_{t=0} = \left. \frac{\mathrm{d}}{\mathrm{d}t} G_l(D_i \overset{+}{\leftarrow} t U_\pm U_\circ R U_\circ^\top U_\pm^\top) \right|_{t=0} U_\pm U_\circ. \tag{27}$$

The proof follows by similar reductions as proving eqs. (17) and (19).

Finally, we consider the in-context risk under the permutation of $U_p$ and $U_\xi$. Since each pair of $(x_i, y_i)$ is equivalently sampled from Gaussian distributions, we have $X_0 \overset{d}{=} X_0 U_\pm U_\circ$. Therefore,

$$
\begin{aligned}
&\left. \frac{\mathrm{d}}{\mathrm{d}t} \mathcal{L}(D_i \overset{+}{\leftarrow} tR) \right|_{t=0} \\
&= 2 \, \mathbb{E}_{X_0, W} \left[ \mathrm{tr} \left( (I - M) Y_L^\top \left. \frac{\mathrm{d}}{\mathrm{d}t} Y_L(D_i \overset{+}{\leftarrow} tR) \right|_{t=0} (I - M) \right) \right] \\
&= 2 \, \mathbb{E}_{X_0, W, U_p, U_\xi} \left[ \mathrm{tr} \left( (I - M) Y_L^\top (X_0 \overset{\diamond}{\leftarrow} U_\pm U_\circ) \left. \frac{\mathrm{d}}{\mathrm{d}t} Y_L(X_0 \overset{\diamond}{\leftarrow} U_\pm U_\circ, D_i \overset{+}{\leftarrow} tR) \right|_{t=0} (I - M) \right) \right] \\
&= 2d \, \mathbb{E}_{X_0, U_p, U_\xi} \left[ \mathrm{tr} \left( (I - M) U_\circ^\top U_\pm^\top G_L^\top \Sigma^{-1} \left. \frac{\mathrm{d}}{\mathrm{d}t} G_L(D_i \overset{+}{\leftarrow} t U_\pm U_\circ R U_\circ^\top U_\pm^\top) \right|_{t=0} U_\pm U_\circ (I - M) \right) \right] \\
&= 2d \, \mathbb{E}_{X_0} \left[ \mathrm{tr} \left( (I - M) G_L^\top \Sigma^{-1} \left. \frac{\mathrm{d}}{\mathrm{d}t} \mathbb{E}_{U_p, U_\xi} \left[ G_L(D_i \overset{+}{\leftarrow} t U_\pm U_\circ^\top R U_\circ U_\pm) \right] \right|_{t=0} (I - M) \right) \right] \\
&= 2d \, \mathbb{E}_{X_0} \left[ \mathrm{tr} \left( (I - M) G_L^\top \Sigma^{-1} \left. \frac{\mathrm{d}}{\mathrm{d}t} G_L(D_i \overset{+}{\leftarrow} t\widetilde{R}) \right|_{t=0} (I - M) \right) \right] = \left. \frac{\mathrm{d}}{\mathrm{d}t} \mathcal{L}(D_i \overset{+}{\leftarrow} t\widetilde{R}) \right|_{t=0},
\end{aligned}
$$

where $\widetilde{R} = \mathbb{E}_{U_p, U_\xi}[U_\pm U_\circ^\top R U_\circ U_\pm] = \mathrm{diag}(I_n \otimes R^1, R^2)$, $R^1 = \frac{1}{n} \sum_{j=1}^n R_j$, $R^2 = R_{n+1}$, and $R_j$ is the $j$-th $2 \times 2$ diagonal block of $R$. The 4th equality uses the fact that $\mathrm{tr}[(I - M)A(I - M)]$ is extracting the right-bottom element of $A$, so it should be equal to $\mathrm{tr}\big[(I - M) U_\circ^\top U_\pm^\top A U_\pm U_\circ (I - M)\big]$ for any matrix $A$. This concludes that eq. (15) holds for $D_i$.

Till now, we have proved that eq. (15) holds for each one of $A_i, B_i, C_i, D_i$. The proof of the whole theorem is then completed by applying Lemma 8. $\qquad \square$

## B.3 Proof of Theorem 2

*Proof.* In this proof, we follow the same notations as the proof of Theorem 1, where the constant $\frac{1}{n}$ factor is dropped and $\widetilde{Z}_0, \widetilde{X}_0, \widetilde{Y}_0$ are simplified as $Z_0, X_0, Y_0$ respectively.

$$Z_0 = \begin{bmatrix} x_1 & 0 & 0 & \cdots & x_n & 0 & 0 & x_{\text{test}} & 0 & 0 \\ 0 & 0 & y_1 & \cdots & 0 & 0 & y_n & 0 & 0 & y_{\text{test}} \end{bmatrix} \in \mathbb{R}^{(2d)\times(3n+3)}. \tag{28}$$

Let $Z_l \in \mathbb{R}^{2d\times(3n+3)}$ be the $l$-th layer's output and let $X_l, Y_l \in \mathbb{R}^{d\times(3n+3)}$ be its first and last $d$ rows. Our goal is to prove that, for any $E \in A \cup B \cup C \cup D$ and an arbitrary matrix $R \in \mathbb{R}^{d\times d}$ ($\mathbb{R}^{d_p \times d_p}$ for $D$), there exists $\widetilde{R} \in \mathcal{S}_I$ ($\mathcal{S}_\Sigma$ for C, $\mathcal{S}_P$ for D) such that

$$\frac{\mathrm{d}}{\mathrm{d}t} \mathcal{L}(E \overset{+}{\leftarrow} t\widetilde{R})\Big|_{t=0} \leq \frac{\mathrm{d}}{\mathrm{d}t} \mathcal{L}(E \overset{+}{\leftarrow} tR)\Big|_{t=0}. \tag{29}$$

The proofs of eq. (29) for $A_i$, $B_i$ and $C_i$ are identical with the proof of Theorem 1 so we omit them. We will be focusing on $D_i$ for the rest of the proof.

Let $U_p^s \in \mathbb{R}^{n\times n}$ and $U_p^t \in \mathbb{R}^{(n+1)\times(n+1)}$ be uniformly sampled permutation matrices. Let $U_\circ^s = \mathrm{diag}(U_p^s, 1) \otimes \mathrm{diag}(1, 0, 1)$ and $U_\circ^t = U_p^t \otimes \mathrm{diag}(0, 1, 0)$. Therefore, $X_0 U_\circ^s$ is shuffling the 1-st and 3-rd columns among each 3-column sub-block of $X_0$ (except for the last 3-column sub-block), and $X_0 U_\circ^s$ is shuffling the 2-nd column among each 3-column sub-block. Next, let $U_\xi^s, U_\xi^t \in \mathbb{R}^{(n+1)\times(n+1)}$ be diagonal matrices with uniformly sampled $\pm 1$ entries. Define $U_\pm^s = U_\xi^s \otimes \mathrm{diag}(1, 0, 1)$ and $U_\pm^t = U_\xi^t \otimes \mathrm{diag}(0, 1, 0)$. It can then be verified that $X_0 U_\pm^s U_\pm^t \overset{d}{=} X_0$.

To simplify the notations, let $U_\equiv$ denote $U_\pm^s U_\pm^t U_\circ^s U_\circ^t$. We will focus on a subset of $\mathcal{S}_P$:

$$\mathcal{S}_P' = \Big\{ \mathrm{diag}(I_n \otimes \Lambda_1, \Lambda_2) + I_{n+1} \otimes \Lambda_3 \,\Big|\, \Lambda_1, \Lambda_2 \in \mathcal{M}\big(\begin{smallmatrix} 1 & 0 & 1 \\ 0 & 0 & 0 \\ 1 & 0 & 1 \end{smallmatrix}\big), \Lambda_3 \in \mathcal{M}\big(\begin{smallmatrix} 0 & 0 & 0 \\ 0 & 1 & 0 \\ 0 & 0 & 0 \end{smallmatrix}\big) \Big\}.$$

Assume $D_k = \mathrm{diag}(I_n \otimes \Lambda_1, \Lambda_2) + I_{n+1} \otimes \Lambda_3 \in \mathcal{S}_P'$ as defined above, one can verify that it is a block-diagonal matrix constructed from the same $3 \times 3$ sub-blocks, and thus is invariant under $U_\equiv D_k U_\equiv^\top$. We will then prove that for any $l \in [1, L]$,

$$X_l(X_0 \overset{\diamond}{\leftarrow} U_\equiv) = X_l U_\equiv, \tag{30}$$

$$G_l(X_0 \overset{\diamond}{\leftarrow} U_\equiv) = G_l U_\equiv, \tag{31}$$

$$\frac{\mathrm{d}}{\mathrm{d}t} X_l(X_0 \overset{\diamond}{\leftarrow} U_\equiv, D_i \overset{+}{\leftarrow} tR)\Big|_{t=0} = \frac{\mathrm{d}}{\mathrm{d}t} X_l(D_i \overset{+}{\leftarrow} tU_\equiv R U_\equiv^\top)\Big|_{t=0} U_\equiv, \tag{32}$$

$$\frac{\mathrm{d}}{\mathrm{d}t} G_l(X_0 \overset{\diamond}{\leftarrow} U_\equiv, D_i \overset{+}{\leftarrow} tR)\Big|_{t=0} = \frac{\mathrm{d}}{\mathrm{d}t} G_l(D_i \overset{+}{\leftarrow} tU_\equiv R U_\equiv^\top)\Big|_{t=0} U_\equiv. \tag{33}$$

These results can be acquired by similar proofs as eqs. (24) to (27). We then consider the in-context risk under the permutations of $U_\equiv$. Similarly, we have $X_0 \overset{d}{=} X_0 U_\equiv$ and

$$\frac{\mathrm{d}}{\mathrm{d}t} \mathcal{L}(D_i \overset{+}{\leftarrow} tR)\Big|_{t=0}$$

$$= 2\,\mathbb{E}_{X_0, W}\Big[\mathrm{tr}\Big( (I - M) Y_L^\top \frac{\mathrm{d}}{\mathrm{d}t} Y_L(D_i \overset{+}{\leftarrow} tR)\Big|_{t=0} (I - M) \Big)\Big]$$

$$= 2d\,\mathbb{E}_{X_0, U_\equiv}\Big[\mathrm{tr}\Big( (I - M) G_L^\top(X_0 \overset{\diamond}{\leftarrow} U_\equiv) \Sigma^{-1} \frac{\mathrm{d}}{\mathrm{d}t} G_L(X_0 \overset{\diamond}{\leftarrow} U_\equiv, D_i \overset{+}{\leftarrow} tR)\Big|_{t=0} (I - M) \Big)\Big]$$

$$= 2d\,\mathbb{E}_{X_0, U_\equiv}\Big[\mathrm{tr}\Big( (I - M) U_\equiv^\top G_L^\top \Sigma^{-1} \frac{\mathrm{d}}{\mathrm{d}t} G_L(D_i \overset{+}{\leftarrow} tU_\equiv R U_\equiv^\top)\Big|_{t=0} U_\equiv (I - M) \Big)\Big]$$

$$= 2d\,\mathbb{E}_{X_0}\Big[\mathrm{tr}\Big( (I - M) G_L^\top \Sigma^{-1} \frac{\mathrm{d}}{\mathrm{d}t} G_L(D_i \overset{+}{\leftarrow} t\,\mathbb{E}_{U_\equiv}[U_\equiv R U_\equiv^\top])\Big|_{t=0} (I - M) \Big)\Big]$$

$$= \frac{\mathrm{d}}{\mathrm{d}t} \mathcal{L}(D_i \overset{+}{\leftarrow} t\widetilde{R})\Big|_{t=0}.$$

533 Let $R_j$ be the $j$-th $3 \times 3$ diagonal block of $R$, then $R^1 = \frac{1}{n} \sum_{j=1}^{n} R_j \circ \left( \begin{smallmatrix} 1 & 0 & 1 \\ 0 & 0 & 0 \\ 1 & 0 & 1 \end{smallmatrix} \right)$, $R^2 = R_{n+1} \circ \left( \begin{smallmatrix} 1 & 0 & 1 \\ 0 & 0 & 0 \\ 1 & 0 & 1 \end{smallmatrix} \right)$,

534 $R^3 = \frac{1}{n+1} \sum_{j=1}^{n+1} R_j \circ \left( \begin{smallmatrix} 0 & 0 & 0 \\ 0 & 1 & 0 \\ 0 & 0 & 0 \end{smallmatrix} \right)$ and $\widetilde{R} = \mathbb{E}_{U_{\equiv}}[U_{\equiv} R U_{\equiv}^\top] = \mathrm{diag}(I_n \otimes R^1, R^2) + I_{n+1} \otimes R^3$. This

535 indicates that eq. (29) holds for each $D_i \in \mathcal{S}'_P$, and thus the proof of the whole theorem completes by

536 applying Lemma 8 and noticing that $\mathcal{S}'_P \subset \mathcal{S}_P$. $\qquad\square$

## B.4 Proof of Theorem 5

538 *Proof.* We keep the same notations as the proof of Theorem 1, dropping the $\frac{1}{n}$ factor and simplifying

539 $\widetilde{X}_0, \widetilde{Y}_0, \widetilde{Z}_0$ as $X_0, Y_0, Z_0$, as follows:

$$Z_0 = \begin{bmatrix} 0 & 0 & \cdots & 0 & 0 & 0 & 0 \\ x_1 & y_1 & \cdots & x_n & y_n & x_{\text{test}} & y_{\text{test}} \end{bmatrix} \in \mathbb{R}^{(2d) \times (2n+2)}. \tag{34}$$

540 Note that we now have $X_0$ and $Y_0$ containing both $x_i$ and $y_i$. Define

$$X = \begin{bmatrix} x_1 & 0 & \cdots & x_n & 0 & x_{\text{test}} & 0 \end{bmatrix},$$
$$\overline{X} = \begin{bmatrix} 0 & x_1 & \cdots & 0 & x_n & 0 & x_{\text{test}} \end{bmatrix},$$
$$Y = \begin{bmatrix} 0 & y_1 & \cdots & 0 & y_n & 0 & y_{\text{test}} \end{bmatrix}.$$

541 we then have $Y_0 = X + Y = X + W\overline{X}$. From the parameter configuration in eq. (12), the update

542 rule of the first attention layer is

$$X_1 = A_1 Y_0 M D_1 = A_1 X M D_1, \quad Y_1 = Y_0 = X + W\overline{X}. \tag{35}$$

543 The update rule for the following layers is the same as eq. (14). We are going to prove that, for any

544 $E \in A \cup B \cup C \cup D$ and an arbitrary matrix $R \in \mathbb{R}^{d \times d}$ ($\mathbb{R}^{d_p \times d_p}$ for $D$), there exists $\widetilde{R} \in \mathcal{S}_I$ ($\mathcal{S}_\Sigma$

545 for $C$, $\mathcal{S}_P$ for $D$) such that

$$\left. \frac{\mathrm{d}}{\mathrm{d}t} \mathcal{L}(E \xleftarrow{\pm} t\widetilde{R}) \right|_{t=0} \leq \left. \frac{\mathrm{d}}{\mathrm{d}t} \mathcal{L}(E \xleftarrow{\pm} tR) \right|_{t=0}. \tag{36}$$

546 Similarly to Theorem 1, we uniformly sample $U_\perp \in \mathbb{R}^{d \times d}$ as an orthonormal random matrix, and let

547 $U_\Sigma = \Sigma^{1/2} U_\perp \Sigma^{-1/2}$. Under the condition that $B_l = b_l I_d$ for some $b_l \in \mathbb{R}$, we have

$$Y_l = Y_1 \prod_{j=2}^{l} \left( I + b_j M \left( X_{j-1}^\top C_j X_{j-1} + D_j \right) \right).$$

548 Let $F_l = X \prod_{j=2}^{l} \left( I + b_j M \left( X_{j-1}^\top C_j X_{j-1} + D_j \right) \right)$, $G_l = \overline{X} \prod_{j=2}^{l} \left( I + b_j M \left( X_{j-1}^\top C_j X_{j-1} + D_j \right) \right)$,

549 we then have $Y_l = F_l + W G_l$. According to Lemma 9,

$$\left. \frac{\mathrm{d}}{\mathrm{d}t} \mathcal{L}(E \xleftarrow{\pm} tR) \right|_{t=0}$$
$$= \left. \frac{\mathrm{d}}{\mathrm{d}t} \mathbb{E}_{X_0, W} \left[ \mathrm{tr} \left( (I - M) Y_L^\top (E \xleftarrow{\pm} tR) Y_L (E \xleftarrow{\pm} tR)(I - M) \right) \right] \right|_{t=0}$$
$$= \left. \frac{\mathrm{d}}{\mathrm{d}t} \mathbb{E}_{X_0, W} \left[ \mathrm{tr} \left( (I - M) F_L^\top (E \xleftarrow{\pm} tR) F_L (E \xleftarrow{\pm} tR)(I - M) \right) \right] \right|_{t=0}$$
$$\quad + \left. \frac{\mathrm{d}}{\mathrm{d}t} \mathbb{E}_{X_0, W} \left[ \mathrm{tr} \left( (I - M) G_L^\top (E \xleftarrow{\pm} tR) W^\top W G_L (E \xleftarrow{\pm} tR)(I - M) \right) \right] \right|_{t=0}$$
$$= 2 \mathbb{E}_{X_0} \left[ \mathrm{tr} \left( (I - M) F_L^\top \left. \frac{\mathrm{d}}{\mathrm{d}t} F_L (E \xleftarrow{\pm} tR) \right|_{t=0} (I - M) \right) \right]$$
$$\quad + 2d \, \mathbb{E}_{X_0} \left[ \mathrm{tr} \left( (I - M) G_L^\top \Sigma^{-1} \left. \frac{\mathrm{d}}{\mathrm{d}t} G_L (E \xleftarrow{\pm} tR) \right|_{t=0} (I - M) \right) \right].$$

550 Next, we will show that eq. (36) holds for each one of $A_i, B_i, C_i, D_i$ for any $i \in [1, L]$.

551 **1. Equation (36) holds for $A_i$.**

One can easily verify that eqs. (16) and (17) still hold. Furthermore, eqs. (18) and (19) hold for both $F_l$ and $G_l$. With these observations, we can then verify

$$\frac{\mathrm{d}}{\mathrm{d}t}\mathcal{L}(A_i \overset{\pm}{\Leftarrow} tR)\bigg|_{t=0}$$

$$= 2\,\mathbb{E}_{X_0,U_\perp}\left[\mathrm{tr}\left((I-M)F_L^\top(X \overset{\times}{\Leftarrow} U_\Sigma)\frac{\mathrm{d}}{\mathrm{d}t}F_L(X \overset{\times}{\Leftarrow} U_\Sigma, A_i \overset{\pm}{\Leftarrow} tR)\bigg|_{t=0}(I-M)\right)\right]$$

$$\quad + 2d\,\mathbb{E}_{X_0,U_\perp}\left[\mathrm{tr}\left((I-M)G_L^\top(X \overset{\times}{\Leftarrow} U_\Sigma)\Sigma^{-1}\frac{\mathrm{d}}{\mathrm{d}t}G_L(X \overset{\times}{\Leftarrow} U_\Sigma, A_i \overset{\pm}{\Leftarrow} tR)\bigg|_{t=0}(I-M)\right)\right]$$

$$= 2\,\mathbb{E}_{X_0,U_\perp}\left[\mathrm{tr}\left((I-M)F_L^\top U_\Sigma^\top U_\Sigma\frac{\mathrm{d}}{\mathrm{d}t}F_L(A_i \overset{\pm}{\Leftarrow} tU_\Sigma^{-1}RU_\Sigma)\bigg|_{t=0}(I-M)\right)\right]$$

$$\quad + 2d\,\mathbb{E}_{X_0,U_\perp}\left[\mathrm{tr}\left((I-M)G_L^\top U_\Sigma^\top \Sigma^{-1}U_\Sigma\frac{\mathrm{d}}{\mathrm{d}t}G_L(A_i \overset{\pm}{\Leftarrow} tU_\Sigma^{-1}RU_\Sigma)\bigg|_{t=0}(I-M)\right)\right]$$

$$= 2\,\mathbb{E}_{X_0}\left[\mathrm{tr}\left((I-M)F_L^\top\frac{\mathrm{d}}{\mathrm{d}t}F_L(A_i \overset{\pm}{\Leftarrow} trI_d)\bigg|_{t=0}(I-M)\right)\right]$$

$$\quad + 2d\,\mathbb{E}_{X_0}\left[\mathrm{tr}\left((I-M)G_L^\top\Sigma^{-1}\frac{\mathrm{d}}{\mathrm{d}t}G_L(A_i \overset{\pm}{\Leftarrow} trI_d)\bigg|_{t=0}(I-M)\right)\right]$$

$$= \frac{\mathrm{d}}{\mathrm{d}t}\mathcal{L}(A_i \overset{\pm}{\Leftarrow} trI_d)\bigg|_{t=0},$$

where $r = \mathbb{E}_{U_\perp}[U_\Sigma^{-1}RU_\Sigma] = \frac{1}{d}\,\mathrm{tr}\left(\Sigma^{-1/2}R\Sigma^{1/2}\right)$.

**2. Equation (36) holds for $B_i$.**

From the definition of $F_l$ and $G_l$, we can verify that

$$\frac{\mathrm{d}}{\mathrm{d}t}Y_l(B_i \overset{\pm}{\Leftarrow} tR)\bigg|_{t=0}$$

$$= R(F_{i-1} + WG_{i-1})M(X_{i-1}^\top C_i X_{i-1} + D_i)\prod_{j=i+1}^{l}\left(I + b_j M(X_{j-1}^\top C_j X_{j-1} + D_j)\right).$$

Define

$$\overline{F}_l^i = \left(F_{i-1} + B_i F_{i-1}M(X_{i-1}^\top C_i X_{i-1} + D_i)\right)\prod_{j=i+1}^{l}\left(I + b_j M(X_{j-1}^\top C_j X_{j-1} + D_j)\right),$$

$$\overline{G}_l^i = \left(WG_{i-1} + B_i WG_{i-1}M(X_{i-1}^\top C_i X_{i-1} + D_i)\right)\prod_{j=i+1}^{l}\left(I + b_j M(X_{j-1}^\top C_j X_{j-1} + D_j)\right),$$

We then have

$$\frac{\mathrm{d}}{\mathrm{d}t}Y_l(B_i \overset{\pm}{\Leftarrow} tR)\bigg|_{t=0} = \frac{\mathrm{d}}{\mathrm{d}t}\overline{F}_l^i(B_i \overset{\pm}{\Leftarrow} tR)\bigg|_{t=0} + \frac{\mathrm{d}}{\mathrm{d}t}\overline{G}_l^i(B_i \overset{\pm}{\Leftarrow} tR)\bigg|_{t=0}.$$

Similar to eqs. (19) and (21), we can prove that

$$\frac{\mathrm{d}}{\mathrm{d}t}\overline{F}_l^i(X_0 \overset{\times}{\Leftarrow} U_\Sigma, B_i \overset{\pm}{\Leftarrow} tR)\bigg|_{t=0} = U_\Sigma\frac{\mathrm{d}}{\mathrm{d}t}\overline{F}_l^i(B_i \overset{\pm}{\Leftarrow} tU_\Sigma^{-1}RU_\Sigma)\bigg|_{t=0},$$

$$\mathbb{E}_W\left[W^\top\frac{\mathrm{d}}{\mathrm{d}t}\overline{G}_l^i(B_i \overset{\pm}{\Leftarrow} tR)\bigg|_{t=0}\right] = \Sigma^{-1}\frac{\mathrm{d}}{\mathrm{d}t}\overline{G}_l^i(B_i \overset{\pm}{\Leftarrow} t\,\mathrm{tr}(R)I_d)\bigg|_{t=0}.$$

Without loss of generality, we assume that $r = \frac{1}{d}\,\mathrm{tr}\left(\Sigma^{-1/2}R\Sigma^{1/2}\right) \le \frac{1}{d}\,\mathrm{tr}(R)$, and let $\gamma = rd/\mathrm{tr}(R) \le 1$. Then, one can verify that

$$\frac{\mathrm{d}}{\mathrm{d}t}\mathcal{L}(B_i \overset{\pm}{\Leftarrow} tR)\bigg|_{t=0}$$

$$= 2\,\mathbb{E}_{X_0,U_\perp}\left[\mathrm{tr}\left((I-M)F_l^\top(X \overset{\times}{\leftarrow} U_\Sigma)\frac{\mathrm{d}}{\mathrm{d}t}\overline{F}_l^i(X \overset{\times}{\leftarrow} U_\Sigma, B_i \overset{\pm}{\leftarrow} tR)\Big|_{t=0}(I-M)\right)\right]$$

$$+ 2\,\mathbb{E}_{X_0,W}\left[\mathrm{tr}\left((I-M)G_l^\top W^\top\frac{\mathrm{d}}{\mathrm{d}t}\overline{G}_l^i(B_i \overset{\pm}{\leftarrow} tR)\Big|_{t=0}(I-M)\right)\right]$$

$$= 2\,\mathbb{E}_{X_0}\left[\mathrm{tr}\left((I-M)F_l^\top\frac{\mathrm{d}}{\mathrm{d}t}\overline{F}_l^i(B_i \overset{\pm}{\leftarrow} trI_d)\Big|_{t=0}(I-M)\right)\right]$$

$$+ 2\,\mathbb{E}_{X_0}\left[\mathrm{tr}\left((I-M)G_l^\top\Sigma^{-1}\frac{\mathrm{d}}{\mathrm{d}t}\overline{G}_l^i(B_i \overset{\pm}{\leftarrow} t\,\mathrm{tr}(R)I_d)\Big|_{t=0}(I-M)\right)\right]$$

$$= 2\,\mathbb{E}_{X_0}\left[\mathrm{tr}\left((I-M)F_l^\top\frac{\mathrm{d}}{\mathrm{d}t}F_l(B_i \overset{\pm}{\leftarrow} trI_d)\Big|_{t=0}(I-M)\right)\right]$$

$$+ \frac{1}{\gamma}2d\,\mathbb{E}_{X_0}\left[\mathrm{tr}\left((I-M)G_l^\top\Sigma^{-1}\frac{\mathrm{d}}{\mathrm{d}t}G_l(B_i \overset{\pm}{\leftarrow} trI_d)\Big|_{t=0}(I-M)\right)\right]$$

$$= \left(\frac{1}{\gamma}-1\right)2d\,\mathbb{E}_{X_0}\left[\mathrm{tr}\left((I-M)G_l^\top\Sigma^{-1}\frac{\mathrm{d}}{\mathrm{d}t}G_l(B_i \overset{\pm}{\leftarrow} trI_d)\Big|_{t=0}(I-M)\right)\right]$$

$$+ \frac{\mathrm{d}}{\mathrm{d}t}\mathcal{L}(B_i \overset{\pm}{\leftarrow} trI_d)\Big|_{t=0} \geq \frac{\mathrm{d}}{\mathrm{d}t}\mathcal{L}(B_i \overset{\pm}{\leftarrow} trI_d)\Big|_{t=0}.$$

The last inequality assumes the positivity of the term involving $G_l$. Otherwise, one can simply flip the numerator and denominator of $\gamma$ and scale the derivative of $\overline{F}_l$ instead of $G_l$ to yield an additional positive term besides the risk term to finish the proof.

**3. Equation (36) holds for $C_i$, $D_i$.**

Similarly, one can verify that eqs. (22) and (23) still hold (also eqs. (24) to (27)), and finish the proof by following the same reductions as Theorem 1 with $F_l$ and $G_l$. $\qquad\square$

### B.5 Proof of Proposition 3

*Proof.* Let $A_l = a_l I_d$, $B_l = b_l I_d$, $C_l = c_l I_d$ and $D_l = \mathrm{diag}(I_n \otimes D_l^1, D_l^2) + I_{n+1} \otimes D_l^3 + D_l^4 \otimes D_l^5$ for $l \in [1,2]$. Let $Z_l \in \mathbb{R}^{2d \times (3n+3)}$ be the output of the $l$-th attention layer, and let $X_l, Y_l \in \mathbb{R}^{d \times (3n+3)}$ be its first and last $d$ rows respectively. Note that $Y_l$ in this proof does not contain $y_{\text{test}}$.

Let $D_1^1 = \begin{pmatrix} d_x^x & 0 & d_x^y \\ 0 & 0 & 0 \\ d_y^x & 0 & d_y^y \end{pmatrix}$, $D_1^2 = \begin{pmatrix} s_x & 0 & s_y \\ 0 & 0 & 0 \\ 0 & 0 & 0 \end{pmatrix}$ (note that the last row of $D_1^2$ is masked out by $M$, so we simply set it to 0), and $D_1^5 = \begin{pmatrix} 0 & t_x & 0 \\ 0 & 0 & 0 \\ 0 & t_y & 0 \end{pmatrix}$. We use $D$ as an abbreviation for $D_1^4$, and use $d_{i,j}$ to denote the elements in $D$. One can verify that

$$X_1 = X_0 + a_1 X_0 M\big(\mathrm{diag}(I_n \otimes D_1^1, D_1^2) + I_{n+1} \otimes D_1^3 + D_1^4 \otimes D_1^5\big)$$

$$= \begin{bmatrix} (1+a_1 d_x^x)x_1 & a_1 t_x \sum_{i=1}^{n+1} d_{i,1}x_i & a_1 d_x^y x_1 \\ & \cdots & \\ (1+a_1 d_x^x)x_n & a_1 t_x \sum_{i=1}^{n+1} d_{i,n}x_i & a_1 d_x^y x_n \\ (1+a_1 d_x^x)x_{\text{test}} & a_1 t_x \sum_{i=1}^{n+1} d_{i,n+1}x_i & a_1 d_x^y x_{\text{test}} \end{bmatrix}.$$

Similarly, we have

$$Y_1 = Y_0 + b_1 Y_0 M\big(\mathrm{diag}(I_n \otimes D_1^1, D_1^2) + I_{n+1} \otimes D_1^3 + D_1^4 \otimes D_1^5\big)$$

$$= \begin{bmatrix} b_1 d_y^x y_1 & b_1 t_y \sum_{i=1}^{n} d_{i,1}y_i & (1+b_1 d_y^y)y_1 \\ & \cdots & \\ b_1 d_y^x y_n & b_1 t_y \sum_{i=1}^{n} d_{i,n}y_i & (1+b_1 d_y^y)y_n \\ 0 & b_1 t_y \sum_{i=1}^{n} d_{i,n+1}y_i & 0 \end{bmatrix}.$$

By the definition of linear attention, we can show that

$$\mathsf{TF}(Z_0; \{V_l, Q_l\}_{l=1}^2) = (Y_2)_{3n+3} = b_2 Y_1 M\big(c_2 X_1^\top (X_1)_{3n+3} + (D_2)_{3n+3}\big)$$

$$= b_2 c_2 a_1 d_x^y \left(\sum_{i=1}^{3n+2}(Y_1)_i(X_1)_i^\top\right)x_{\text{test}}.$$

Define $\Delta X_1 = [0 \quad a_1 t_x d_{n+1,1} x_{\text{test}} \quad 0 \quad \cdots \quad 0 \quad a_1 t_x d_{n+1,n+1} x_{\text{test}} \quad 0]$, and let $\overline{X}_1 = X_1 - \Delta X_1$, then $\mathsf{TF}(Z_0; \{V_l, Q_l\}_{l=1}^2) = \mathsf{TF}(Z_0; \{V_l, Q_l\}_{l=1}^2, X_1 \leftarrow \overline{X}_1) + \mathsf{TF}(Z_0; \{V_l, Q_l\}_{l=1}^2, X_1 \leftarrow \Delta X_1)$. Let $b_1 d_y^x (1 + a_1 d_x^x) + (1 + b_1 d_y^x) a_1 d_x^x = a$, $b_1 t_y a_1 t_x = b$, $b_2 c_2 a_1 d_x^y = c$, we then have

$$\mathsf{TF}(Z_0; \{V_l, Q_l\}_{l=1}^2, X_1 \leftarrow \overline{X}_1) = c \left( a \sum_{i=1}^n y_i x_i^\top + b \sum_{i=1}^{n+1} \left( \sum_{j=1}^n d_{j,i} y_j \right) \left( \sum_{j=1}^n d_{j,i} x_j^\top \right) \right) x_{\text{test}}$$

$$= c \left( a \sum_{i=1}^n y_i x_i^\top + b \sum_{j=1}^n \sum_{k=1}^n \left( \sum_{i=1}^{n+1} d_{j,i} d_{k,i} \right) y_j x_k^\top \right) x_{\text{test}}, \quad (37)$$

$$\mathsf{TF}(Z_0; \{V_l, Q_l\}_{l=1}^2, X_1 \leftarrow \Delta X_1) = bc \sum_{i=1}^{n+1} \sum_{j=1}^n d_{j,i} y_j d_{n+1,i} x_{\text{test}}^\top x_{\text{test}}$$

$$= bc \sum_{j=1}^n \left( \sum_{i=1}^{n+1} d_{j,i} d_{n+1,i} \right) y_j x_{\text{test}}^\top x_{\text{test}}. \quad (38)$$

Now consider the in-context risk,

$$\mathcal{L}(V, Q) = \mathbb{E}_{Z_0, W} \left\| \mathsf{TF}(Z_0; \{V, Q\}) + W x_{\text{test}} \right\|_2^2$$

$$= \mathbb{E}_{Z_0, W} \left[ \left( \mathsf{TF}(Z_0; \{V, Q\}) + W x_{\text{test}} \right)^\top \left( \mathsf{TF}(Z_0; \{V, Q\}) + W x_{\text{test}} \right) \right]$$

$$= \mathbb{E}_{Z_0, W} \left[ \left( \mathsf{TF}(Z_0; \{V, Q\}, X_1 \leftarrow \overline{X}_1) + W x_{\text{test}} \right)^\top \left( \mathsf{TF}(Z_0; \{V, Q\}, X_1 \leftarrow \overline{X}_1) + W x_{\text{test}} \right) \right]$$

$$+ 2 \mathbb{E}_{Z_0, W} \left[ \mathsf{TF}(Z_0; \{V, Q\}, X_1 \leftarrow \Delta X_1)^\top \left( \mathsf{TF}(Z_0; \{V, Q\}, X_1 \leftarrow \overline{X}_1) + W x_{\text{test}} \right) \right]$$

$$+ \mathbb{E}_{Z_0, W} \left[ \mathsf{TF}(Z_0; \{V, Q\}, X_1 \leftarrow \Delta X_1)^\top \mathsf{TF}(Z_0; \{V, Q\}, X_1 \leftarrow \Delta X_1) \right].$$

In the equation above, the 3-rd part is always positive. We then examine the second part:

$$\mathbb{E}_{Z_0, W} \left[ \mathsf{TF}(Z_0; \{V, Q\}, X_1 \leftarrow \Delta X_1)^\top \left( \mathsf{TF}(Z_0; \{V, Q\}, X_1 \leftarrow \overline{X}_1) + W x_{\text{test}} \right) \right]$$

$$= \mathbb{E}_{Z_0, W} \left[ x_{\text{test}}^\top x_{\text{test}} v_1 x_{\text{test}} + x_{\text{test}}^\top x_{\text{test}} v_2 x_{\text{test}} \right] = 0,$$

where $v_1 = bc \sum_{j=1}^n \left( \sum_{i=1}^{n+1} d_{j,i} d_{n+1,i} \right) y_j^\top c \left( a \sum_{i=1}^n y_i x_i^\top + b \sum_{j=1}^n \sum_{k=1}^n \left( \sum_{i=1}^{n+1} d_{j,i} d_{k,i} \right) y_j x_k^\top \right)$

and $v_2 = bc \sum_{j=1}^n \left( \sum_{i=1}^{n+1} d_{j,i} d_{n+1,i} \right) y_j^\top W$ are independent of $x_{\text{test}}$. Therefore, $\mathcal{L}(V, Q)$ attains its minimum only if $\mathsf{TF}(Z_0; \{V, Q\}, X_1 \leftarrow \Delta X_1) = 0$, implying $d_{n+1,i} = 0$ for $i \in [1, n+1]$.

In the following analysis, we will assume that the last row of $D$ is 0, and let $M \in \mathbb{R}^{n \times (n+1)}$ be the first $n$ rows of $D$. Additionally, we will drop the $c$ factor in eq. (37), since its position could be substituted by $a$ and $b$. We then define $\widetilde{W} = a \sum_{i=1}^n y_i x_i^\top + b \sum_{j=1}^n \sum_{k=1}^n \left( \sum_{i=1}^{n+1} d_{j,i} d_{k,i} \right) y_j x_k^\top$, $X = [x_1 \quad \cdots \quad x_n]$ and $Y = [y_1 \quad \cdots \quad y_n]$. One can verify that

$$\widetilde{W} = a Y X^\top + b Y M M^\top X^\top = a W X X^\top + b W X M M^\top X^\top. \quad (39)$$

Furthermore, the in-context risk could be expanded as

$$\mathcal{L}(V, Q) = \mathbb{E}_{Z_0, W} \left\| \widetilde{W} x_{\text{test}} + W x_{\text{test}} \right\|_2^2 = \mathbb{E}_{Z_0, W} \left[ x_{\text{test}}^\top (\widetilde{W} + W)^\top (\widetilde{W} + W) x_{\text{test}} \right]$$

$$= \mathbb{E}_{Z_0, W} \left[ \mathrm{tr} \left( (\widetilde{W} + W)^\top (\widetilde{W} + W) \right) \right]$$

$$= \mathbb{E}_{Z_0, W} \left[ \mathrm{tr} \left( \widetilde{W}^\top \widetilde{W} \right) + 2 \mathrm{tr} \left( W^\top \widetilde{W} \right) + \mathrm{tr} \left( W^\top W \right) \right].$$

We will use the identity $\mathbb{E}_X [X A X^\top X B X^\top] = \left( \mathrm{tr}(A) \mathrm{tr}(B) + \mathrm{tr}(AB^\top) + d \, \mathrm{tr}(AB) \right) I_d$ for any $A, B \in \mathbb{R}^{n \times n}$, which can be acquired by expanding each element and applying Isserlis' theorem. Let $T_1 = \mathrm{tr}(MM^\top)$ and $T_2 = \mathrm{tr}(MM^\top MM^\top)$, then

$$\mathbb{E}_{Z_0, W} \left[ \mathrm{tr} \left( (a W X X^\top + b W X M M^\top X^\top)^\top (a W X X^\top + b W X M M^\top X^\top) \right) \right]$$

$$
\begin{aligned}
&= \mathbb{E}_{Z_0,W}\left[a^2 \operatorname{tr}\left(XX^\top W^\top WXX^\top\right) + 2ab \operatorname{tr}\left(XX^\top W^\top WXMM^\top X^\top\right)\right] \\
&\quad + \mathbb{E}_{Z_0,W}\left[b^2 \operatorname{tr}\left(XMM^\top X^\top W^\top WXMM^\top X^\top\right)\right] \\
&= d\,\mathbb{E}_{Z_0}\left[a^2 \operatorname{tr}\left(XX^\top XX^\top\right) + 2ab \operatorname{tr}\left(XX^\top XMM^\top X^\top\right) + b^2 \operatorname{tr}\left(XMM^\top X^\top XMM^\top X^\top\right)\right] \\
&= a^2 d^2 n(n+1+d) + 2abd^2(n+1+d)T_1 + b^2 d^2(T_1^2 + (1+d)T_2).
\end{aligned}
$$

Simultaneously, we can verify that $\mathbb{E}_{Z_0,W}[\operatorname{tr}(W^\top W)] = d^2$ and

$$
\mathbb{E}_{Z_0,W}\left[\operatorname{tr}\left(W^\top \widetilde{W}\right)\right] = \mathbb{E}_{Z_0,W}\left[aW^\top WXX^\top + bW^\top WXMM^\top X^\top\right] = ad^2 n + bd^2 T_1.
$$

Combining the results above, we aim to find the optimal $a, b, M$ that minimize

$$
\frac{1}{d^2}\mathcal{L}(V, Q) = c_0 + c_1 T_1 + c_2 T_1^2 + c_3 T_2,
$$

where

$$
c_0 = a^2 n(n+1+d) + 1 + 2an, \quad c_1 = 2ab(n+1+d) + 2b,
$$
$$
c_2 = b^2, \quad c_3 = b^2(1+d).
$$

Since $c_3 \geq 0$, to minimize $\mathcal{L}(V, Q)$ we need to minimize $T_2$. Given that $MM^\top$ is symmetric, we denote its $n$ eigenvalues as $\lambda_i$, $i \in [1, n]$. Then by Cauchy–Schwarz inequality,

$$
\operatorname{tr}\left(MM^\top MM^\top\right) = \sum_{i=1}^{n} \lambda_i^2 \geq \frac{1}{n}\left(\sum_{i=1}^{n} \lambda_i\right)^2 = \frac{1}{n} \operatorname{tr}^2(MM^\top).
$$

Therefore, $\mathcal{L}(V, Q)$ is minimized only if the inequality above holds with equality, which implies that $\lambda_i = \lambda_j$ for any $i \neq j$. This concludes the proof by showing that there exists $\lambda \in \mathbb{R}$ such that $MM^\top = \lambda I_d$, and thus $DD^\top = \operatorname{diag}(\lambda I_d, 0)$. $\qquad\square$

### B.6 Proof of Proposition 6

*Proof.* We will continue from eqs. (37) and (38). After applying token-wise dropout, we have

$$
\begin{aligned}
\mathsf{TF}(Z_0; \{V_l, Q_l\}_{l=1}^2, X_1 \leftarrow \overline{X}_1) &= \sum_{i=1}^{n}(ao_2^{3i-2} + bo_2^{3i})o_1^{3i-2}o_1^{3i}y_i x_i^\top o_1^{3n+1}o_2^{3n+3}x_{\text{test}} \\
&\quad + c\sum_{j=1}^{n}\sum_{k=1}^{n}\left(\sum_{i=1}^{n+1} o_2^{3i-1}d_{j,i}d_{k,i}\right)o_1^{3j}o_1^{3k-2}y_j x_k^\top o_1^{3n+1}o_2^{3n+3}x_{\text{test}}, \qquad (40)\\
\mathsf{TF}(Z_0; \{V_l, Q_l\}_{l=1}^2, X_1 \leftarrow \Delta X_1) &= co_2^{3n+3}\sum_{j=1}^{n}\left(\sum_{i=1}^{n+1} d_{j,i}d_{n+1,i}\right)o_1^{3j}o_1^{3n+1}y_j x_{\text{test}}^\top x_{\text{test}},
\end{aligned}
$$

where $a = b_2 c_2 a_1 d_x^y b_1 d_y^x(1 + a_1 d_x^x)$, $b = b_2 c_2 a_1 d_x^y(1 + b_1 d_y^x)a_1 d_x^x$ and $c = b_2 c_2 a_1 d_x^y b_1 t_y a_1 t_x$. One can verify that our previous analysis about $\mathsf{TF}(Z_0; \{V_l, Q_l\}_{l=1}^2, X_1 \leftarrow \Delta X_1)$ still holds and we thus have $d_{n+1,:} = 0$. We then define:

$$
O_l^1 = \operatorname{diag}(o_l^1, \cdots, o_l^{3n-2}) \in \mathbb{R}^{n \times n}, \quad O_l^2 = \operatorname{diag}(o_l^3, \cdots, o_l^{3n}) \in \mathbb{R}^{n \times n}, \quad \text{for } l \in [2],
$$
$$
O_2^3 = \operatorname{diag}(o_2^2, \cdots, o_2^{3n+2}) \in \mathbb{R}^{(n+1) \times (n+1)}.
$$

By defining

$$
\widetilde{W} = \sum_{i=1}^{n}(ao_2^{3i-2} + bo_2^{3i})o_1^{3i-2}o_1^{3i}y_i x_i^\top + c\sum_{j=1}^{n}\sum_{k=1}^{n}\left(\sum_{i=1}^{n+1} o_2^{3i-1}d_{j,i}d_{k,i}\right)o_1^{3j}o_1^{3k-2}y_j x_k^\top,
$$

One can verify that

$$
\widetilde{W} = A + B + C \triangleq aYO_1^2 O_2^1 O_1^1 X^\top + bYO_1^2 O_2^2 O_1^1 X^\top + cYO_1^2 MO_2^3 M^\top O_1^1 X^\top.
$$

Then, we will compute the expectation of each term in the following decomposition:

$$\mathcal{L}(V, Q) = \mathbb{E}_{Z_0, W}\left[\text{tr}\left(\widetilde{W}^\top \widetilde{W}\right) + 2\,\text{tr}\left(W^\top \widetilde{W}\right) + \text{tr}\left(W^\top W\right)\right],$$

Specifically, let $T_1 = \text{tr}\left(MM^\top\right)$, $T_2 = \text{tr}\left(MM^\top MM^\top\right)$, $T_3 = \|M\|_4^4$, $T_4 = \sum_{i=1}^{n} \|M_{i,:}\|_2^4$, $T_5 = \sum_{j=1}^{n+1} \|M_{:,j}\|_2^4$, we then have

$$\mathbb{E}[\text{tr}\left(A^\top A\right)] = a^2 d^2 (np^3 + n(n-1)p^6 + (1+d)np^3),$$
$$\mathbb{E}[\text{tr}\left(B^\top B\right)] = b^2 d^2 (np^3 + n(n-1)p^6 + (1+d)np^3),$$
$$\mathbb{E}[\text{tr}\left(C^\top C\right)] = c^2 d^2 (p^6 T_1^2 + (1+d)(p^4 - p^6)T_4 + (1+d)(p^5 - p^6)T_5$$
$$+ (1+d)(p^3 - p^4 - p^5 + p^6)T_3 + (p^3 - p^4)T_4 + p^4 T_2 + dp^6 T_2),$$
$$\mathbb{E}[\text{tr}\left(A^\top B\right)] = abd^2 (np^4 + n(n-1)p^6 + (1+d)np^4),$$
$$\mathbb{E}[\text{tr}\left(A^\top C\right)] = acd^2 ((p^4 + (n-1)p^6)T_1 + (1+d)p^4 T_1),$$
$$\mathbb{E}[\text{tr}\left(B^\top C\right)] = bcd^2 ((p^4 + (n-1)p^6)T_1 + (1+d)p^4 T_1),$$
$$\mathbb{E}[\text{tr}\left(W^\top A\right)] = ad^2 np^3, \quad \mathbb{E}[\text{tr}\left(W^\top B\right)] = bd^2 np^3, \quad \mathbb{E}[\text{tr}\left(W^\top C\right)] = cd^2 p^3 T_1.$$

Summarizing our analysis above, $\min_M \mathcal{L}(V, Q)$ is equivalent to:

$$\min_M \left\{c_0 + c_1 T_1 + c_2 T_2 + c_3 T_3 + c_4 T_4 + c_5 T_5 + c_6 T_1^2\right\},$$

where

$$c_0 = 1 + n(2+d)p^3(a^2 + b^2) + 2np^3(a+b) + 2n(2+d)p^4 ab + n(n-1)p^6(a+b)^2,$$
$$c_1 = 2(a+b)c(p^4 + (n-1)p^6 + (1+d)p^4) + 2cp^3,$$
$$c_2 = c^2(p^4 + dp^6),$$
$$c_3 = c^2(1+d)(p^3 - p^4 - p^5 + p^6),$$
$$c_4 = c^2((1+d)(p^4 - p^6) + (p^3 - p^4)),$$
$$c_5 = c^2(1+d)(p^5 - p^6),$$
$$c_6 = c^2 p^6.$$

It is easy to verify that $c_2, c_3, c_4, c_5, c_6 \geq 0$. $\qquad\square$

## B.7 Proof of Proposition 7

**Proposition 7** (Restate). *Let $d_p$ denote the number of non-EOS tokens. Given any $L$-layer, single-head, $d$-dimensional linear-attention transformer with EOS tokens:*

$$\text{TF}\left(Z_0; \{V_l, Q_l, P_l\}_{l\in[L]}\right) = (Z_L)_{:, d_p+1}, \quad (Z_0)_{:, d_p+1} = 0,$$

*where*

$$Z_l \in \mathbb{R}^{d\times(d_p+1)}, \ V_l, Q_l \in \mathbb{R}^{d\times d}, \ P_l \in \mathbb{R}^{(d_p+1)\times(d_p+1)},$$
$$Z_l = Z_{l-1} + V_l Z_{l-1} M(Z_{l-1}^\top Q_l Z_{l-1}^\top + P_l), \quad M = \text{diag}(I_{d_p}, 0).$$

*There exists an $L$-layer, two-head, $2d$-dimensional linear-attention transformer operating without EOS tokens:*

$$\text{TF}\left(\overline{Z}_0; \{\overline{V}_l^h, \overline{Q}_l^h, \overline{P}_l^h\}_{l\in[L], h\in[2]}\right) = (\overline{Z}_L)_{d:2d, d_p},$$

*where*

$$\overline{Z}_l \in \mathbb{R}^{2d\times d_p}, \ \overline{V}_l^h, \overline{Q}_l^h \in \mathbb{R}^{2d\times 2d}, \ \overline{P}_l^h \in \mathbb{R}^{d_p\times d_p},$$
$$\overline{Z}_l = \overline{Z}_{l-1} + \sum_{h=1}^{2} \overline{V}_l^h \overline{Z}_{l-1}(\overline{Z}_{l-1}^\top \overline{Q}_l^h \overline{Z}_{l-1}^\top + \overline{P}_l^h).$$

*Such that for any $Z \in \mathbb{R}^{d\times d_p}$, by letting $Z_0 = [Z \quad 0]$ and $\overline{Z}_0 = \begin{bmatrix} Z \\ 0 \end{bmatrix}$, we have*

$$\text{TF}\left(Z_0; \{V_l, Q_l, P_l\}_{l\in[L]}\right) = \text{TF}\left(\overline{Z}_0; \{\overline{V}_l^h, \overline{Q}_l^h, \overline{P}_l^h\}_{l\in[L], h\in[2]}\right).$$

*Proof.* We construct $\overline{V}_l^h$, $\overline{Q}_l^h$, and $\overline{P}_l^h$ as follows:

$$\overline{V}_l^1 = \begin{bmatrix} V_l & 0 \\ 0 & 0 \end{bmatrix}, \quad \overline{Q}_l^1 = \begin{bmatrix} Q_l & 0 \\ 0 & 0 \end{bmatrix}, \quad \overline{P}_l^1 = (P_l)_{1:d_p,1:d_p},$$

$$\overline{V}_l^2 = \begin{bmatrix} 0 & 0 \\ V_l & 0 \end{bmatrix}, \quad \overline{Q}_l^2 = \begin{bmatrix} 0 & Q_l \\ 0 & 0 \end{bmatrix}, \quad \overline{P}_l^2 = \begin{bmatrix} 0 & (P_l)_{:,d_p+1} \end{bmatrix}.$$

We will show that for any $l \in [L]$, it satisfies $\overline{Z}_l = \begin{bmatrix} (Z_l)_{:,(1:d_p-1)} & (Z_l)_{:,d_p} \\ 0 & (Z_l)_{:,d_p+1} \end{bmatrix}$. One can verify that it holds trivially for $l = 0$. Then, suppose it holds for some $l = k-1$, we have

$$\overline{Z}_k = \overline{Z}_{k-1} + \overline{V}_k^1 \overline{Z}_{k-1}(\overline{Z}_{k-1}^\top \overline{Q}_k^1 \overline{Z}_{k-1}^\top + \overline{P}_k^1) + \overline{V}_k^2 \overline{Z}_{k-1}(\overline{Z}_{k-1}^\top \overline{Q}_k^2 \overline{Z}_{k-1}^\top + \overline{P}_k^2)$$

$$= \overline{Z}_{k-1} + \begin{bmatrix} V_k(Z_{k-1})_{:,1:d_p}\left((Z_{k-1})_{:,1:d_p}^\top Q_k(Z_{k-1})_{:,1:d_p} + (P_k)_{1:d_p,1:d_p}\right) \\ 0 \end{bmatrix}$$

$$+ \begin{bmatrix} 0 \\ V_k(Z_{k-1})_{:,1:d_p} \end{bmatrix} \left( \begin{bmatrix} 0 & (Z_{k-1})_{:,1:d_p}^\top Q_k(Z_{k-1})_{:,d_p+1} \end{bmatrix} + \begin{bmatrix} 0 & (P_k)_{:,d_p+1} \end{bmatrix} \right)$$

$$= \overline{Z}_{k-1} + \begin{bmatrix} V_k Z_{k-1} M\left(Z_{k-1}^\top Q_k(Z_{k-1})_{:,1:d_p} + (P_k)_{:,1:d_p}\right) \\ 0 \end{bmatrix}$$

$$+ \begin{bmatrix} 0 & 0 \\ 0 & V_k Z_{k-1} M\left(Z_{k-1}^\top Q_k(Z_{k-1})_{:,d_p+1} + (P_k)_{:,d_p+1}\right) \end{bmatrix}$$

$$= \begin{bmatrix} (Z_k)_{:,1:d_p} \\ 0 \end{bmatrix} + \begin{bmatrix} 0 & 0 \\ 0 & (Z_k)_{:,d_p+1} \end{bmatrix}.$$

The proof is complete. $\qquad\square$

# C Experiment Details and Additional Results

In this section, we present experiment details and additional results not included in the main text due to space limitations. Our experiments are conducted on an A100 40G GPU. It takes around 30 GPU hours to fully reproduce our results[1].

## C.1 Synthetic Experiments on Linear Transformers

We consider training linear-attention transformers on random linear regression instances. We take embedding dimension $d = 4$, and the distributions for generating $x_i$ and $w_i$ are both $P_x = P_w = \mathcal{N}(0, I_d)$. We optimize the ICL risk for $L$-layer linear transformers with $n$ in-context demonstrations using AdamW, where $L \in [3]$ and $n \in [5, 30]$. Each gradient step is computed from a batch size of 1000. We additionally apply $\ell_1$ regularization to simplify the found solutions. For training efficiency and stability, we restrict the $A_l$, $B_l$, and $C_l$ matrices to $\mathcal{S}_I$ during training, and initialize $D_l \in \mathbb{R}^{d_p \times d_p}$ with i.i.d. Gaussian matrices. For each case, we train 40 models with different random seeds, and report the minimum achieved ICL risk to approximate the global minimum.

To reproduce the task vector mechanism, we focus on transformers trained with triplet-formatted prompts. The training procedure is identical to the above. For inference, we restrict $P_w$ to rank-one coefficient matrices, by letting $W = w_1 w_2^\top$, where $w_1, w_2 \sim \mathcal{N}(0, I_d)$. We first generate normal ICL prompts to generate task vectors as the hidden states of the last arrow token after the first attention layer, and then inject them into zero-shot prompts after normalization. The final outputs $\hat{y}_{\text{test}}$ are taken as the output of these injected zero-shot prompts after being processed with the same transformer model. We compute the final risk as $\mathbb{E}\left\|\frac{\hat{y}_{\text{test}}}{\|\hat{y}_{\text{test}}\|} + \frac{y_{\text{test}}}{\|y_{\text{test}}\|}\right\|$ to simulate the layer normalization blocks in practical LLMs. The reported scores are averaged for $n \in [5, 30]$.

## C.2 Experiments on Practical LLMs

**Datasets.** Following the settings of the original task vector method [7], our study covers 33 tasks in 5 categories. The detailed description for each task is provided in Table 3.

---

[1]The source code is available in supplementary materials.

Table 3: Descriptions of the tasks used in our empirical studies.

| Category | Task | Example | Description |
|---|---|---|---|
| Knowledge | Contry to Capital | France → Paris | Output the capital city of the given country. |
| | Person to Language | Macron → French | Output the native language of the given person. |
| | Location to Continent | Paris → Europe | Output the corresponding continent of the given location. |
| | Religion | Saladin → Muslim | Output the associated religion of the given location or person. |
| Algorithmic | List First | [a,b,c] → a | Output the first item in the given list. |
| | List Last | [a,b,c] → c | Output the last item in the given list. |
| | Next Letter | a → b | Output the next letter of the given letter in the alphabet. |
| | Prev Letter | b → a | Output the previous letter of the given letter in the alphabet. |
| | To Upper | a → A | Output the corresponding uppercase letter of the given lowercase letter. |
| | To Lower | A → a | Output the corresponding lowercase letter of the given uppercase letter. |
| Translation | English to French | hello → bonjour | Translate the given word in English to French. |
| | English to Italian | hello → ciao | Translate the given word in English to Italian. |
| | English to Spanish | hello → hola | Translate the given word in English to Spanish. |
| | French to English | bonjour → hello | Translate the given word in French to English. |
| | Italian to English | ciao → hello | Translate the given word in Italian to English. |
| | Spanish to English | hola → hello | Translate the given word in Spanish to English. |
| Linguistic | Present to Gerund | go → going | Output the corresponding gerund form of the given verb in present simple tense. |
| | Present to Past | go → went | Output the corresponding past simple form of the given verb in present simple tense. |
| | Present to Past Perfect | go → gone | Output the corresponding past perfect form of the given verb in present simple tense. |
| | Gerund to Present | going → go | Output the corresponding present simple form of the given verb in gerund form. |
| | Past to Present | went → go | Output the corresponding present simple form of the given verb in past simple tense. |
| | Past Perfect to Present | gone → go | Output the corresponding present simple form of the given verb in past perfect tense. |
| | Singular to Plural | dog → dogs | Output the corresponding plural form of the given noun in singular form. |
| | Plural to Singular | dogs → dog | Output the corresponding singular form of the given noun in plural form. |
| | Antonym | happy → sad | Output the antonym of the given adjective. |
| Bijection | To Upper & Lower | a ↔ A | Output the given letter in uppercase if it is in lowercase, and vice versa. |
| | English & French | hello ↔ bonjour | Translate the given word to French if it is in English, and vice versa. |
| | English & Italian | hello ↔ ciao | Translate the given word to Italian if it is in English, and vice versa. |
| | English & Spanish | hello ↔ hola | Translate the given word to Spanish if it is in English, and vice versa. |
| | Present & Gerund | go ↔ going | Output the given verb in gerund form if it is in present simple tense, and vice versa. |
| | Present & Past | go ↔ went | Output the given verb in past simple form if it is in present simple tense, and vice versa. |
| | Present & Past Perfect | go ↔ gone | Output the given verb in past perfect form if it is in present simple tense, and vice versa. |
| | Singular & Plural | dog ↔ dogs | Output the given noun in plural form if it is in singular form, and vice versa. |

Table 4: Accuracy comparison between standard ICL (Baseline), the task vector method (TaskV), and our strategy (TaskV-M). The experiment is conducted on Pythia-12B with $n = 10$.

| | Method | Knowledge | Algorithmic | Translation | Linguistic | Bijection | Average |
|---|---|---|---|---|---|---|---|
| 0-shot | Baseline | $6.60 \pm 1.59$ | $14.07 \pm 1.45$ | $8.60 \pm 0.68$ | $12.53 \pm 1.57$ | $10.31 \pm 0.70$ | $10.82 \pm 0.48$ |
| | TaskV | $\mathbf{63.30} \pm 2.62$ | $\mathbf{84.73} \pm 1.22$ | $\mathbf{62.07} \pm 0.98$ | $\mathbf{82.58} \pm 1.22$ | $\mathbf{42.27} \pm 0.92$ | $\mathbf{66.40} \pm 0.96$ |
| 1-shot | Baseline | $61.80 \pm 5.45$ | $72.80 \pm 1.15$ | $43.27 \pm 2.92$ | $57.07 \pm 1.15$ | $41.91 \pm 2.83$ | $53.95 \pm 1.02$ |
| | TaskV | $76.40 \pm 2.40$ | $\mathbf{84.20} \pm 1.05$ | $\mathbf{71.47} \pm 1.41$ | $\mathbf{87.16} \pm 2.04$ | $53.11 \pm 2.37$ | $73.59 \pm 0.79$ |
| | TaskV-M | $\mathbf{77.70} \pm 2.52$ | $83.73 \pm 1.37$ | $71.00 \pm 1.48$ | $86.80 \pm 1.59$ | $\mathbf{53.87} \pm 2.90$ | $\mathbf{73.68} \pm 0.90$ |
| 2-shot | Baseline | $70.30 \pm 3.71$ | $82.13 \pm 0.54$ | $60.80 \pm 1.81$ | $81.16 \pm 1.57$ | $50.76 \pm 2.17$ | $68.41 \pm 0.64$ |
| | TaskV | $80.30 \pm 2.46$ | $\mathbf{87.00} \pm 1.63$ | $76.13 \pm 3.77$ | $89.33 \pm 0.70$ | $58.67 \pm 2.44$ | $77.41 \pm 0.50$ |
| | TaskV-M | $\mathbf{81.60} \pm 1.56$ | $86.47 \pm 0.40$ | $\mathbf{77.27} \pm 2.53$ | $\mathbf{89.51} \pm 0.88$ | $\mathbf{59.24} \pm 2.48$ | $\mathbf{77.87} \pm 0.76$ |
| 3-shot | Baseline | $77.60 \pm 2.40$ | $81.87 \pm 0.81$ | $68.13 \pm 2.02$ | $86.31 \pm 1.93$ | $55.73 \pm 1.60$ | $73.20 \pm 0.31$ |
| | TaskV | $84.00 \pm 2.76$ | $86.33 \pm 1.17$ | $\mathbf{79.53} \pm 2.27$ | $92.00 \pm 0.67$ | $58.76 \pm 1.53$ | $79.06 \pm 0.67$ |
| | TaskV-M | $\mathbf{85.40} \pm 2.31$ | $\mathbf{87.07} \pm 1.18$ | $78.13 \pm 1.86$ | $\mathbf{92.84} \pm 0.68$ | $\mathbf{59.56} \pm 1.27$ | $\mathbf{79.54} \pm 0.35$ |
| 4-shot | Baseline | $78.40 \pm 1.83$ | $82.73 \pm 0.44$ | $72.40 \pm 1.24$ | $88.89 \pm 1.25$ | $57.91 \pm 1.46$ | $75.46 \pm 0.64$ |
| | TaskV | $83.80 \pm 1.12$ | $87.60 \pm 1.81$ | $\mathbf{80.20} \pm 2.39$ | $\mathbf{92.18} \pm 0.96$ | $59.38 \pm 0.47$ | $79.59 \pm 0.62$ |
| | TaskV-M | $\mathbf{84.30} \pm 1.50$ | $\mathbf{88.13} \pm 0.81$ | $80.00 \pm 2.67$ | $91.87 \pm 1.25$ | $\mathbf{60.31} \pm 0.86$ | $\mathbf{79.87} \pm 0.51$ |

**Prompt Template.** The template used to construct ICL demonstrations is "Example:$\{x_i\} \rightarrow \{y_i\}$, where $x_i$ and $y_i$ are subsequently replaced by the input and output of the semantic mapping. For the query part, $y_i$ is omitted from the prompt. After concatenating each demonstration with "\n", an example of the full input prompt is:

$$\text{Example:}\{x_1\} \rightarrow \{y_1\}\backslash n \cdots \text{Example:}\{x_n\} \rightarrow \{y_n\}\backslash n\text{Example:}\{x_{\text{test}}\} \rightarrow$$

**Evaluation.** To evaluate the $N$-shot performance, we generate $50 \times (N + 1)$ i.i.d. prompts for each task with number of demonstrations $n = 10$ for task vector extraction. The hidden states of the last $\rightarrow$ token, which is also literally the last token in the prompt, are recorded for every layer in the transformer. Thereafter, we generate another 50 i.i.d. prompts with $N$ demonstrations, where $x_{\text{test}}$ is selected to be distinct from the previous chosen ones. The final accuracy is measured by whether the next word predicted matches the expected answer. The performance of the standard ICL method (Baseline) is acquired by inferring without interference. For the task vector method (TaskV) and our multi-vector variant (TaskV-M), the extracted task vectors are injected to replace the hidden states of the arrow $\rightarrow$ tokens at a specified layer $l$. For TaskV, only the last arrow token is injected, while for TaskV-M, each of the $N + 1$ arrow tokens is injected with the $N + 1$ extracted task vectors for the same task. The performance is reported for the one layer $l \in L$ achieving the highest accuracy. For each case, the mean and standard deviation are evaluated through 5 independent trials.

**Additional Results.** Besides Llama-13B, we also observe consistent accuracy improvement of our TaskV-M method on the Pythia-12B model, as reported in Table 4.

# D  Additional Discussions

## D.1  Last Task Vector Weights the Most

While our analysis of linear-attention models suggests that each formed task vector (i.e., the hidden state at each arrow token) contributes equally to the final prediction, this assumption does not fully hold in practical LLMs. As demonstrated by the conflicting tasks experiment in [7], injecting a task vector from task $B$ into an ICL prompt designed for task $A$ causes the model to predominantly perform task $B$. This behavior indicates that LLMs largely rely on the last arrow token to determine the task identity. We attribute this to the causal attention mechanism used in practical LLMs, which is not captured by our current theoretical analysis. In causal attention, only the final arrow token can aggregate information from the entire preceding context, making it the most informative and influential for prediction. This explains why our multi-vector strategy offers modest, though consistent, performance gains. The improvement suggests that intermediate arrow tokens do participate in the inference process, albeit less effectively. Enhancing how LLMs utilize information from all arrow tokens remains a promising direction for improving task vector accuracy and robustness.

 **D.2  Decoding the Vocabulary of Task Vectors**

Multiple prior works [7, 16] have observed an interesting phenomenon: when task vectors are extracted and passed through the final classification layer, the top predicted tokens often belong to the output space of the corresponding task. This effect is particularly prominent in the GPT-J model. Interestingly, we find that this behavior can be naturally explained by our analysis of linear models. Specifically, we assume that the hidden state space has dimensionality at least $2d$, where the first $d$ dimensions represent the input ($x_i$) and the last $d$ dimensions represent the output ($y_i$). Task vectors constructed under this architecture preserve this layout: the first half encodes a linear combination of $x_i$, and the second half encodes a linear combination of $y_i$. In the final layer, the model predicts $y_{\text{test}}$ by extracting the last $d$ dimensions of the final token. When this same mechanism is applied to a task vector, it naturally produces a linear combination of the $y_i$ values, thereby generating outputs aligned with the task's output space. This indicates that practical LLMs adopt a similar partition in the hidden state space, justifying our prompt structure for linear model analysis.

**D.3  Limitations**

While our analysis provides new insights into the emergence and functionality of task vectors, it is primarily conducted on simplified linear-attention transformers and synthetic tasks, which may not fully capture the complexity of real-world LLMs. Moreover, our theoretical framework focuses on middle-layer representations and does not fully account for deeper interactions across layers or the role of fine-tuned components such as layer normalization and multi-head attention.

**D.4  Broader Impacts**

This work advances the theoretical understanding of in-context learning and task vector mechanisms, which can lead to more efficient and interpretable language models. By enabling faster inference through task vectors, it may reduce the computational cost and energy consumption of large-scale deployment, thereby making AI systems more accessible and environmentally sustainable. Improved interpretability could also enhance trust and transparency in AI applications across education, healthcare, and other socially beneficial domains.

As task vector methods improve efficiency and transferability, they may also be misused to replicate or extract functionality from proprietary models without authorization, raising concerns around model intellectual property. Additionally, while interpretability is often framed as a benefit, deeper insights into model internals could be exploited to engineer adversarial inputs or extract sensitive training data. Careful consideration and mitigation strategies are essential to ensure that such work aligns with the broader goals of safe and beneficial AI.

