# OpenReview forum: "Understanding Task Vectors in In-Context Learning: Emergence, Functionality, and Limitations"
_NeurIPS.cc/2025/Conference — Submitted to NeurIPS 2025_

### Official Review · Reviewer_pehy · 2025-07-01

**Clarity:** 3
**Significance:** 3
**Originality:** 2
**Rating:** 5
**Confidence:** 4

**Summary:**

This paper attempts to provide a theoretical observation of how task vectors of ICL emerge within self-attention. Specifically, the authors simulate a linear regression task and augment it to an ICL-style linear regression form (triplet-formatted prompts). On these tasks, they identify low-loss solutions of self-attention in which the task vector can be described as a weighted average of the ICL demonstrations. As a limitation of such task vector, they investigate and successfully predict the failure of ICL (in both synthetic and real-world scenarios) under bijective mappings (or full-rank mappings). Furthermore, they present several empirical observations and applications in real LLMs, aiming to transfer their theoretical findings into practical settings.

**Questions:**

1. The paper does not provide a detailed analysis of how the theoretical results are reflected in the parameters of real-world LLMs. Could the authors provide more insights into how the theoretical framework can be applied to real LLMs, or at least discuss the challenges, limitations, and potential future directions for such applications?
2. Regarding Weakness 2, could you provide a detailed explanation (and incorporate it into the revised paper) of your justification for using such parameter constraints?
3. Regarding Weakness 4, could you provide additional results on more LLMs and data samples to strengthen the empirical validation of the theoretical framework?

I would be happy to raise the score to 5 if the author can properly address these concerns in the rebuttal.

**Ethical Concerns:**

["NO or VERY MINOR ethics concerns only"]

**Final Justification:**

After reading the rebuttal, I think most of my concerns have been addressed. I decide to raise my rating to 5.

**Limitations:**

yes

**Quality:**

2

**Strengths And Weaknesses:**

**Strengths**

1. The theoretical simulations presented by the authors are based on reasonable assumptions and yield credible results. Given that empirical findings on ICL have indicated the critical role of these structured tokens (such as the $\rightarrow$ in this paper) in LLMs, theoretically investigating these tokens is valuable. Moreover, the theoretical results align well with existing empirical evidence (both of these based on feature correlation patterns (e.g., saliency maps or induction heads) or task vectors) thus offering potential theoretical support for how ICL emerges in LLMs.

2. The discussion of bijective tasks (Proposition 4) is both reasonable and highly insightful. Intuitively, I believe that tasks where the input space is significantly larger than the output space (e.g., classification) operate under fundamentally different ICL mechanisms compared to bijective tasks (e.g., mapping countries to their capitals). This paper successfully brings attention to this issue and uses its theoretical framework to make predictions and offer explanations, which is highly commendable.

3. The paper is well-written and easy to follow.

**Weaknesses**

1. The paper did not examine how the theoretically optimal parameter patterns are reflected in the parameters of real-world LLMs. Instead, the paper only investigated the alignment between the saliency patterns in actual LLMs and their theoretical results, which is insufficient to uniquely confirm the correctness of the theoretical framework. While I acknowledge that analyzing the parameters of real LLMs can be very challenging, the lack of such analysis inevitably weakens the credibility of the paper’s conclusions. Additionally, it remains unclear how the simulated simple linear regression tasks map onto LLMs trained on real-world data. Thus, although the authors propose an elegant theoretical framework, applying it to real-world LLMs still requires considerable effort.

2. This paper seem to impose strong conditions in constructing the theoretical framework. For instance, in Equation (6), they assume that $V_l$ and $Q_l$ are block-diagonal matrices, which may lack intuitive or empirical justification. Specifically, if the goal is to simulate the training of real LLMs, where parameters are almost unconstrained, such explicit constraints may distance the current theoretical framework from practical scenarios. Did I miss any crucial justification for this assumption?

3. The proposed TaskV-M application method appears to yield only marginal performance gains. While I appreciate the authors’ effort to translate their theoretical findings into practical applications, the method may not serve as a strong empirical validation of their conclusions.

4. The empirical results (Fig. 3, Table 1) presented in the paper appear to be anecdotal. I would like to see additional results on more LLMs and input samples.

---

> ### Author Rebuttal · Authors · 2025-07-31
>
> We thank Reviewer pehy for the thoughtful comments and for recognizing the significance and insights of our analysis. Below, we address each of the reviewer’s questions in detail:
>
> **Q1. Parameter Analysis in Practical LLMs**
>
> We agree that directly analyzing the optimal parameter structure of practical LLMs is a valuable and exciting direction. However, it poses substantial challenges and is beyond the current scope of our work. Unlike our theoretical setting, which explicitly minimizes the ICL risk, real-world LLMs are pretrained on diverse corpora with objectives not specifically designed for ICL. As a result, the parameter patterns in pretrained models may reflect a mixture of capabilities with ICL being just one among many, making principled isolation and analysis particularly difficult. Nevertheless, we believe our analysis lays foundational insights that will aid future efforts in understanding and designing LLMs specialized for in-context learning.
>
> We acknowledge this limitation in Appendix D.3, noting that our theoretical analysis is conducted under a simplified linear-attention setting. Nonetheless, we believe our results offer clear mechanistic insights into the emergence, function, and limitations of task vectors, and can guide future improvements to ICL in practice. Concretely, our contributions have the following practical implications:
>
> * Our analysis of embedding concatenation helps identify the optimal layer for task vector extraction.
>
> * The bijection task counterexample clarifies when and why task vectors fail, despite ICL success.
>
> * Our study of attention weights helps determine the preferred ordering of in-context demonstrations.
>
> * The validation of multi-vector injection highlights how task vectors can be extended to few-shot settings.
>
> * Our critical point analysis sheds light on the conditions that encourage task vector emergence during fine-tuning.
>
> We hope these insights will inspire future work on analyzing or even designing architectures that more explicitly support ICL-style learning.
>
> **Q2. Justification for the Block Diagonal Formulation**
>
> The block-diagonal formulation of the \$V\_l\$ and \$Q\_l\$ matrices is a widely adopted assumption in theoretical studies of ICL for transformer models, as it facilitates tractable analysis [1, 14, 20, 25]. Prior work by Ahn et al. [1] demonstrates that the global minimizer of single-layer linear-attention transformers indeed exhibits such block-diagonal structure. Although finding exact solutions for multi-layer transformers is more involved, it is reasonable to conjecture that similar structural patterns hold. Empirically, we observe that when optimizing the full matrices, gradient-based training also tends to converge to block-diagonal solutions.
>
> Intuitively, given the high dimensionality of hidden states in modern LLMs, it is plausible to assume that the $x$ and $y$ components can be projected into orthogonal or nearly orthogonal subspaces. This motivates a decomposition of the projection matrices $V_l$ and $Q_l$ into two separate parts that operate independently on $x_i$ and $y_i$, which can be equivalently formulated as the block-diagonal structures.
>
> We will clarify and expand upon this rationale in the revised manuscript.
>
> **Q3. Performance of TaskV-M**
>
> We agree that the performance gains of TaskV-M over TaskV are not dramatic across all ICL tasks. However, the goal of TaskV-M is not to surpass state-of-the-art ICL techniques, but to demonstrate that the task vector framework can be systematically extended by injecting multiple vectors simultaneously. This is especially valuable for complex tasks that inherently require higher-rank representations.
>
> Our results on bijection tasks clearly validate this motivation: TaskV-M yields notable improvements over the standard TaskV method. For other simpler tasks, the marginal gains from TaskV-M suggest that the expressiveness of $W$ may not be the primary performance bottleneck. We believe these insights facilitate the design of future ICL and task vector methods.
>
> **Q4. Further Experimental Results**
>
> Due to rebuttal constraints, we are unfortunately unable to include additional figures. However, we reproduce below extended results from Table 1 that illustrate the failure of task vectors on bijection tasks across a broader range of LLMs and varying numbers of input demonstrations. These results further support our claims that:
>
> * Task vectors systematically fail on bijection tasks, even when increasing the number of demonstrations.
>
> * The failure is consistent across multiple model architectures, reinforcing that the issue stems from a fundamental expressiveness limitation rather than model-specific artifacts.
>
> Below are the summarized performances for various LLMs on bijection tasks with $n = 10$ demonstrations. Left is ICL accuracy, and right is task vector accuracy.
>
> | Task | GPT-J | Pythia-6.9B | Pythia-12B | Llama-7B | Llama-13B |
> |---|---|---|---|---|---|
> | To Upper | 1.00/0.08 | 0.90/0.28 | 0.96/0.24 | 1.00/0.55 | 1.00/0.58 |
> | English $\to$ French | 0.64/0.50 | 0.38/0.28 | 0.52/0.28 | 0.54/0.35 | 0.64/0.32 |
> | English $\to$ Italian | 0.68/0.56 | 0.62/0.48 | 0.60/0.56 | 0.70/0.47 | 0.72/0.44 |
> | English $\to$ Spanish | 0.70/0.52 | 0.62/0.56 | 0.66/0.56 | 0.64/0.43 | 0.84/0.56 |
> | Present $\to$ Gerund | 0.64/0.36 | 0.44/0.32 | 0.40/0.22 | 0.80/0.41 | 0.74/0.26 |
> | Present $\to$ Past | 0.60/0.38 | 0.48/0.36 | 0.54/0.16 | 0.52/0.33 | 0.68/0.44 |
> | Present $\to$ Perfect | 0.46/0.14 | 0.38/0.24 | 0.46/0.28 | 0.55/0.33 | 0.54/0.42 |
> | Singular $\to$ Plural | 0.66/0.50 | 0.56/0.28 | 0.44/0.28 | 0.76/0.51 | 0.80/0.52 |
> | Antonym | 0.86/0.78 | 0.76/0.66 | 0.76/0.70 | 0.83/0.73 | 0.78/0.72 |
>
> Below are the results for $n = 20$ demonstrations.
>
> | Task | GPT-J | Pythia-6.9B | Pythia-12B | Llama-7B | Llama-13B |
> |---|---|---|---|---|---|
> | To Upper | 1.00/0.12 | 1.00/0.32 | 0.94/0.38 | 1.00/0.48 | 1.00/0.60 |
> | English $\to$ French | 0.74/0.54 | 0.44/0.40 | 0.52/0.40 | 0.52/0.34 | 0.58/0.34 |
> | English $\to$ Italian | 0.62/0.54 | 0.66/0.46 | 0.68/0.48 | 0.78/0.50 | 0.74/0.48 |
> | English $\to$ Spanish | 0.80/0.58 | 0.54/0.38 | 0.56/0.40 | 0.78/0.58 | 0.84/0.58 |
> | Present $\to$ Gerund | 0.54/0.26 | 0.54/0.22 | 0.46/0.14 | 0.84/0.44 | 0.94/0.38 |
> | Present $\to$ Past | 0.66/0.26 | 0.54/0.30 | 0.58/0.28 | 0.72/0.30 | 0.76/0.44 |
> | Present $\to$ Perfect | 0.42/0.18 | 0.44/0.20 | 0.46/0.24 | 0.48/0.30 | 0.52/0.48 |
> | Singular $\to$ Plural | 0.64/0.40 | 0.62/0.36 | 0.52/0.28 | 0.80/0.52 | 0.94/0.42 |
> | Antonym | 0.84/0.76 | 0.84/0.70 | 0.90/0.82 | 0.90/0.84 | 0.90/0.84 |
>
> We will include these results and further saliency analysis results in the revision.

---

> > ### Comment · Reviewer_pehy · 2025-08-05
> >
> > Thanks for your detailed response. After reading the rebuttal, I think most of my concerns have been addressed. I decided to raise my rating to 5.

---

### Official Review · Reviewer_QZNA · 2025-07-02

**Clarity:** 3
**Significance:** 2
**Originality:** 3
**Rating:** 4
**Confidence:** 4

**Summary:**

This paper studies the task vector method under the setting of ICL of linear regressions using linear transformers. The authors first show that if the $x_i$ and $y_i$ are separate tokens in the input, a critical point solution for transformers on the ICL loss is to mix up each $(x_i,y_i)$ pairs using the positional encoding (PE) at each layer and to perform GD++ after the first layer. Then they studied the setting where the input consists of triplets $(x_i,\to,y_i)$. In this setting, there is a critical point solution that not only performs the GD++ and $x,y$ pairs mixing, but also uses the arrow token to attend to all $x_i$ and $y_i$ to make linear combinations as new examples of linear regressions. The authors then argue that the activations of these arrows (after the 1st layer) could be used as the task vectors (Task V), which are inserted after query ICL example to improve zero-shot performance. The traditional Task V method only uses the activation vector of the last arrow vector and the authors propose to use the activation vectors of all arrows, which is termed Task V-M. The authors also empirically show that Task V method does not work well in the bijection tasks and the explaination relies on the conjecture that the task vector is a linear combination of demonstrations.

**Questions:**

1. Is there any constraint on $\Lambda_1$ in Theorem 1? It seems to me that $\Lambda_1=\begin{pmatrix} 0 & 1 \\\ 1 & 0 \end{pmatrix}$ would work while $\Lambda_1=I$ doesn't.
2. Line 135.  *"As a result, an L-layer linear transformer allocates one layer for embedding concatenation and utilizes the remaining L − 1 layers to perform gradient descent."* Seems like the concatenation is happening at each layer instead of just the first layer from the statement of theorem 1 where $D\in\mathcal{S}_P^L$. Is there an explaination on that?
3. Line 160. Is $\alpha_1$ equal to $\alpha_2$? In eq(10) they are equal but the Theorem 2 does not tell me anything about that.
4. Line 161. *"These vectors can then be injected into zero-shot prompts and function as single-token demonstrations"* Are these task vectors the hidden activations of arrows after the first layer or every layer? The discussions are clear to me if they are extracted after the 1st layer. It is a bit hard to understand if they are extracted in deeper layers.
5. Line 164. *"task vectors naturally emerge from the optimization dynamic"* Why would the transformers converge to such critical points? Specifically, in proposition 3 the $\lambda$ could be $0$ and it corresponds to the case where there is no task vectors. In my understanding, as the authors also mentioned in line 238, the task vectors do not help with minimizing the ICL loss and it is possible to converge to a critical point where there is no task vector.
6. The idea of bijection task is interesting. Is it possible to empirically show that Task V-M works for the bijection task as multiple task vectors (from multiple independent linear combinations) allow a $W$ of a higher rank.
7. For ICL of linear regressions, it is somewhat reasonable to conjecture that task vectors are linear combinations of demonstrations. It is not clear to me if this conjecture makes sense for non-linear tasks. The linear combinations of demonstrations are easy to obtain only after the 1st layer if the 1st layer's $V$ matrix is identity. For tasks whose task vectors are extracted in deeper layers with non-linear transformers, the task vectors seem to be non-linear on the input. I'm wondering if the optimal depth to extract task vector is 1 for linear tasks, while it is deeper for non-linear tasks.
8. The message from Fig 3 is unclear to me.

I'm willing to raise the score if questions are well addressed.

**Ethical Concerns:**

["NO or VERY MINOR ethics concerns only"]

**Final Justification:**

The observations on bijection task are interesting. The main drawback is that the critical point analysis does not imply too much about emergence and dynamics. However this would not be a huge problem if the limitation is clearly stated in the paper. Hence I raise the score to borderline accept.

**Limitations:**

Yes.

**Paper Formatting Concerns:**

No such concern.

**Quality:**

3

**Strengths And Weaknesses:**

Strength:
1. This paper is of rich content: the analysis of loss landscapes for three settings of input, the analogy of the activation vector of arrows to the task vectors, the limitation of task vector method (bijection task) and empirical verifications.
2. The writing is mostly clear.

Weakness:
1. Most explainations work for the 2-layer transformer while they are not clear to me for deeper models. I didn't find any discussion on that.
2. It is not clear to me under what conditions the transformers will converge to the critical points that authors describe. There should be infinitely many critical points and some of them might be essentially different (please see the questions for details). This takes some more careful optimization dynamics analysis involving concrete training algorithms.

I didn't read the proofs in the appendix so it is possible that there are some strengths/weaknesses that I'm not aware of.

---

> ### Author Rebuttal · Authors · 2025-07-31
>
> We thank Reviewer QZNA for the constructive feedback and for recognizing the breadth and depth of our contributions. Below, we address each of the reviewer’s questions in detail:
>
> **Q1. The Strength of our Theoretical Results**
>
> In addition to the breadth of our empirical and theoretical contributions, we would like to highlight the technical depth and novelty of the proof techniques underlying our main results. Extending the analysis to pairwise and triplet-formatted demonstrations necessitated a careful construction and exploration of symmetry in both the input and parameter spaces, going beyond standard linear regression setups. To the best of our knowledge, our work provides the first rigorous theoretical framework that explains the emergence and functionality of task vectors, offering a principled understanding of their behavior, grounded in a formal analysis rather than empirical observation alone. We will make this point clearer in the revision.
>
> **Q2. Constraint on $\Lambda_1$**
>
> We agree with the reviewer that setting $\Lambda_1 = I$ would impede the model from performing the embedding concatenation necessary for ICL, as our analysis shows. More precisely, let $\Lambda_1 = \begin{pmatrix} a & b \\\\ c & d \end{pmatrix}$; then, at least one of $b$ or $c$ must be non-zero to enable meaningful interaction between the covariates and responses.
>
> However, in Theorem 1, we characterize all possible critical points of the expected ICL risk, regardless of whether they solve the problem. Since $\Lambda_1 = I$ is indeed a valid critical point, even if meaningless, we intentionally do not exclude this case from our formal analysis. We will make this nuance clearer in the final version.
>
> **Q3. Layers for Embedding Concatenation**
>
> Indeed, embedding concatenation can occur at multiple layers in linear transformers. However, our theoretical focus is on the first layer because it plays a pivotal role in enabling subsequent layers to perform gradient-based updates. Once the covariate-response pairs $(x_i, y_i)$ are concatenated in the first layer, any additional mixing in later layers does not contribute further to task inference under the ICL risk, as it merely reprocesses already concatenated embeddings. Thus, only the first-layer concatenation directly facilitates gradient descent optimization.
>
> **Q4. $\alpha_1$ and $\alpha_2$**
>
> We apologize for the lack of clarity. $\alpha_1$ and $\alpha_2$ are not necessarily equal in practice. In eq. (10), we omit them to simplify notation. Since the effective coefficient matrix is given by $W' = \alpha_2 Y\beta (\alpha_1 X\beta)^\top$, the overall scale is determined by the product $\alpha_1 \alpha_2$, and their individual values do not affect the optimization dynamics or our conclusions. We will clarify this simplification in the revised manuscript.
>
> **Q5. Task Vector Injection Layers**
>
> The arrow-token hidden states after the first linear-attention layer are theoretically optimal for task vector extraction. This is because subsequent layers apply preconditioned gradient descent updates (i.e., on the Gram matrix $X^\top X$), which distort the original demonstration representations. Therefore, extracting task vectors after the first layer preserves a cleaner representation aligned with the base mapping $W$. In practical LLMs, task vectors are often extracted from deeper layers because the early transformer layers are responsible for preprocessing token representations.
>
> **Q6. Emergence of Task Vectors**
>
> We agree with the reviewer that Proposition 3 allows for the stationary solution $\lambda = 0$. However, in practice, we find this trivial solution is attained with almost zero probability. Due to random initialization and the presence of gradient noise in early training, small but nonzero $\lambda$ values emerge. Since these weights do not contribute to minimizing the ICL risk, they are not penalized during optimization (even under $L_2$ regularization), and thus persist during the pretraining stage.
>
> Once present, these nonzero $\lambda$ values tend to converge to nearby nontrivial stationary points described in Proposition 3. In our experiments with linear transformers, we have not observed convergence to $\lambda = 0$ or even $\lambda \approx 0$. Moreover, as shown in Proposition 6, the presence of external stochasticity (e.g., dropout) can further promote the persistence and stabilization of these weights, making the emergence of task vectors more robust in practical settings. We will incorporate the discussion in the revision.
>
> **Q7. Empirical Validation of High-Rank $W$**
>
> We appreciate the reviewer’s interest in the bijection task and the implications of our multi-vector injection strategy. To empirically verify that injecting multiple task vectors leads to a higher-rank coefficient matrix $W$, we conduct experiments using a 2-layer linear-attention model with hidden dimension $d = 4$. We vary the number of injected task vectors and report the top singular values of $W$ ($\lambda\_1$ to $\lambda\_4$), along with the corresponding ICL risk.
>
> | Num. of Vectors | $\lambda\_1$ | $\lambda\_2$ | $\lambda\_3$ | $\lambda\_4$ | ICL Risk |
> |---|---|---|---|---|---|
> | 1 | 0.2735 | 0 | 0 | 0 | 1.2639 |
> | 2 | 0.2142 | 0.1094 | 0 | 0 | 0.9843 |
> | 3 | 0.5616 | 0.2029 | 0.0482 | 0 | 0.8403 |
> | 4 | 0.7762 | 0.2822 | 0.0461 | 0.0090 | 0.7309 |
>
> These results clearly demonstrate that increasing the number of injected task vectors enhances the rank of \$W\$, thereby reducing the ICL risk, verifying our analysis. We will add this experiment in the revision, either in the main paper or the appendix, depending on space.
>
> **Q8. Regarding the Linear Combination Conjecture**
>
> We appreciate the opportunity to clarify a potential misunderstanding regarding what is being linearly combined. In our main conjecture, the task vector is formed by a linear combination of the **hidden states** corresponding to the input demonstrations, not their raw token embeddings. We do not claim that task vector formation excludes non-linearity; in fact, we believe non-linear pre- and post-processing layers are essential to enhance the ICL ability of LLMs.
>
> Our conjecture posits that practical LLMs perform task vector formation and prediction in middle transformer layers using mechanisms akin to those observed in linear-attention models, specifically embedding concatenation followed by gradient-like updates. The input $(x_i, y_i)$ pairs in our linear model analysis correspond to the **hidden states** of those tokens after early-layer processing in real-world LLMs. Therefore, while the optimal extraction depth is 1 in our linear setting, it lies deeper in real-world LLMs. This is consistent with our cross-validation results, where the empirically optimal extraction layer immediately follows the layer displaying the strongest weighted summation pattern (see Figure 3b).
>
> What we aim to emphasize with this conjecture is not the “linear combination” itself — an inherent property of attention mechanisms where hidden states are aggregated via attention-weighted summations over value projections, but rather the implication it carries: task vectors cannot encode the full information of the task. Instead, they represent at most a single, prototypical demonstration, limiting their intrinsic expressiveness and capacity to generalize to complex tasks.
>
> **Q9. Interpreting Figure 3**
>
> Figure 3 provides empirical support for our conjecture that practical LLMs adopt similar inner mechanisms to those observed in our linear-attention model analysis. Specifically, the inference stage of ICL consists of:
>
> * Early layers perform representation preprocessing;
>
> * Intermediate layers execute embedding concatenation across $(x_i, y_i)$ pairs;
>
> * Task vector then emerges via attention-weighted summation over these embeddings;
>
> * Final layers handle gradient-based updates and generate the prediction.
>
> For a conceptual overview, please refer to Figure 1. Figure 3a and 3b visualize saliency maps that clearly show embedding concatenation and task vector formation patterns. In particular, we observe that each $y_i$ token attends to its corresponding $(x_i, y_i)$ pair (concatenation), and the final arrow token aggregates across all $y_i$ tokens (weighted summation), closely mirroring our main conjecture. Together, these findings suggest that real-world LLMs implement structurally similar learning procedures for ICL. We will add the above description in the revision.

---

> > ### Comment · Reviewer_QZNA · 2025-08-02
> > **Some follow ups**
> >
> > 1. I appreciate the new experiments on empirical validation of high-rank $W$. Is it conducted on bijection task?
> >
> > 2. It is reasonable that the task vector emerges with high probability in practice due to randomness in training. But I don't think this is sufficiently implied by the theory part of the current paper since most of the theoretical results are on the critical points caculations. Thus the following statement seems overclaiming.
> > >To the best of our knowledge, our work provides the first rigorous theoretical framework that explains the emergence and functionality of task vectors ......
> >
> > Another similar issue is the constraint of $\Lambda_1.$ A valid $\Lambda_1$ might emerge in experiments but this emergence cannot be characterized by critical points caculations. This could weaken the theoretical signifigance of this paper.

---

> > > ### Author Response · Authors · 2025-08-03
> > > **Re: Some follow ups**
> > >
> > > We thank the reviewer for the thoughtful follow-up. Please find our detailed responses below:
> > >
> > > **Q1. Empirical Validation of High-Rank $W$**
> > >
> > > Thank you for the clarification. The original experiment in the rebuttal was conducted on standard linear regression tasks with full-rank coefficient matrices. To better align with the reviewer’s question, we have rerun the experiments using a rank-2 $W$ matrix to simulate bijection-style tasks more closely. The revised results are shown below:
> > >
> > > | Num. of Vectors | $\lambda_1$ | $\lambda_2$ | $\lambda_3$ | $\lambda_4$ | ICL Risk |
> > > |---|---|---|---|---|---|
> > > | 1 | 0.6872 | 0 | 0 | 0 | 0.8106 |
> > > | 2 | 1.1335 | 0.0448 | 0 | 0 | 0.5725 |
> > > | 3 | 1.7636 | 0.1386 | 0 | 0 | 0.4969 |
> > > | 4 | 2.2156 | 0.2171 | 0 | 0 | 0.4727 |
> > >
> > > These results offer similar implications that injecting multiple task vectors increases the effective rank of the learned $W$, thereby improving inference performance—while still being bounded by the inherent rank ($2$) of the task.
> > >
> > > **Q2. On the Emergence of Task Vectors and $\Lambda_1$**
> > >
> > > We agree with the reviewer that our theoretical results are restricted to critical point analyses, which are weaker than convergence guarantees or sample complexity bounds. However, we also emphasize that analyzing the learning dynamics of multi-layer transformers is extremely hard, with the current results primarily limited to simplified settings like one-layer transformers. Within this context, we believe our current results offer meaningful progress: the derivation of structured critical points for multi-layer, linear-attention transformers under realistic prompt formats.
> > >
> > > We also acknowledge that our results cannot fully capture every single aspect of task vector behavior in practice. But still, the current results provide a coherent and empirically grounded explanation for key observed phenomena, including how task vectors form via embedding concatenation and support gradient-like inference.
> > >
> > > To more accurately reflect the scope of our contribution, we will revise the original claim to:
> > >
> > > > Our work provides the first rigorous theoretical framework to explain the inner mechanisms of task vectors, providing a principled understanding of how they emerge and function in attention-based models.
> > >
> > > We hope this refinement better balances between theoretical significance and empirical grounding, and we thank the reviewer for encouraging a more precise framing.

---

### Official Review · Reviewer_tBHD · 2025-07-02

**Clarity:** 1
**Significance:** 3
**Originality:** 2
**Rating:** 3
**Confidence:** 3

**Summary:**

The paper theoretically analyses the emergence of task vectors in bidirectional linear self-attention models trained on in-context linear regression tasks.
It finds that when data points are presented in-context as three tokens (x, ->, y) instead of two tokens (x, y), task vectors form as a weighted summation of all demonstrations.
As a result, task vectors are limited in expressivity since they can only realize rank-one coefficient matrices in the linear regression setting.
Based on this insight, the paper conjectures that LLMs using task vectors should struggle on bijection tasks, e.g. $y = W x$ and $x = W y$, which can in general not be solved with rank-one matrices $W$.
This conjecture is validated empirically on pretrained large language models showing degraded performance on bijection tasks using task vectors while in-context learning performance is relatively high.

**Questions:**

- What is the alternative hypothesis to the linear combination conjecture? What evidence would have disproved it? I am wondering this especially since the theoretical model is a stack of linear attention layers with skip connections without nonlinearities. Linear attention is basically a weighted combination of other tokens so I fail to see what type of result this hypothesis excludes.

- As a follow-up to the previous questions, is there some meaning to the weights in this weighted combination? Maybe an even stronger empirical consequence of your result would be to show that task vectors can in practice be created without running the model simply by adding the examples as suggested by the finding? What prevents demonstrating this?

- I am confused how to reconcile the results presented in Table 1 and Table 2. Table 1 using Llama-7B seems to suggest that task vectors do not work for bijection tasks whereas Table 2 using Llama-13B shows that bijection tasks can be solved with high accuracy using task vectors (TaskV rows).

> Notably, the structure of $S_P$ closely aligns with our visualization of $D_l$ in Figure 2b, confirming our theoretical analysis

- I am not sure I can follow this statement. What I can clearly see in the attention pattern is the concatenation operation. What other properties predicted by Theorem 2 are visible?

- In the caption of Figure 3, you use a blue rectangle to refer to the ith demonstration but I am not sure what it visually corresponds to in the figure?

> These vectors serve as redundancy against information loss induced by dropout, thereby improving robustness.
> [...] While dropout is not always applied during LLM pretraining or fine-tuning, the injection of position encodings and use of normalization act as alternative sources of perturbation, thereby promoting the emergence of such redundancy.

- I am not sure I follow this line of argument. Token-wise dropout is very uncommon as far as I know but I am not sure why position encodings or normalization would introduce a similar pressure to copy information. Could you elaborate on this point, i.e. how would they create a pressure to redundantly copy information? In general, I am surprised by the prominence of the conclusion that task vectors serve as a redundancy against dropout given that this is a relatively minor component of your theoretical investigation.

- To what extent does the analysis rely on using the expected ICL risk instead of the more common autoregressive loss? I am asking this primarily out of curiosity whether you have considered it and less because I think it is necessarily a problem. However, I am wondering to what extent having a causal mask might change the main conclusions?


## Suggestions

- I find it misleading to refer to the models consisting of purely linear self-attention as (linear) transformers (e.g. L136) given that they lack the feedforward blocks. One suggestion would be to simply call them linear self-attention model instead.

- I found it confusing that section 5 was called "Further Discussions". Based on this non-descriptive title and the position in the paper, I was expecting a related work section (which is missing). Maybe incorporating it into section 4 or finding a more expressive title would be helpful to guide the reader.

> As a result, an L-layer linear transformer allocates one layer for embedding concatenation and utilizes the remaining L − 1
layers to perform gradient descent.

- The split into a copy/concatenation operation and GD operation is similarly found in e.g. [1]. It think this should be mentioned.

[1] Uncovering mesa-optimization algorithms in Transformers, von Oswald et al., 2023

**Ethical Concerns:**

["NO or VERY MINOR ethics concerns only"]

**Final Justification:**

I thank the authors for their responses that have helped answer my questions.
I think there are findings in this submission that have the potential to make a meaningful contribution to understanding task vectors but unfortunately I believe the paper needs to be fundamentally restructured.

There are several findings I believe are interesting and potentially impactful:

- The theoretical analysis of linear attention models in an ICL linear regression setting suggests that task vectors are limited to expressing rank-one coefficient matrices
- The implication that task vectors should therefore struggle with bijection tasks is found empirically to be true
- The saliency map analysis of an LLM in Figure 3 (which one? I could not find this information) reveals copying and aggregation structure inside a transformer that provides insights into how task vectors are formed

Unfortunately the current structure of the paper buries the lead in mathematical notation and a heavy focus on what is called the "linear combination conjecture". The authors promise to make changes in this regard but I think there are too many fundamental changes needed, requiring an entirely new review.

- I made this point in more detail in my review but the paper analyses a stack of linear attention layers. Linear attention is a linear combination of the original demonstrations. The "linear combination conjecture" in the submission states "The injected task vector functions as a single in-context demonstration, formed through a linear combination of the original demonstrations (hidden states)". This statement is simply too close to the definition of linear attention in my opinion. There is more structure in the finding though (copying mechanism, what tokens are being aggregated, task vector can only express rank 1 weight) that could be worked in to a hypothesis
- The failure of task vectors in LLMs on bijection tasks has potential to be strong evidence but it needs a more elaborate empirical analysis. For example, how exactly does it fail? Can it be fixed, i.e. how does ICL represent the information that allows it to solve the task that is not captured by the task vector?
- There is no dedicated related work section. The authors state that due to space constraints related work was only mentioned in the introduction but I do not think that is sufficient. They promise to change this but this is not a minor modification in my eyes and would require a proper re-reading
- The limitations are only discussed deep in the appendix (D.3) and the paper should be restructured to allow adding it to the main text.

Taken together I would like to encourage the authors to rethink the structure of the paper. This work has potential but I do not think it is ready for acceptance, even with the changes promised by the authors.

**Limitations:**

A discussion of limitations is missing. I would have for instance expected some contextualization of the possible caveats of trying to extrapolate the results to full transformers given the simplified theoretical model (beyond the fact that this type of model has been considered by prior work).

**Quality:**

2

**Strengths And Weaknesses:**

The paper makes an interesting case for understanding task vectors through its theoretical analysis that leads to the surprising finding that task vectors do not work well on bijection tasks.
Understanding how and why task vectors work has the potential to significantly improve our understanding of transformer models.
Unfortunately, the paper lacks clarity from the way it is structured and how it presents the main conclusions from its theoretical study.
I am not very convinced by what is positioned as the main hypothesis, namely "_The injected task vector functions as a single in-context demonstration, formed through a linear combination of the original demonstrations (hidden states)._", given that the analyzed models are stacked, bidirectional linear self-attention models which -- as far as I understand, see questions -- almost by definition satisfy this hypothesis.
Importantly, a dedicated related work section is completely missing and the result is not sufficiently contextualized within the research context.

---

> ### Author Rebuttal · Authors · 2025-07-31
>
> We thank Reviewer tBHD for the insightful comments and for acknowledging the significance of our work in advancing the understanding of transformer models. Below, we address the reviewer’s questions in detail:
>
> **Q1. Regarding the Linear Combination Conjecture**
>
> We appreciate the opportunity to clarify a potential misunderstanding regarding what is being linearly combined. In our main conjecture, the task vector is formed by a linear combination of the **hidden states** corresponding to the input demonstrations, not their raw token embeddings. We do not claim that task vector formation excludes non-linearity; in fact, we believe non-linear pre- and post-processing layers are essential to enhance the ICL ability of LLMs.
>
> Our conjecture posits that practical LLMs perform task vector formation and prediction in middle transformer layers using mechanisms akin to those observed in linear-attention models, specifically embedding concatenation followed by gradient-like updates. The input $(x_i, y_i)$ pairs in our linear model analysis correspond to the **hidden states** of those tokens after early-layer processing in real-world LLMs. This interpretation is further supported by our saliency analysis, which reveals embedding concatenation and aggregation patterns in the middle layers of pretrained LLMs.
>
> What we aim to emphasize with this conjecture is not the “linear combination” itself — an inherent property of attention mechanisms where hidden states are aggregated via attention-weighted summations over value projections, but rather the implication it carries: task vectors cannot encode the full information of the task. Instead, they represent at most a single, prototypical demonstration, limiting their intrinsic expressiveness and capacity to generalize to complex tasks. We will clarify this in the revision.
>
> **Q2. Alternative Hypothesis for Task Vectors**
>
> To our knowledge, there is currently no established theoretical framework explaining how task vectors emerge and function. A commonly held belief is that task vectors directly encode the task mapping $f: \mathcal{X} \to \mathcal{Y}$ into a single vector [7]. However, this hypothesis fails to explain why task vectors underperform on simple bijection tasks, while standard ICL succeeds.
>
> Our work provides a refinement of this view: task vectors encode limited information, essentially equivalent to a single demonstration, thus explaining their failure in tasks requiring higher-rank representations, such as bijections. We believe this insight is a substantial step toward understanding the internal mechanisms of transformer models. If we have misunderstood the reviewer’s comments, we would appreciate further clarification and would be happy to address them.
>
> **Q3. Availability of Running-Free Task Vector Extraction**
>
> Thank you for suggesting this creative approach. As noted above, since task vectors are formed via linear combinations of the model's internal hidden states, they inherently depend on processing the prompt through the model. Consequently, extracting task vectors without executing the model may not be feasible in general.
>
> Regarding the weights used in forming these linear combinations, our analysis (see Figure 3c) shows that causal attention induces an increasing weighting pattern (i.e., later demonstrations contribute more to the task vector). This finding aligns with our practical observations.
>
> **Q4. Table 1 and Table 2**
>
> Thank you for pointing out this possible source of confusion. The settings for Table 1 and Table 2 are indeed different:
>
> * Table 1: Task vectors are injected into zero-shot prompts (TV).
>
> * Table 2: Task vectors are injected into few-shot prompts (TaskV).
>
> Given this, it is not surprising that TaskV achieves higher performance on bijection tasks in Table 2 due to the extra information provided by few-shot demonstrations. In contrast, TaskV yields ~50\% accuracy on bijection tasks in the zero-shot setting (Table 2), consistent with random guessing. In the revision, we will make this distinction more explicit.
>
> **Q5. The structure of $\mathcal{S}_P$**
>
> We appreciate the reviewer’s careful examination of Figure 2b. As observed, each $3 \times 3$ sub-matrix in $D_l$ contains only two non-zero entries: the top and bottom cells in the second column, besides the concatenation patterns represented by $\Lambda_1$. This structural pattern aligns precisely with the $\Lambda_4 \otimes \Lambda_5$ component described in Theorem 2, which plays a central role in task vector formation.
>
> **Q6. Blue bars in Figure 3c**
>
> Thank you for raising this point. In Figure 3c, the blue bars correspond to the left y-axis $\\|z_{\text{tv}}^{-i} - z_{\text{tv}}\\|$, where we examine the contribution of each individual demonstration to the final task vector. For each demonstration, we introduce a perturbation and compute the deviation of the resulting task vector $z_{\text{tv}}^{-i}$ from the original $z_{\text{tv}}$. The observed pattern closely matches the theoretically predicted weight distribution shown by the orange line, thereby validating our analysis. We will clarify this in the revision.
>
> **Q7. Redundancy Against Dropout**
>
> We would like to clarify that we do not claim dropout as the main cause of task vector emergence, but rather that it amplifies the tendency towards such redundancy. Importantly, the $\Lambda_4$ weights for task vector formation emerge naturally because of the random initialization and noisy gradients in early updates, even without dropout. As these weights do not contribute to the ICL risk, they are not penalized during optimization and persist as part of the model’s learned structure.
>
> We adopt token-wise dropout mainly due to its theoretical simplicity, and to intuitively demonstrate how random information loss promotes redundancy. While we acknowledge that token-wise dropout is rare in practical LLMs, we believe the broader insight generalizes: any mechanism that perturbs hidden states during training can push toward such redundancy.
>
> In this spirit, we mention positional encodings and normalization layers as alternative sources of perturbations in the hidden state space that introduce extra scaling and shifting. These facts may subtly distort hidden states and lead to the same conclusion.
>
> Note that these analyses are not the core contribution of our paper, but are offered as a potential supporting mechanism that may promote the emergence of task vectors. We will revise the manuscript to make this distinction clearer.
>
> **Q8. Expected ICL Risk vs. Auto-regressive Loss**
>
> We appreciate this insightful distinction. Our current analysis focuses on the ICL setting commonly used in recent works on task vectors [7, 12, 16], where prediction is modeled as a single-token generation task. It would be interesting to investigate whether our analysis could be extended to the auto-regressive loss.
>
> Firstly, we consider combining the causal mask with the current expected ICL risk. We believe the current techniques used in our proof can be extended to causal linear attention after minor modifications. In particular, we expect that the structures of $A_l$, $B_l$, and $C_l$ remain unchanged, while $D_l$ becomes block-diagonal with upper-triangular sub-blocks. Therefore, our main conclusions are expected to hold under causal attention.
>
> Further extension to auto-regressive loss is equivalent to optimizing the ICL risk over varying numbers of demonstrations in expectation. Since our current results characterize the critical points for arbitrary $n$, we believe they also provide insight into the stationary behavior of such auto-regressive training. One expected difference lies in $\mathcal{S}_P$, such that the diagonal sub-blocks may no longer be identical.
>
> We have conducted preliminary experiments using causal linear attention, and the learned parameters match our analysis above. Due to rebuttal constraints, we cannot include additional plots, but we will add these results and visualizations in the revised version.
>
> **Q9. Related Works and Other Suggestions**
>
> We thank the reviewer for these helpful suggestions. In response, we have: (1) renamed the model to improve clarity; (2) restructured Section 5 for better organization and narrative flow; and (3) added the missing citation. Due to space constraints, we initially incorporated discussions of related work into the introduction. Following the reviewer’s suggestion, we will reorganize these discussions into a dedicated Related Work section in the revised version.
>
> **Q10. Limitations of our Work**
>
> We want to clarify that the limitation discussion is provided in Appendix D.3 due to limited space. Our analysis is conducted on simplified settings, particularly linear-attention models, which may not capture the full complexity of real-world LLMs. However, the primary goal of our work is not to exhaustively model practical LLMs, but to provide clear, mechanistic insights into the emergence, functionality, and limitations of task vectors in ICL. Our theoretical results offer several actionable contributions:
>
> * Our analysis of embedding concatenation helps identify the optimal layer for task vector extraction.
>
> * The bijection task counterexample clarifies when and why task vectors fail, despite ICL success.
>
> * Our study of attention weights helps determine the preferred ordering of in-context demonstrations.
>
> * The validation of multi-vector injection highlights how task vectors can be extended to few-shot settings.
>
> * Our critical point analysis sheds light on the conditions that encourage task vector emergence during fine-tuning.
>
> We believe that, while these insights are grounded in a tractable, simplified theoretical model, the conclusions drawn from it provide valuable guidance for better understanding and improving the ICL capabilities of modern transformer-based models. We will add these discussions and move the limitation section to the main paper in the revision.

---

> > ### Comment · Reviewer_tBHD · 2025-08-01
> > **Follow-up discussion**
> >
> > Thank you for your response, please allow me to follow up on a few points.
> >
> > > We appreciate the opportunity to clarify a potential misunderstanding regarding what is being linearly combined. In our main conjecture, the task vector is formed by a linear combination of the hidden states corresponding to the input demonstrations, not their raw token embeddings.
> >
> > Yes, I understand that, but thank you for making sure there was no misunderstanding.
> >
> > > Our conjecture posits that practical LLMs perform task vector formation and prediction in middle transformer layers using mechanisms akin to those observed in linear-attention models, specifically embedding concatenation followed by gradient-like updates.
> > > What we aim to emphasize with this conjecture is not the “linear combination” itself — an inherent property of attention mechanisms where hidden states are aggregated via attention-weighted summations over value projections, but rather the implication it carries: task vectors cannot encode the full information of the task.
> >
> > I think these statements contain much more information and would help to make the hypothesis more precise. But this is not what you state in the "linear combination conjecture": _The injected task vector functions as a single in-context demonstration, formed through a linear combination of the original demonstrations (hidden states)._ Do you have an improved statement in mind for the revision that you could share here?
> >
> > > Note that these analyses are not the core contribution of our paper, but are offered as a potential supporting mechanism that may promote the emergence of task vectors.
> >
> > I agree, so it would probably be best to remove / rephrase it in the first of your two paper highlights in the introduction where you state in lines 47-48: _These vectors serve as redundancy against information loss induced by dropout, thereby improving robustness_

---

> > > ### Author Response · Authors · 2025-08-01
> > > **Re: Follow-up discussion**
> > >
> > > Thank you for the thoughtful follow-up. Below are responses and revisions addressing the points you raised:
> > >
> > > **Q1. Refined Conjecture Statement**
> > >
> > > We agree that the current statement may be overly simplistic or imprecise. To better reflect the intended scope and implications, we propose the following improved version for the revision:
> > >
> > > > **Conjecture (Task Vectors as Representative Demonstrations)**: The injected task vector facilitates gradient-based inference in attention-based models by encoding a single representative demonstration distilled from the original in-context examples.
> > >
> > > We will also incorporate further explanations for the core implications of this main conjecture in the introduction to facilitate understanding.
> > >
> > > **Q2. Rephrased Highlight Sentence**
> > >
> > > We agree that the current highlight may be misleading, since our core claims are robust to its presence or absence. Instead, we propose the following revision to this highlight:
> > >
> > > > These vectors naturally emerge in attention-based models trained via gradient-based optimization, potentially enhancing robustness under representational perturbations by redundantly encoding task information.
> > >
> > > These improvements will be reflected in our revision. Please let us know if there are any further questions or concerns, and we will be happy to address them.

---

### Official Review · Reviewer_eQNr · 2025-07-03

**Clarity:** 4
**Significance:** 3
**Originality:** 3
**Rating:** 5
**Confidence:** 3

**Summary:**

This paper proposes the Linear Combination Conjecture as a mechanism by which task vectors are formed from linear combinations of in-context example demonstrations. The authors use this assumption to derive the emergence of task vectors by analyzing critical points of linear transformers trained on in-context learning (ICL) triplet examples for random linear regression. Next, the paper correctly predicts that modern LLMs with task vectors perform poorly on bijection tasks. Finally, this conjecture is utilized to add multiple task vectors to a few-shot prompt, resulting in improved performance.

**Questions:**

1. Are Tables 1 and 2 directly comparable? For bijection tasks, TaskV performs better than the ICL Baseline in Table 2 but worse than ICL in Table 1.
2. In addition to Sections 6.3 and C.2, another multi-vector variant could extract multiple vectors from each arrow in a few-shot prompt and then inject them into the same corresponding positions. Would this provide additional insight to support or refute the Linear Combination Conjecture?

**Ethical Concerns:**

["NO or VERY MINOR ethics concerns only"]

**Final Justification:**

The author rebuttal addressed all of my concerns. After reading other reviewers' comments I am keeping my initial score and recommending Accept.

**Limitations:**

yes

**Paper Formatting Concerns:**

No formatting issues

**Quality:**

4

**Strengths And Weaknesses:**

### Quality
- Proofs and experimental methodology appear to be correct
- Each of these points provide evidence supporting the central claim of Linear Combination Conjecture

### Significance
- Important contribution that increases understanding of in-context learning, a fundamental component of many LLM applications.
- Potentially high impact if it leads to follow up works that further enhance the task vector method

### Originality
- This paper seems to build on existing work (task vectors, loss landscape analysis) in a novel manner to produce novel insights.

### Clarity
- Generally well-organized and a pleasure to read. Ideas are presented clearly with sufficient background information
- Notation for dimensions is confusing at times, in particular the relationship between $d_p$ (number of tokens) and $n$ (number of examples) could be more explicit.

- Typo: In the equation below line 402, I believe $\tilde{R}_i$ should be $R_i$

---

> ### Author Rebuttal · Authors · 2025-07-31
>
> We thank Reviewer eQNr for the thoughtful feedback and for recognizing the importance of our contribution to the understanding of ICL. Below, we address the reviewer’s comments and questions in detail:
>
> **Q1. Relation Between $d_p$ and $n$**
>
> We apologize for the confusion here. The value of $d_p$ depends on the structure of the input prompt, as summarized below:
>
> | Demonstration | $d_p$ |
> |:---:|:---:|
> | Single | $n+1$ |
> | Pairwise | $2n+2$ |
> | Triplet | $3n+3$ |
>
> We will improve the presentation by clarifying the explicit relations between $n$ and $d_p$ to avoid confusion at their first appearance.
>
> **Q2. Typo: $\tilde{R}_i$ Should Be $R_i$**
>
> Thank you for catching this typo. We have corrected it in the revised version.
>
> **Q3. Clarification on Table 1 and Table 2**
>
> The performance of ICL (in Table 1) and Baseline (in Table 2) should not be directly compared, as they are based on different prompt settings. Specifically:
>
> * ICL in Table 1 uses many-shot prompts ($n=10$ demonstrations), while task vectors are injected into zero-shot prompts. As expected, task vectors underperform standard ICL, since they serve primarily as an acceleration mechanism.
>
> * Baseline in Table 2 uses few-shot prompts (ranging from 0 to 4 demonstrations), the same setting for task vector injection. These results show that injecting task vectors improves ICL performance under these constrained settings.
>
> We will make this difference more explicit in the revision.
>
> **Q4. The Multi-Vector Variant**
>
> Thank you for highlighting the alternative method of extracting multiple task vectors from a single prompt. This approach is indeed consistent with our theoretical findings. As discussed in Proposition 3, the task vector weights that emerge at each arrow ($\to$) token are approximately orthonormal, suggesting they encode distinct information subsets and can be simultaneously injected to enhance model performance (e.g., by increasing the rank of the induced coefficient matrix $W$).
>
> We have also tried this method before. Below is a comparison between the current multi-vector method (TaskV-M) and this single-prompt variant (TaskV-MS) under a 4-shot setting:
>
> | Method | Knowledge | Algorithmic | Translation | Linguistic | Bijection | Average |
> |---|---|---|---|---|---|---|
> | Baseline | 84.80 | 88.07 | 83.27 | 88.89 | 67.16 | 81.52 |
> | TaskV | 88.70 | 89.53 | 86.27 | 92.76 | 70.44 | 84.66 |
> | TaskV-M | 89.60 | 91.00 | 87.20 | 92.36 | 72.53 | 85.64 |
> | TaskV-MS | 90.10 | 90.67 | 87.00 | 92.22 | 72.09 | 85.45 |
>
> While TaskV-MS also delivers strong performance, it slightly underperforms TaskV-M. We believe this is due to the causal attention mechanism in real LLMs, where earlier arrow tokens can only aggregate information from a subset of demonstrations. Nonetheless, TaskV-MS is a promising alternative for accelerating inference. We will add these results as part of the revised paper.

---

> > ### Comment · Reviewer_eQNr · 2025-08-05
> > **Follow up**
> >
> > Thank you for addressing all of my questions and comments.

---

### Official Review · Reviewer_jCtd · 2025-07-05

**Clarity:** 2
**Significance:** 3
**Originality:** 3
**Rating:** 4
**Confidence:** 2

**Summary:**

This paper investigates the underlying mechanisms of task vectors, a technique used to accelerate inference in in-context learning (ICL) by condensing task-specific information into a single reusable representation. The authors propose the Linear Combination Conjecture, which suggests that task vectors function as single in-context demonstrations formed via linear combinations of original demonstrations. The paper provides both theoretical justification and empirical evidence, showing that task vectors naturally arise in linear transformers trained on triplet-formatted prompts, as revealed through loss landscape analysis. The authors also predict and confirm that task vectors struggle to represent high-rank mappings in practice. Additional validation through saliency analysis and parameter visualization supports their conclusions. To address limitations, the study proposes injecting multiple task vectors into few-shot prompts to improve performance.

**Questions:**

See the weaknesses.

**Ethical Concerns:**

["NO or VERY MINOR ethics concerns only"]

**Final Justification:**

After reading the rebuttal and the comments from other reviewers, I have decided to maintain my original score.

**Limitations:**

A discussion on the potential limitations of the method would strengthen the paper.

**Quality:**

3

**Strengths And Weaknesses:**

Strengths:
The paper presents a strong theoretical foundation to support the proposed Linear Combination Conjecture, offering valuable insights into the formation and function of task vectors. The authors conduct comprehensive zero-shot and few-shot experiments, which effectively validate the core ideas of the work across practical settings.

Weaknesses:
The paper does not discuss its limitations, leaving questions about the generalizability and constraints of the proposed approach.
The heavy use of mathematical notation and formulas may hinder readability and make the paper difficult to follow for some readers.
It remains unclear whether the proposed method is applicable to multimodal task.

---

> ### Author Rebuttal · Authors · 2025-07-31
>
> We thank Reviewer jCtd for the constructive comments and for acknowledging the significance of our theoretical analysis and its insights into understanding task vectors. Below, we respond to the reviewer’s specific concerns:
>
> **Q1. Discussion of Potential Limitations**
>
> We want to clarify that the limitation discussion is provided in Appendix D.3 due to limited space. We acknowledge that our theoretical analysis focuses on a simplified linear-attention transformer architecture, which may not fully encapsulate the complexity of real-world LLMs. However, we provide several strong empirical observations that support the practical relevance of our theory: (1) consistent failure of task vectors on bijection tasks, (2) matching saliency patterns in real LLMs that align with our theoretical predictions, and (3) consistent improvements from multi-vector injection. Together, these results offer meaningful insight into the emergence, functionality, and limitations of task vectors in practical transformer models.
>
> **Q2. Heavy Use of Mathematical Notation**
>
> To improve accessibility, we will add a dedicated notation table in the appendix to clearly define all symbols used throughout the paper. Additionally, we emphasize key takeaways and theoretical insights in bold text to help readers grasp the main conclusions. We will also add summaries and interpretations in plain language to help readers understand the key ideas easily.
>
> **Q3. Extension to Multimodal Settings**
>
> While our current analysis primarily focuses on text-based LLMs, it naturally extends to any ICL scenario where the input follows a triplet-formatted structure (e.g., "a $\to$ b, c $\to$ d, e $\to$"). This includes certain multimodal models that adopt similar prompting schemes, such as the one proposed in [A]. In particular, we believe our multi-vector injection strategy applies to multimodal task vectors. Formally extending our theoretical framework to the multimodal setting is an exciting and valuable direction. We have noted this in our future work.
>
> [A] Huang Brandon, et al. Multimodal task vectors enable many-shot multimodal in-context learning, 2024.

---

### Note · Authors · 2025-08-11

We sincerely thank all reviewers and the area chair for their thoughtful evaluations, constructive feedback, and engagement throughout the discussion phase. We are especially grateful for the recognition of our paper’s contributions toward deepening the theoretical and empirical understanding of task vectors in ICL.

Multiple reviewers acknowledged the novelty and rigor of our theoretical framework, the quality of our empirical analyses, and the paper’s potential to shed light on the emergence and limitations of task vectors in attention-based models. We are encouraged that reviewers appreciated our use of linear transformer models to derive insights that are consistent with patterns observed in practical LLMs (e.g., the gradient-based inner inference mechanism, the role of task vectors as representative demonstrations, and the proposal of multi-vector injection strategies).

During the rebuttal, we addressed the major concerns:

* We clarified our main conjecture, shifting emphasis away from the “linear combination” wording toward the core implication that task vectors encode a single representative demonstration.
* We provided new empirical results showing how injecting multiple task vectors improves the effective rank of the coefficient matrix $W$, especially for bijection tasks.
* We acknowledged the theoretical limitations of critical point analysis and clarified that our results provide mechanistic insight rather than full convergence guarantees.
* We revised overstatements about dropout, clarifying its role as one of several factors that may promote redundancy, not a central claim.
* We addressed other questions concerning $\Lambda_1$, block-diagonal structure, and applicability to causal and multimodal settings.

For the final version, we will:

* Refine our conjecture statement and integrate clearer motivation and implications in the introduction.
* Rephrase overclaimed highlights and limit theoretical assertions to what is strictly supported by our analysis.
* Incorporate additional experiments from the rebuttal (e.g., bijection and causal settings).
* Add a dedicated related work section, an appendix of key notations, and visually emphasize core theoretical conclusions for clarity.

We believe the final version of our paper will benefit greatly from these suggestions and will serve as a meaningful contribution to the community’s understanding of ICL and task vector mechanisms.

---

### Decision · Program_Chairs · 2025-09-17

**Decision:**

Reject

**Comment:**

This submission presents a theoretical analysis of the emergence of task vectors in a linear self-attention model for in-context regression. The landscape analysis shows that the task vector is a weighted sum of the in-context demonstrations, implying a loss of information when the task is not low-rank. To support this claim, the authors provide empirical evidence that task vectors fail to represent bijective tasks in LLMs.
The submission received mixed reviews. While reviewers appreciated the insight that task vectors cannot solve bijective tasks, they raised concerns about (i) the missing related work section, (ii) the absence of convergence guarantees beyond critical point analysis, (iii) the limited implications of the theoretical results given that linear transformers inherently satisfy the linear combination property, and (iv) the relatively weak empirical validation. The meta-reviewer encourages the authors to carefully revise the manuscript and address the raised concerns, and resubmit to a future venue.